# Development and evaluation of $CO_2$ transport in MPAS-A v6.3

Tao Zheng[1,2], Sha Feng[3,*], Kenneth J. Davis[3], Sandip Pal[4], and Josep Anton Morguí[5]

[1]Department of Geography and Environmental Studies, Central Michigan University, Mount Pleasant, MI. USA
[2]Institute for Great Lakes Research, Central Michigan University, Mount Pleasant, MI. USA
[3]Department of Meteorology and Atmospheric Science, The Pennsylvania State University, University Park, PA. USA
[4]Department of Geosciences, Texas Tech University, Lubbock, TX. USA
[5]Environmental Science and Technology Institute, Universitat Autònoma de Barcelona, ICTA-UAB, Bellaterra, Spain
[*]Now at Atmospheric Sciences and Global Change Division, Pacific Northwest National Laboratory, Richland, WA, USA

*Correspondence to:* Tao Zheng (zheng1t@cmich.edu)

**Abstract.**

Chemistry transport models (CTMs) play an important role in understanding fluxes and atmospheric distribution of carbon dioxide ($CO_2$). They have been widely used for modeling $CO_2$ transport through forward simulations and inferring fluxes through inversion systems. With the increasing availability of high resolution observations, it has been become possible to

estimate $CO_2$ fluxes at higher spatial resolution. In this work we implemented $CO_2$ transport in Model Prediction Across Scales-Atmosphere (MPAS-A). The objective is to use the variable-resolution capability of MPAS-A to enable high resolution $CO_2$ simulation at limited region with a global model. Treating $CO_2$ as an inert tracer, we implemented in MPAS-A (v6.3) the $CO_2$ transport processes, including advection, vertical mixing by boundary layer scheme, and convective transport. We first evaluated the newly implemented model's tracer mass conservation and then its $CO_2$ simulation accuracy. A one-year (2014)

MPAS-A simulation is evaluated at the global scale using $CO_2$ measurements from 50 near-surface stations and 18 Total Carbon Column Observing Network (TCCON) stations. The simulation is also compared with two global models: National Oceanic and Atmospheric Administration (NOAA) CarbonTracker v2019 (CT2019) and European Center for Medium-Range Weather Forecasts (ECMWF) Integrated Forecasting System (IFS). A second set of simulation (2016-2018) is used to evaluate MPAS-A at regional scale using Atmospheric Carbon and Transport-America (ACT-America) aircraft $CO_2$ measurements over the

eastern United States. This simulation is also compared with CT2019 and a 27-km WRF-Chem simulation. The global scale evaluations show that MPAS-A is capable of representing the spatial and temporal $CO_2$ variation with comparable level of accuracy as IFS of similar horizontal resolution. The regional scale evaluations show that MPAS-A is capable of representing the observed atmospheric $CO_2$ spatial structures related with the mid-latitude synoptic weather system, including the warm versus cold sector distinction, boundary layer to free troposphere difference, and frontal boundary $CO_2$ enhancement. MPAS-

A's performance in representing these $CO_2$ spatial structures are comparable with the global model CT2019 and regional model WRF-Chem.

# 1  Introduction

Carbon dioxide ($CO_2$) is the most important greenhouse gas, and our knowledge about its sources and sinks still have large gaps. Inversion systems are tools for inferring surface $CO_2$ fluxes based on observations and chemistry transport models (CTMs). Two types of CTMs are commonly used: global models and regional models. Global models are commonly used for inferring $CO_2$ fluxes at coarse spatial scales (Patra et al., 2008; Schuh et al., 2019; Jacobson et al., 2007, 2020). With the fast increasing number of atmospheric $CO_2$ observations, including those acquired by ground based, airborne, and satellite instruments, regional inversion system have been developed and applied to estimate carbon fluxes at higher resolution (Gerbig et al., 2009; Pillai et al., 2012; Lauvaux et al., 2012; Hu et al., 2019; Zheng et al., 2018, 2019).

A major challenge of atmospheric $CO_2$ inversion modeling is how to partition the model-data mismatch (MDM) among the transport model error, observation error, and prior flux error (Baker et al., 2006). In the Bayesian inversion framework, the error covariance matrix $\mathbf{R}$ is commonly used to represent the combined error of transport model and observations. While it is important to correctly represent the transport model error in an inversion system, it is also important to reduce the error in order to estimate the fluxes with less uncertainty. One approach to reduce the transport model error is to increase the horizontal resolution of a simulation. For instance, Feng et al. (2016) found high-resolution WRF-Chem simulation improved $CO_2$ model-data comparison because of better resolved planetary boundary layer (PBL) and better representation of spatial variability of $CO_2$ fluxes. In a recent study, Agusti-Panareda et al. (2019) investigated the impacts of transport model's horizontal resolutions on simulated $CO_2$ accuracy, and they found that $CO_2$ variability is generally better represented by higher resolution simulations.

Global high resolution $CO_2$ simulations require large computational resources. Regional (limited area) models, which have lower computational cost than their global model counterpart at the same horizontal resolution, are often used for high resolution $CO_2$ transport (Feng et al., 2016; Diaz-Isaac et al., 2019, 2018) and inverse modeling (Sarrat et al., 2007; Gerbig et al., 2008; Lauvaux et al., 2012; Zheng et al., 2019). However a regional model requires $CO_2$ transported from outside its model domain to be prescribed. For a $CO_2$ inversion system, having lateral boundaries increase the size of the control vector to be optimized (Rayner et al., 2019). A number of approaches have been applied to the $CO_2$ lateral boundary problem, such as assuming the boundary inflow is perfectly known (Gockede et al., 2010), correcting the lateral boundary condition using observation prior to inversion (Lauvaux et al., 2012; Schuh et al., 2013), or jointly optimizing flux and lateral boundary condition (Zheng et al., 2018). When $CO_2$ lateral boundary is optimized, an inversion system adjusts its $CO_2$ fields at the boundary prescribed by a parent global model in addition to adjusting surface fluxes. This could be problematic for inversion systems that use satellite derived column averaged $CO_2$ measurements ($XCO_2$) because model-data mismatches in the free troposphere (FT) are often originated from outside a regional model's limited area domain (Feng et al., 2019; Lauvaux and Davis, 2014).

The objective of the present paper is to provide an alternative high-resolution $CO_2$ transport modeling approach to regional transport models. This approach is to use a global variable-resolution model which allows for local grid refinement that enables

high resolution simulation over an interested region without incurring the prohibitively high computational cost or the lateral boundary condition. Variable-resolution through local grid refinement has been widely used in Numerical Weather Prediction (NWP) models, such as MPAS-A (Skamarock et al., 2012), Ocean-Land-Atmosphere Model (OLAM) (Walko and Avissar, 2008a, b), Energy Exascale Earth System Model (E3SM) (Golaz et al., 2019), and Finite-Volume Cubed-Sphere model (FV3)
(Putman and Lin, 2007). One benefit of local mesh refinement is enabling regional high-resolution modeling without incurring the lateral boundary condition and its associated problems, such as solution mismatches between the driving global model and the evolving regional model (Davies, 2014). Model Prediction Across Scales-Atmosphere (MPAS-A) is a fully compressible non-hydrostatic global atmospheric model which uses finite-volume numeric solver discretized on centroidal Voronoi mesh with C-grid staggering of its prognostic variables (Skamarock et al., 2012; Thuburn, 2007; Ringler et al., 2010). The centroidal
Voronoi mesh allows for local refinement and variable-resolution horizontal mesh which can be gradually changed from coarse to fine resolutions (Skamarock et al., 2012; Ringler et al., 2008).

To enable $CO_2$ transport modeling, we implemented atmospheric $CO_2$ transport processes, including advection, vertical mixing by PBL scheme, and convective transport in MPAS-A v6.3. Because the $CO_2$ transport processes are fully integrated
into the model's meteorological time steps, the resulting MPAS-A $CO_2$ is an online CTM. We used the newly developed model to conduct two sets of simulations over a 60-15 km variable-resolution global domain. Then the simulation results are evaluated using an extensive set of airborne observations over the eastern United States and near-surface observations from surface and tower stations across the globe. The simulation accuracy of MPAS-A is compared with three established $CO_2$ modeling systems based on the same observational data: WRF-Chem (Skamarock et al., 2008; Feng et al., 2019), Carbontracker (v2019,
CT2019 hereafter) (Jacobson et al., 2020), and ECMWF IFS (Agusti-Panareda et al., 2014, 2019).

## 2   Implementation of $CO_2$ transport in MPAS-A

This section describes the major modifications to MPAS-A that we made to implement $CO_2$ tracer transport. We represent $CO_2$ by its dry air mixing ratio ($q_{co_2}$) and model its atmospheric transport by adding its continuity equation in MPAS-A following
Eq. 7 of Skamarock et al. (2012).

$$\frac{\partial(\tilde{\rho} q_{co_2})}{\partial t} = -(\nabla \cdot \tilde{\rho} q_{co_2} \mathbf{V})_\zeta + F_{bl} + F_{cu} \qquad (1)$$

where $\tilde{\rho} = \rho_d/(\partial \zeta/\partial z)$, $\rho_d$ is dry air density, $\zeta$ is the vertical coordinate, $z$ is geometric height, $t$ is time, and $\mathbf{V} = (u, v, w)$ is the velocity vector ($u$, $v$, and $w$ are the zonal, meridional, and vertical wind respectively). The left hand side of the equation is the total $CO_2$ time tendency ($\partial(\tilde{\rho} q_{co_2})/\partial t$), and the first, second, and third terms on the right hand side represent the con-
tributions from advection, vertical mixing, and convective transport respectively. $CO_2$ tendency from advection is modeled in

flux form (Section 2.1), while tendency from vertical mixing ($F_{bl}$) and convective transport ($F_{cu}$) are modeled in uncoupled form ($\partial q_{co_2}/\partial t$) which are coupled to $\tilde{\rho}$ before being added to the total tendency. We choose to implement $CO_2$ vertical mixing in the Yonsei University (YSU) PBL scheme (Hong et al., 2006), and $CO_2$ convective transport in Kain-Fritsch (KF) scheme (Kain, 2004) because they are widely used in CTM and have been validated using observations (Borge et al., 2008; Hu et al., 2010; Kretschmer et al., 2012; Polavarapu et al., 2016). Details of the three terms on the right hand side of Eq. 1 are described in the following sections. We note that because the monotonicity constraint in the third-order scalar horizontal advection scheme (Skamarock and Gassmann, 2011) introduces dissipation MPAS-A does not use any explicit horizontal diffusion for scalar. Accordingly we did not include horizontal diffusion for $CO_2$.

## 2.1  $CO_2$ advection

Advection is the most significant component of $CO_2$ atmospheric transport. Following the example of other scalars in MPAS-A (Skamarock and Gassmann, 2011), we model $CO_2$ advection as:

$$(\nabla \cdot \tilde{\rho} q_{co_2} \mathbf{V})_\zeta = \Big[ \frac{\partial(\tilde{\rho} u q_{co_2})}{\partial x} + \frac{\partial(\tilde{\rho} v q_{co_2})}{\partial y} \Big]_\zeta + \frac{\partial(\tilde{\rho} w q_{co_2})}{\partial \zeta} \tag{2}$$

The first item on the right hand side enclosed in the square bracket is the $CO_2$ horizontal flux divergence, and second item is the vertical flux divergence. The horizontal flux divergence is transformed via the divergence theorem into an integral of flux over each control volume, which is modeled as:

$$\Big[ \frac{\partial(\tilde{\rho} u q_{co_2})}{\partial x} + \frac{\partial(\tilde{\rho} v q_{co_2})}{\partial y} \Big]_\zeta = \frac{1}{A_i} \sum_e^{n_e} l_e F_e(\mathbf{V}_H, \tilde{\rho} q_{co_2}) \tag{3}$$

where $e$ indexes the edges of a cell and $n_e$ represents the number of edges the cell has, $l_e$ is the length of an edge, $A_i$ is the cell's areal size, $F_e(\mathbf{v}_H, \tilde{\rho} q_{co_2})$ is the instantaneous horizontal $CO_2$ flux that crosses the cell edge $e$, and $\mathbf{V}_H = (u, v)$ is the horizontal wind vector. The details of MPAS-A instantaneous horizontal flux calculation can found in Skamarock and Gassmann (2011). The vertical $CO_2$ flux divergence in Eq. 2 is calculated using finite difference

$$\frac{\partial(\tilde{\rho} w q_{co_2})}{\partial \zeta} = \frac{1}{\Delta \zeta} \Big[ F(w, \tilde{\rho} q_{co_2})_{k+\frac{1}{2}} - F(w, \tilde{\rho} q_{co_2})_{k-\frac{1}{2}} \Big] \tag{4}$$

where $F(w, \tilde{\rho} q_{co_2})$ is the vertical $CO_2$ flux that crosses a cell's vertical face, and $k$ indexes the vertical coordinate.

## 2.2  $CO_2$ vertical mixing

Like in WRF (Skamarock et al., 2008), a PBL parameterization in MPAS-A treats the vertical mixing of momentum and scalars not only in the boundary layer (BL) but in the entire atmospheric column. The YSU scheme (Hong et al., 2006) is one of the PBL schemes available in MPAS-A 6.3. The present YSU scheme treats vertical mixing of momentum, potential temperature,

and water species, but not atmospheric tracers. We modified the scheme to treat $CO_2$ vertical mixing.

In the YSU scheme, after the top of BL is determined, the vertical mixing of momentum, potential temperature, and water vapor are treated separately: above BL, local K-profile approach (Louis, 1979) is used for vertical diffusion of momentum and scalars (Noh et al., 2003; Hong et al., 2006). Within BL, an entrainment flux at the inversion layer is included for momentum and scalars diffusion. In addition, a countergradient mixing term is included for the diffusion of momentum and potential temperature to account for the convective-driven mixing ($\gamma_c$ of Eq. 4 in Hong et al. (2006)), but this term is not used for water vapor.

Following the treatment of water vapor, we parameterize $CO_2$ vertical mixing in BL as

$$\frac{\partial q_{co_2}}{\partial t} = \frac{\partial}{\partial z}\left[K_h(\frac{\partial q_{co_2}}{\partial z}) - \overline{(w'q'_{co_2})_h}(\frac{z}{h})^3\right] \tag{5}$$

where $z$ is the vertical distance to surface, $h$ is BL top height, $K_h$ is vertical eddy diffusivity. Note that this formulation does not include a countergradient mixing term following the treatment of water vapor in the original YSU scheme (Hong et al., 2006). The second term in the square bracket of Eq. 5 represents the contribution from $CO_2$ entrainment flux at the inversion layer, which is parameterized as:

$$\overline{(w'q'_{co_2})_h} = w_e \Delta q_{co_2}|_h \tag{6}$$

where $\Delta q_{co_2}|_h$ is the $CO_2$ mixing ratio difference across the inversion layer, and $w_e$ is the entrainment rate at the inversion layer calculated by Eq. A11 of Hong et al. (2006). Above BL top, vertical mixing of $CO_2$ is parameterized as:

$$\frac{\partial q_{co_2}}{\partial t} = \frac{\partial}{\partial z}\left[K_h(\frac{\partial q_{co_2}}{\partial z})\right] \tag{7}$$

We use the same value for $CO_2$ vertical diffusivity as water vapor. The details of $K_h$ calculation can be found in the appendix of Hong et al. (2006), and its value is limited between 0.01 and 1000 $m^2s^{-1}$ to prevent too weak or strong vertical mixing. The term $\partial q_{co_2}/\partial t$ from Eqs. 5 is coupled with dry air density before being applied to the continuity equation (Eq. 1).

## 2.3 $CO_2$ convective transport

For convective transport, we modified the Kain-Fritsch scheme (hereafter KF) (Kain, 2004) to include the $CO_2$ treatment. KF is a mass-flux convection scheme which rearranges mass in an air column using convective updrafts, downdrafts, and environmental mass fluxes. Both the updraft and downdraft entrain from and detrain to the environment, thus altering the vertical profile of an air column's thermodynamic properties. We added the $CO_2$ convective transport as:

$$\frac{\partial q_{co_2}}{\partial t} = \frac{(M_u + M_d)}{\rho A}\frac{\partial q_{co_2}}{\partial z} + \frac{M_{ud}}{M}(q^u_{co_2} - q_{co_2}) + \frac{M_{dd}}{M}(q^d_{co_2} - q_{co_2}) \tag{8}$$

where $q_{co_2}$, $q_{co_2}^u$, and $q_{co_2}^d$ are the $CO_2$ mixing ratio in the environment, updraft, and downdraft respectively, $M_u$ and $M_d$ are the updraft and downdraft mass respectively, $\rho$ is the environment air density, $A$ is the horizontal area of a cell, $M = \rho A \delta z$ is the mass of environmental air in a grid box, and $M_{ud}$ and $M_{dd}$ are the detrainment from the updraft and downdraft respectively.

In the KF scheme, the updraft and downdraft mass and the rates for the entrainment and detrainment are determined by a steady-state plume model and a convective available potential energy (CAPE) closure assumption: 90% of the existing CAPE should be removed by the convection parameterization (Kain and Fritsch, 1990; Fritsch and Chappell, 1980; Kain, 2004). Because the calculation of the updraft and downdraft mass fluxes is related to a cell's horizontal area, the KF scheme may behave differently at different areas of MPAS-A's variable-resolution grid. The updraft source layers are determined by a search from the model's lowest vertical level for a group of consecutive layers that is buoyant and at least 50 hPa deep (Kain, 2004). The initial value of $CO_2$ mixing ratio in the updraft is modeled as a pressure weighted average of the source layers:

$$q_{co_2}^u = \sum_k \frac{\delta q_{co_2,k}\,\delta p_k}{\delta p_k} \tag{9}$$

where $\delta p_k$ is layer's pressure depth, and $q_{co_2,k}$ is the layer's $CO_2$ mixing ratio. $CO_2$ mixing ratio of the updraft is modified by the entrainment of the environmental air through its ascent from its starting level to the cloud top.

$$q_{co_2}^u = \frac{q_{co_2}^u M_u + q_{co_2} M_{ue}}{M_u + M_{ue}} \tag{10}$$

where $M_{ue}$ is the updraft entrainment. The initial $CO_2$ mixing ratio of a downdraft ($q_{co2}^d$) is the same as that of the environment ($q_{co2}$) at the downdraft starting level and it is modified by entrainment through the downdraft descent:

$$q_{co_2}^d = \frac{q_{co_2}^d M_d + q_{co_2} M_{de}}{M_d + M_{de}} \tag{11}$$

where $M_{de}$ is the downdraft entrainment.

## 3 Model evaluation

In this section we evaluate the newly developed MPAS-A $CO_2$ transport model by comparing its simulation results with observations and other models. After describing the simulation configuration (Sect. 3.1), we assess the model's global mass conservation property (Sect. 3.2). Then we evaluate the model's $CO_2$ transport accuracy at the global scale using hourly near-surface $CO_2$ observations from 50 in situ stations and column-averaged $CO_2$ dry air mole fraction ($XCO_2$) measurements from 18 Total Carbon Column Observing Network (TCCON) stations (Sect. 3.3). Finally, we evaluate MPAS-A at the regional scale using high-resolution airborne measurements from ACT campaign over the eastern United States (Sect. 3.4). MPAS-A $CO_2$ transport are also compared with three established CTMs: NOAA CT2019 (Jacobson et al., 2020), ECMWF IFS (Agusti-Panareda et al., 2019), and WRF-Chem (Skamarock et al., 2008). In the following model evaluation, we use root mean square

error (RMSE), bias ($\mu$), and random error (STDE) as the model accuracy metrics:

$$\text{RMSE} = \sqrt{\frac{1}{N} \sum_{i=1}^{N} (m_i - o_i)^2} \tag{12}$$

$$\mu = \frac{1}{N} \sum_{i=1}^{N} (m_i - o_i) \tag{13}$$

$$\text{STDE} = \sqrt{\frac{1}{N} \sum_{i=1}^{N} (m_i - o_i - \mu)^2} \tag{14}$$

where $o_i$ and $m_i$ represent the observed and modeled values respectively.

For model-data intercomparison, MPAS-A model data need to be interpolated to the observation space. Following Patra
et al. (2008), the model is sampled in the horizontal by taking the nearest cell overland. MPAS-A uses a height-based terrrain-
following vertical coordinate (Skamarock et al., 2012). At a given cell, the height of the $k^{\text{th}}$ vertical layer boundary is denoted
as $z_k^h$. The height of the layer center is $z_k = 0.5 \times (z_k^h + z_{k+1}^h)$. In MPAS-A, horizontal wind fields are defined at the vertical
layer boundaries and $CO_2$ fields are defined at layer centers. For horizontal wind fields validation using radiosonde data (Sect.
3.3.1), the column profile of air pressure and horizontal wind fields defined at layer boundaries are used to interpolate to the
15 measurements' pressure levels. To compare with near-surface $CO_2$ observations from in-situ stations (Sect. 3.3.3) and aircraft
observations (Sect. 3.4), model $CO_2$ defined at layer centers are interpolated to the measurement heights. Vertical interpolation
and integration for the comparison with TCCON $XCO_2$ are described in Sect. 3.3.4. MPAS-A simulation outputs are saved at
1-hour intervals. For comparison with radiosonde observations and near-surface $CO_2$ observations, no temporal interpolations
are applied: observations are paired with the closest hourly MPAS-A output. For comparison with aircraft observations, the
20 hourly model outputs that bracket an observation's time stamp are used for the temporal interpolation.

## 3.1 Simulation experiment configuration

For all subsequent simulations, MPAS-A uses a 60-15km variable-resolution global mesh. Figure 1 shows the cell size (in km$^2$)
of the simulation domain, where the highest resolution (15 km) over North America has cell size smaller than 250 km$^2$ which
gradually increases to about 3,600 km$^2$ for the rest of the global domain. On the vertical direction, there are 55 levels spanning
from surface to 30 km above the mean sea level. Model time step is 90 seconds in accordance with the highest (15km) hori-
zontal resolution. For physical parameterizations, in addition to the modified YSU PBL (Hong et al., 2004) and Kain-Fristch
cumulus schemes (Kain, 2004) described in Section 2, we use RRTMG for longwave and shortwave radiation (Iacono et al.,

2008), Noah land scheme (Chen and Dudhia, 2001), Monin-Obukhov surface layer scheme, and WRF single-moment 6-class microphysics scheme (Hong and Lim, 2006). The third-order accuracy advection is used for all scalars and the $CO_2$ tracer. A summary of the physics parameterizations used in the simulations is given in Table 1.

Initial meteorological fields are generated from the ERA-Interim reanalysis (Dee et al., 2011). To keep model meteorological fields close to the reanalysis, MPAS-A meteorological fields are re-initialized using the analysis at 00:00 UTC each day throughout a simulation period. $CO_2$ mixing ratio is kept unchanged during the meteorology re-initializations, thus a free-running simulation. This configuration is the same as that used by Agusti-Panareda et al. (2014, 2019) in their IFS global $CO_2$ simulations. The first $CO_2$ initial condition for a simulation is from CT2019 $3° \times 2°$ posterior dry mole fraction product and
surface $CO_2$ fluxes are prepared by interpolating the CT2019 3-hourly $1° \times 1°$ posterior flux product (Jacobson et al., 2020). The four components of CT2019 fluxes (biosphere, ocean, fossil fuel, and fire) are interpolated to MPAS-A model grid and ingested at 3-hour intervals throughout a simulation.

### 3.2    $CO_2$ mass conservation

For CTM, it is very important to maintain the global $CO_2$ mass conservation (Agusti-Panareda et al., 2017; Polavarapu et al., 2016). Because meteorological re-initializations introduce changes in dry air mass, they impact MPAS-A's global $CO_2$ mass conservation. We first examine MPAS-A's inherent mass conservation property through a simulation without the meteorological re-initializations in Sect. 3.2.1. Then we examine and treat the impacts of the meteorological re-initializations in Sect. 3.2.2.

#### 3.2.1    Mass conservation without meteorology re-initialization

To examine MPAS-A's mass conservation property, we conducted a MPAS-A simulation that lasts from January 1 to December 31 2014. The simulation is initialized with the CT2019 $CO_2$ mole fraction and is driven with 3-hourly CT2019 surface $CO_2$ fluxes. Meteorological re-initializations are not applied during the simulation and the model outputs are saved using double-precision. MPAS-A's global dry air mass ($M_{air}$) is then calculated at 00:00 UTC each day through the one-year simulation using Eq. 15,

$$25 \quad \mathrm{M_{air}} = \sum_{k}^{L} \left( \sum_{i}^{N} A_i \, h_{i,k} \, \rho_{i,k} \right) \tag{15}$$

where subscript $i$ indexes the horizontal cell, subscript $k$ indexes the vertical level, $A_i$ is cell size, $h_{i,k}$ is cell height, and $\rho_{i,k}$ is dry air density ($\mathrm{kg\,m^{-3}}$). After the model's global dry air mass is calculated at 00:00 UTC each day of the simulation period, its variation is quantified as a ratio $E_{air}^t = (M_{air}^t - M_{air}^0)/M_{air}^0$, where $M_{air}^0$ and $M_{air}^t$ are the model's global dry air mass at the simulation start (00:00 UTC January 1 2014) and the current time step respectively. The top panel of Fig. 2 shows $E_{air}^t$ at
00:00 UTC of each day through the one-year simulation period. The figure shows that the maximal magnitude of $E_{air}^t$ is less

than $4 \times 10^{-12}$ during the one-year simulation. In comparison, the total dry air mass of ECMWF IFS increases about $0.01\%$ of its initial value in a 10-day forecast (Diamantakis and Flemming, 2014). Similarly, the Environment and Climate Change Canada (ECCC) Global Environmental Multiscale (GEM-MACH-GHG) model loses about $0.01\%$ of its initial total dry air mass in a 10-day forecast (Polavarapu et al., 2016). MPAS-A has a significantly lower global dry air mass variation than the two global models because its explicit grid point advection scheme conserves mass (Skamarock and Gassmann, 2011) while the semi-Lagrangian advection scheme used by IFS and GEM-MACH-GHG does not conserves mass (Williamson, 1990). Thus, no mass fixer (Diamantakis and Flemming, 2014; Polavarapu et al., 2016) is used in MPAS-A.

MPAS-A's global $CO_2$ mass ($M_{co_2}$) is calculated using Eq. 16,

$$M_{co_2} = \sum_{k}^{L}(\sum_{i}^{N} A_i\, h_{i,k}\, \rho_{i,k}\, q_{i,k}) \tag{16}$$

where $q_{i,k}$ is the $CO_2$ dry air mixing ratio (kg/kg) and the rest of the terms are the same as in Eq. 15. To assess the global $CO_2$ mass conservation, $M_{co_2}$ calculated using Eq. 16 is adjusted for the $CO_2$ mass introduced through the ingestion of the 3-hourly surface $CO_2$ fluxes. For a 3-hour period, total $CO_2$ mass introduced through the surface $CO_2$ fluxes is $\sum_{i}^{N} A_i F_i \Delta t$, where $F_i$ is the combined biosphere, ocean, fossil fuel, and fire $CO_2$ fluxes ($kg\,m^{-2}\,s^{-1}$) at a surface cell, $A_i$ is the cell's areal size, $N$ is number of surface cell, and $\Delta t$=3 hours. After the adjustment, the variation of global mass of $CO_2$ is quantified as a ratio, $E_{co_2}^t = (M_{co_2}^t - M_{co_2}^0)/M_{co_2}^0$, where $M_{co_2}^0$ and $M_{co_2}^t$ are the global $CO_2$ mass at the initial and current time step respectively. $E_{co_2}^t$ at 00:00 UTC of each day of the simulation period is shown in the lower panel of Fig. 2. The figure shows that the maximal magnitude of $E_{co_2}^t$ is about $10^{-5}$. This is much higher compared to $E_{air}^t$ and it is due to the strong gradients caused by surface $CO_2$ flux which challenge the model's numerical scheme.

## 3.2.2 CO$_2$ mass conservation during meteorology re-initialization

When meteorological re-initialization is applied during a simulation, the values of dry air density in MPAS-A are replaced by values from the initialization files generated from the ERA-Interim reanalysis. In most cases, this will cause dry air density change which in turn will introduce $CO_2$ mass change if $CO_2$ dry air mixing ratios are kept unchanged during the re-initialization. To assess this possible change in global $CO_2$ mass, we conducted another one-year long MPAS-A simulation identical to that used in Section 3.2.1 except that meteorological re-initialization is applied at 24-hour intervals during the simulation. The variation of global $CO_2$ mass caused by a meteorological re-initialization is quantified as a ratio $E = (M'_{co_2} - M_{co_2})/M_{co_2}$, where $M_{co_2}$ and $M'_{co_2}$ are the global $CO_2$ mass before and after a meteorological re-initialization. The top panel of Fig. 3 shows the value of $E$ at each meteorological re-initialization. The figure indicates that a meteorological re-initialization could cause a change of more than $0.01\%$ of the global $CO_2$ mass.

To keep the $CO_2$ mass conservation after a meteorological re-initialization, we adjust MPAS-A's $CO_2$ fields by a spatially uniform scaling factor: $q'_{i,k} = r \times q_{i,k}$, where $q_{i,k}$ and $q'_{i,k}$ are the $CO_2$ dry air mixing ratio, before and after the adjustment, respectively. The scaling factor $r$ is calculated as,

$$r = \frac{\sum_k^L (\sum_i^N A_i \, h_{i,k} \, \rho_{i,k} \, q_{i,k})}{\sum_k^L (\sum_i^N A_i \, h_{i,k} \, \rho'_{i,k} \, q_{i,k})} \tag{17}$$

where the notations are the same as in Eq. 16 except that $\rho'_{i,k}$ is the dry air density after a meteorology re-initialization and $\rho_{i,k}$ is the value before the re-initialization. To test the effectiveness of this scaling method, the one-year MPAS-A simulation with meteorological re-initialization was conducted again but this time with the $CO_2$ dry air mixing ratio adjustment applied after each meteorological re-initialization. The resulting variation in total $CO_2$ mass is plotted in the lower panel of Fig 3. The figure shows the maximal magnitude of the variation caused by a meteorological re-initialization has been reduced from $\sim 10^{-4}$ to $\sim 10^{-6}$ of the global $CO_2$ mass. Note the different scales in the y-axis used in the top and bottom panels of Fig. 3.

An alternative approach to restore mass conservation is to scale $CO_2$ mixing ratio at each grid box individually by

$$q'_{i,k} = \frac{\rho_{i,k}}{\rho'_{i,k}} \times q_{i,k} \tag{18}$$

where notation is the same as Eq. 17. This scaling approach can maintain global $CO_2$ mass conservation as allowed by machine precision but it will introduce artificial spatial variations in $CO_2$ mixing ratio. In the simulations in the following sections, we chose to use the first scaling approach to avoid the artificial $CO_2$ mixing ratio variation by accepting the small changes in global $CO_2$ mass.

### 3.3 Model evaluation at global scale

In this section, we evaluate the MPAS-A CO2 transport at the global scale. For the model evaluation, MPAS-A was initialized at 00:00 UTC July 1 2013 and ran till December 31 2014. The model configuration for this simulation is as described in Sect. 3.1. With the first six-month as model spin-up, we use the one-year simulation of 2014 for the model evaluation. First MPAS-A simulated horizontal wind fields are evaluated using radiosonde measurements from 457 stations. Then the model's $CO_2$ fields are compared with CT2019, near-surface $CO_2$ measurements from 50 stations, and $XCO_2$ retrievals from 18 TCCON stations.

### 3.3.1 Evaluation of horizontal wind fields

Accurate meteorological fields are critical for an accurate $CO_2$ transport simulation. Before evaluating the simulated $CO_2$, we first evaluate the MPAS-A simulated horizontal wind fields considering their importance in $CO_2$ advection. We compare MPAS-A simulated horizontal wind fields at 12:00 and 00:00 UTC each day of the simulation period with raidosonde observations from 457 stations located around the globe at four pressure levels:1000, 850, 500, 850, and 200 hPa. Note that because of

the 24-hourly meteorological re-initialization, the 00:00 and 12:00 UTC simulation results are 12-hour and 24-hour forecasts respectively. The locations of the 457 radiosonde stations are shown in Fig. S1 of the supplement.

To compare with the similar validation results reported in Agusti-Panareda et al. (2019), the horizontal wind fields evaluation results for January and July of 2014 are listed in Table 2. The table shows that while the mean difference in wind direction decreases with altitude, the mean RMSE vector wind generally increases with altitude, which agree with the IFS validation results (Agusti-Panareda et al., 2019). At 1000 hPa level, MPAS-A has a slightly lower accuracy than IFS during the same time period. For instance, MPAS-A's mean RMSE vector wind at 1000 hPa is 3.83 m/s for January 2014, and IFS results range from 3.2 m/s to 3.75 m/s for its 9 km and 80 km horizontal resolution simulations. For July 2014, the mean RMSE vector wind at 1000 hPa is 3.47 m/s from MPAS-A and 3.0 m/s to 3.6 m/s for the IFS 9 km and 80 km simulations. At upper level, MPAS-A has a slightly higher accuracy than IFS: at 500 hPa, MPAS-A mean RMSE vector wind is 3.72 m/s and 3.39 m/s for January and July of 2014 respectively, while IFS results in 4.0-4.1 m/s and 3.5-3.6 m/s for the same time period.

An important finding of Agusti-Panareda et al. (2019) is that higher horizontal resolution generally leads to higher meteorological and $CO_2$ simulation accuracy. To examine the influence of horizontal resolution on MPAS-A's meteorological simulation accuracy, we conducted an additional set of simulation using the identical configuration except that it uses a global 60 km uniform-resolution grid instead of the 60-15 km variable-resolution grid (Fig. 1). Out of the 475 radiosonde stations, 131 are located at 15 km cells in the 60-15 km variable-resolution simulation. These 131 radiosonde stations are all located at 60 km cells in the 60 km uniform-resolution simulation. In Table 3, we calculated and compared horizontal wind accuracy at these 131 radiosonde stations between the 60 km uniform-resolution simulation (labeled as 60 km) and the 60-15 km variable-resolution simulation (labeled as 15 km). The table shows that the horizontal wind fields at these 131 stations are simulated with considerably higher accuracy on the 15 km grid than its 60 km grid counterpart. For instance at 1000 hPa, the mean RMSE wind vector for January 2014 is 3.46 m/s and 3.98 m/s at the 15 km and 60 km grids respectively. The values are 3.10 m/s and 3.64 m/s for July 2014. Table 3 also shows that the difference in the mean RMSE wind vector between the 15 km and 60 km grids is larger near the surface at 850 and 1000 hPa than in the middle and upper troposphere (500 and 200 hPa), which is consistent with the findings of Agusti-Panareda et al. (2019). For both January and July at the four pressure levels, the mean RMSE wind vector at the 131 radiosonde stations at MPAS-A's 15 km grid is either similar to or slightly lower than the mean RMSE wind vector of the around 400 stations from the IFS 9 km resolution simulation (Agusti-Panareda et al., 2019).

### 3.3.2 Comparison of $CO_2$ fields with CarbonTracker

Having established that the horizontal wind fields simulated by MPAS-A are sufficiently accurate, the $CO_2$ fields can be evaluated. Here we directly compare the simulated $XCO_2$ by MPAS-A and CT2019 at the grid scale. CT2019 (Jacobson et al., 2020) is an operational carbon data-assimilation system which uses Transport Model 5 (TM5) (Krol et al., 2005) for atmospheric transport. TM5 is an offline global CTM which includes $CO_2$ advection, deep and shallow convection, and vertical diffusion in both PBL and FT (Krol et al., 2005). In producing CT2019 $CO_2$ mole fraction (Jacobson et al., 2020), TM5 simulation ran

over a $3° \times 2°$ global domain.

First, $XCO_2$ are calculated at the native grid for MPAS-A (60-15km) and CT2019 ($3° \times 2°$). $XCO_2$ at a given model cell is calculated as the pressure weighted $CO_2$ dry air mixing ratio.

$$5 \quad XCO_2 = (\sum_{k=1}^{N} p_k q_k^{co_2})/(\sum_{k=1}^{N} p_k) \tag{19}$$

where $p_k$ is modeled air pressure at layer $k$ corrected for water vapor, $q_k^{co_2}$ is $CO_2$ dry air mole fraction at the same level. $N$ is the number of vertical levels in a model. Then, $XCO_2$ from MPAS-A and CT2019 are regridded from their respective grids an identical $1 \times 1°$ grid for a direct comparison. Figure 4 shows the comparison of $XCO_2$ from MPAS-A (top) and CT2019 (middle) and their difference (bottom) for July 1 and December 1 2014 at 00:00 UTC. The figure shows that $XCO_2$ from 10 MPAS-A and CT2019 are generally consistent at the large scales, but differences exist at small spatial scales. For instance, the difference in horizontal resolution between MPAS-A and CT2019 can be clearly observed in $XCO_2$ in July over both northeast and southern China. In December, MPAS-A has higher $XCO_2$ than CT2019 within the Arctic Circle and southern China. Overall the differences between MPAS-A and CT2019 are evident. The magnitude of differences are mostly within 3 ppm, which is similar to the magnitude reported in Polavarapu et al. (2016) for the GEM-MACH-GHG model. Because both models 15 used the same surface $CO_2$ fluxes, the difference in the simulated CO2 fields is only caused by the different model transport: spatial resolution, dynamics, and physical parameterizations. The differences between MPAS-A and CT2019 are expected due to the differences in the two models' horizontal resolution, dynamics, and physical parameterizations. Because no CTM can be expected to have perfect transport, the acceptability of transport is generally judged through comparisons of model simulation with measurements.

### 3.3.3 Comparison with near-surface CO$_2$ measurements

This section compares MPAS-A simulated $CO_2$ with hourly measurements from 50 stations that were used for the IFS model evaluation in Agusti-Panareda et al. (2019). The information of the 50 stations, including location, elevation, intake height, reference, and type is listed Table 4. Like in Agusti-Panareda et al. (2019), only the highest intake level is used at towers that have 25 multiple intake heights. When multiple observations within an hour are available (such as those with 30-min or shorter time interval), they are averaged to yield a single hourly value. For a given station this results in 744 ($24 \times 31$) hourly measurements per month at the maximum.

The MPAS-A hourly $CO_2$ statistics, including RMSE, STDE, and bias at the 50 stations are listed in Tables S1 and S2 of the 30 supplement for January and July of 2014 respectively. For comparison, Tables S1 and S2 also include the statistics from the IFS 9 km and 80 km resolution simulations (Agusti-Panareda et al., 2019) at the same sites for the same time periods. Table S1

shows that RMSE of the MPAS-A simulated hourly $CO_2$ ranges from 0.17 ppm at the SPO station to 16.65 ppm at the KAS station. In comparison, the IFS simulations also resulted in a much lower RMSE at the SPO than KAS, the latter of which has a RMSE of 4.44 ppm from the 9 km resolution simulation and 10.71 ppm from the 80 km simulation.

5     The comparison of RMSE and STDE from MPAS-A and IFS are show in Figs 5 and 6 for January and July of 2014, respectively. Table 5 uses paired $t$ test to provide a quantitative summary of the hourly $CO_2$ RMSE between MPAS-A and the IFS 9 km and 80 km simulations. The table shows that for January 2014, the mean RMSE at the 50 stations is 4.20 ppm from MPAS-A, which is higher than IFS 9 km simulation (3.12 ppm, $p = 0.01$) and similar to the IFS 80 km simulation (4.94 ppm, $p = 0.25$). For July 2014, the mean RMSE at the 50 stations is 8.09 ppm from MPAS-A, which is similar to IFS 9 km 10   simulation (8.04, $p = 0.95$) and lower than the IFS 80 km simulation (11.77 ppm, $p = 0.04$). The above comparisons indicate that the 60-15 km MPAS-A simulation has a level of accuracy between the IFS 9 km and 80 km simulations.

    Agusti-Panareda et al. (2019) found that atmospheric $CO_2$ transport is generally better represented at higher horizontal resolutions, and mountain stations display the largest improvement at higher resolution as they directly benefit from the more 15   realistic orography. There are 12 mountain stations of the 50 stations used for the model validation. Table 6 lists the 12 mountain stations in two groups: the first group includes the six mountain stations located at the 15 km cells of the MPAS-A's 60-15 km variable-resolution grid, and the second group includes the other six stations that are located at the 60 km cells of the grid. The table lists the hourly $CO_2$ RMSE for each of the 12 stations from MPAS-A and IFS 9 km and 80 km simulations are listed for January and July 2014. The table shows that at each of the six mountain stations located at 15 km cells, MPAS-A has lower 20   hourly $CO_2$ RMSE than the IFS 9 km simulation for July 2014. For January 2014, MPAS-A has lower RMSE than IFS 9 km simulation at five out the six stations (the exception is NWR). In comparison, at the six mountain stations located at its 60 km cells, MPAS-A has higher hourly $CO_2$ RMSE than IFS 9 km simulation for both January and July of 2014 with the exception of JFJ for July 2014.

### 3.3.4   Comparison with TCCON XCO$_2$ measurements

After the comparison with the near-surface $CO_2$ in the last section, we evaluate MPAS-A $CO_2$ fields using $XCO_2$ measurements from 18 TCCON sites listed in Table 7. To compare with TCCON retrieved $XCO_2$, smoothed MPAS-A $XCO_2$ is calculated following Wunch et al. (2010):

$$X_{CO_2}^{model} = c_a + \mathbf{h}^T \mathbf{a}^T (\mathbf{x}_m - \mathbf{x}_a) \tag{20}$$

30   where $X_{CO_2}^{model}$ is the smoothed MPAS-A $XCO_2$, $c_a$ is the a priori total column, $\mathbf{a}^T$ is TCCON column averaging kernel, $\mathbf{h}^T$ is a dry-pressure weighting function, $\mathbf{x}_m$ is MPAS-A $CO_2$ dry mole fraction profile, $\mathbf{x}_a$ is the a priori $CO_2$ dry mole fraction profile. The column profile of $CO_2$, air pressure, and water vapor mixing ratio extracted from MPAS-A hourly output are in-

terpolated to the same vertical grid as $\mathbf{x}_a$, and dry-pressure weighting function $\mathbf{h}^T$ is calculated following O'Dell et al. (2012) and Eq. A7 of Agusti-Panareda et al. (2014).

At a given TCCON site, averaged hourly XCO$_2$ (denoted as $X_{CO_2}^{TCCON}$) is calculated as the mean value of all valid XCO$_2$ retrievals within the hour. $X_{CO_2}^{TCCON}$ are then matched with the calculated hourly XCO2 from MPAS-A (denoted as $X_{CO_2}^{model}$). The comparisons of $X_{CO_2}^{model}$ and $X_{CO_2}^{TCCON}$ at the 18 TCCON sites for the year of 2014 are shown in Fig. 7. The results indicate that the observed seasonal variation in TCCON XCO$_2$ are in general well represented by MPAS-A. The hourly average XCO$_2$ comparison between MPAS-A and TCCON are summarized in Table 8. In the table $N$ is the number of data pairs used for calculating the statistics, including RMSE, bias, and correlation coefficient $R$. The mean RMSE of the 18 sites is 1.35 ppm, which is comparable to the IFS simulations (1.02 to 1.25 ppm) Agusti-Panareda et al. (2019). We then calculated the average daily XCO$_2$ as the mean value of all the hourly XCO$_2$ within a given day. The statistics of comparison of daily XCO$_2$ between MPAS-A and TCCON are also included in Table 8. In the table $N$ is the number of average daily XCO$_2$ used for calculating the statistics. Compared to their hourly counterparts, the average daily XCO$_2$ have both lower RMSEs and higher correlation coefficients. The mean value of the average daily XCO$_2$ RMSE of the 18 TCCON sites is 1.23 ppm, which is comparable to IFS simulations (0.97 to 1.25 ppm ) reported in Agusti-Panareda et al. (2019).

### 3.4 Model evaluation at regional scale

In this section, we present an evaluation of the MPAS-A CO$_2$ simulation accuracy using an extensive high resolution CO$_2$ observation data acquired through the ACT aircraft campaigns. ACT is a National Aeronautics and Space Administration (NASA) Earth Venture Suborbital 2 (EVS-2) mission, and its goal is to improve atmospheric inversion estimates of CO$_2$ and CH$_4$ through extensive airborne measurements over the eastern United Stated during multiple seasons (Davis et al., 2018a). Through four campaign seasons from Summer 2016 to Spring 2018 with two research aircraft (C130 and B200), the ACT project has collected an extensive dataset of highly resolved CO$_2$ measurements in both BL and FT. The duration of the ACT campaign seasons is given in Table 9. To use ACT airborne CO$_2$ measurements for model evaluation, we conducted a MPAS-A simulation lasts from January 1 2016 to May 31 2018. The first 6 months are for the model spin-up. The simulation uses the domain and configurations as described in Section 3.1, and model outputs are saved at 1-hour intervals.

First we compare MPAS-A simulated horizontal wind fields during the ACT campaign seasons using the same procedure described in Section 3.3.1. Table 10 lists the statistics of horizontal wind fields evaluation at the four ACT campaign seasons. The table indicates the same pattern as in 2014 (Table 2): mean RMSE vector wind increases with altitude and mean difference of wind direction decreases with altitude. The magnitude of the statistics of the four ACT campaign seasons are comparable to that of 2014 (Table 2).

Next we use the ACT campaign airborne measurements to evaluate MPAS-A CO$_2$ simulation regarding its overall accuracy and its performance measured by three model evaluation metrics proposed by Pal et al. (2020). To provide an objective refer-

ence, we also compare MPAS-A performance with two established $CO_2$ model systems: WRF-Chem (Skamarock et al., 2008) and CT2019 (Jacobson et al., 2020) using the same set of airborne measurements. WRF-Chem is an online CTM based on the regional model WRF (Grell et al., 2011; Skamarock et al., 2008). WRF-Chem simulations have been carried out at 27 km horizontal grid (Fig. S2) over North America as a part of the ACT campaign (Feng et al., 2020). The WRF-Chem simulations use

ERA5 reanalysis (Hersbach et al., 2020) for meteorological initial and lateral boundary conditions, CarbonTracker (Jacobson et al., 2020) posterior mole fraction for $CO_2$ initial and boundary conditions, and CarbonTracker posterior fluxes for surface $CO_2$ fluxes. The WRF-Chem simulations use meteorological nudging and 120-hour meteorological re-initialization to keep meteorological fields close to the reanalysis.

We use the ACT 5-second averaged $CO_2$ measurement dataset (Davis et al., 2018b), which has a horizontal resolution approximately 500 m given the average aircraft velocity. MPAS-A simulated $CO_2$ fields are sampled as described in the second paragraph of Section 3 to match the 5-second airborne data points. WRF-Chem simulated $CO_2$ fields are also interpolated to match the ACT 5-second data point using the same approach as MPAS-A. CT2019 $CO_2$ used for the evaluation is obtained from CarbonTracker ObsPack (v5.0) (Masarie et al., 2014), which is the CT2019 posterior mole fraction interpolated to the

ACT 5-second data points.

For each ACT flight day, $CO_2$ measurements from the two aircraft are combined if both are available, and their corresponding modeled $CO_2$ values from MPAS-A, WRF-Chem, and CT2019 are combined in the same way. With the four seasons combined, there are a total of 97 flight days (Pal and Davis, 2020), each one presented by an observation-model dataset con-

20 sisted of observed $CO_2$, modeled $CO_2$ from the three models, along with the time, latitude, longitude, and altitude of each observation data point. Using the ACT maneuver flag dataset Pal et al. (2020), we further divide each flight day's data into two groups: one for BL and another for FT. For each ACT campaign season, all the BL data-model pairs are combined for each of the three models for model comparison. Figure 8 shows the Taylor diagram of the model comparison in BL for the four campaign seasons. $N$ in the title of each figure is the number of model-data pairs used for plotting the diagram. Similarly the

25 model comparison in FT is summarized in the Taylor diagrams of Fig. 9. A comparison of Figs. 8 and 9 show that all three models have higher accuracy (lower RMSE) in FT than BL, which could be attributed to the larger error in the weather forecast in BL than FT associated with the accuracy of PBL height in the model simulation. Figure 8 shows that in BL, MPAS-A has higher RMSE and higher standard deviation than CT2019. MPAS-A has more accurate estimation of the observations' standard deviation than CT2019 in all but summer 2016. Compared with WRF-Chem, MPAS-A has lower RMSE and more accurate

estimation of the observations' standard deviation. Figure 9 shows that in FT, MPAS-A has higher RMSE than CT2019 in all four campaign seasons and but it has more accurate estimation of the observations' standard deviation than CT2019 in all but summer 2016 season. Compared to WRF-Chem, MPAS-A has lower RMSE and more accurate estimation of observations' standard deviation in all but summer 2016.

### 3.4.1 Model representation of $CO_2$ difference between warm and cold sectors

Through analyzing the ACT Summer 2016 campaign data, Pal et al. (2020) identified three consistent features in $CO_2$ mole fraction and proposed to use these features as transport model assessment metrics. The three features are the differences between the warm and cold sectors, the difference between BL and FT, and the $CO_2$ enhancement bands in the vicinity of frontal

5     boundaries. Here and in the next two sections, we evaluate how MPAS-A simulated $CO_2$ represents the three features.

    Using the ACT maneuver flag dataset (Pal et al., 2020), we identified flights that crossed a weather front and their associated warm and cold sectors. The $CO_2$ mole fraction statistics for the warm and cold sectors are calculated from the aircraft measurements and the modeled $CO_2$ by MPAS-A, WRF-Chem, and CT2019, respectively. The results are shown in Fig. 10, which

10    summarizes the statistics of $CO_2$ mole fraction differences between the warm and cold sectors measured by 15 front-crossing flights: 10 from the summer 2016 season and 5 from the winter 2017 season. The figure confirms that the warm sector has higher average $CO_2$ mole fraction in BL than the cold sector during summer 2016 as reported by Pal et al. (2020). The figure also shows that the average $CO_2$ mole fraction in the warm sectors are lower than than the colder sectors in winter 2017, opposite of summer 2016.

    Table 11 lists the mean $CO_2$ of the warm sector, cold sectors, and their difference as calculated from the ACT measurements, MPAS-A, WRF-Chem, and CT2019. The table shows that the MPAS-A simulations are similar to WRF-Chem, and both tend to have larger $CO_2$ differences between the warm and cold sectors than CT2019. For instance, the 2016-08-08 case where the observed mean $CO_2$ difference between warm and cold sector is 26.9 ppm, MPAS-A and WRF simulations resulted in 36.9

20    ppm and 21.2 ppm respectively, while CT2019 resulted in a 15.3 ppm difference. The above evaluation indicates that MPAS-A $CO_2$ model is capable of well representing the observed $CO_2$ difference between the warm and cold sectors, and its accuracy in this respect is comparable to WRF-Chem and CT2019.

### 3.4.2 Model representation of $CO_2$ vertical difference

25    The second feature identified by Pal et al. (2020) is the vertical difference of $CO_2$ mole fraction between BL and FT. During ACT campaign season, two research aircraft (B200 and C130) took many vertical profile measurements during take off, landing, spiral up and down, and inline ascend and descend maneuvers (Pal, 2019). These profile observations characterize the vertical variation of the atmospheric $CO_2$ mole fraction. From the vertical profile measurements taken during the summer 2016 season, Pal et al. (2020) calculated the mean $CO_2$ mole fraction in BL and FT, denoted as $[CO_2]_{BL}$ and $[CO_2]_{FT}$ respectively.

30    They further defined BL-to-FT $CO_2$ difference as $\Delta[CO_2] = [CO_2]_{BL} - [CO_2]_{FT}$. They found that $\Delta[CO_2]$ tend to be positive in the warm sector and negative in the cold sector. In this section, we evaluate how well MPAS-A represents the BL-to-FT $CO_2$ difference and compare its performance with WRF-Chem and CT2019.

Using the ACT maneuver flag dataset (Pal et al., 2020), we identified all vertical profiles taken during the four campaign seasons, from which we selected profiles that meet two criteria: (1) a vertical profile must include at least 20 5-second measurements in BL and 20 measurements in FT; and (2) a vertical profile must extend at least 2 km in the vertical direction. These two criteria are used to ensure that the resulting $[CO_2]_{BL}$ and $[CO_2]_{FT}$ are statistically representative. A total of 199 qualified vertical profiles are identified from the four campaign seasons, including 72 from the summer 2016 season, 27 from winter 2017, 40 from fall 2017, and 60 from spring 2018. For each of the vertical profiles, $\Delta[CO_2]$ is calculated for the aircraft $CO_2$ measurements, and the simulated $CO_2$ by MPAS-A, WRF-Chem, and CT2019. We compare $\Delta[CO_2]$ from the models with that from the observations to assess how each model represents the observed BL-to-FT $CO_2$ difference. Figure 11 shows the comparisons grouped by the campaign seasons. The figure indicates a clear distinction in $\Delta[CO_2]$ between the summer 2016 and the other three seasons: There are a substantial number of both positive and negative $\Delta[CO_2]$ in the summer 2016 season, but the vast majority of cases in the rest of the three campaign seasons have positive $\Delta[CO_2]$. The positive BL-to-FT $CO_2$ differences from the winter 2017 season measurements could be at least partially attributable to the lack of $CO_2$ draw-down during the non-growing season. In comparison, the fall 2017 and spring 2018 seasons have more mixed results probably because of their partial overlap with the growing season. For the summer 2016 season, vertical profiles with negative $\Delta[CO_2]$ (lower mean $CO_2$ in BL than FT) suggest photosynthesis during the growing season, but those with positive $\Delta[CO_2]$ values are probably caused by the interaction between photosynthesis and frontal passage (Pal et al., 2020).

To compare the three models' accuracy in representing the BL-TO-FT $CO_2$ difference, we calculated the mean absolute error (MAE) for each model at each season, where $AE = |\Delta[CO_2]_{model} - \Delta[CO_2]_{obs}|$ (the absolute difference in $\Delta[CO_2]$ between a model and the ACT observations).

$$MAE = \frac{1}{N}\sum_{i=1}^{N} AE_i \tag{21}$$

Table 12 summarize the MAE of the three models for each season. The table shows that MPAS-A has smaller MAE than CT2019 in fall 2017 ($p = 0.04$) and a larger MAE in summer 2016 ($p = 0.06$). The differences between the two models in the other two seasons are not significant ($p \geq 0.23$). Compared with WRF-Chem, MPAS-A has smaller MAEs in winter 2017 ($p = 0.09$) and spring 2018 ($p = 0.01$) while differences in the other two seasons are not significant ($p \geq 0.11$). In summary, the above model evaluation and comparison demonstrate that MPAS-A $CO_2$ transport model is capable of representing the aircraft observed $CO_2$ difference between BL and FT at least as accurately as WRF-Chem and CT2019.

### 3.4.3 Model representation of $CO_2$ enhancement at frontal boundaries

The third feature identified by Pal et al. (2020) in the summer 2016 aircraft measurements is the bands of enhanced $CO_2$ close to frontal boundaries in BL. They found these $CO_2$ enhancement bands are typically about 100 km wide and speculated that it would require a 20-km horizontal resolution model to effectively represent the feature. In this section, we identify

the frontal boundary $CO_2$ enhancements in the four campaign seasons and examine how well they are represented by MPAS-A.

Using the same approach as Pal et al. (2020), a total of 48 front-crossing constant-altitude flight segments are identified from the four seasons (15 from Summer 2016, 5 from Winter 2017, 17 from Fall 2017, and 11 from Fall 2018). To evaluate how well

MPAS-A represents the frontal boundary $CO_2$ enhancements and compare its performance with WRF-Chem and CT2019, $CO_2$ mole fraction measured by the aircraft and simulated by the three models are plotted together for each of the identified front-crossing constant-altitude flight segment. Figure 12 includes 8 of the front-crossing flight segments and the full set is included in Fig. S3 of supplement. For each flight segment in Fig. 12, the pair of vertical dashed lines mark $CO_2$ enhancement observed by the aircraft along a frontal boundary. The warm and cold sectors associated with the frontal boundary in each flight

are labeled as warm and cold respectively. The figure indicates that frontal boundary $CO_2$ enhancements can be identified in most but not all of the cases. For instance, there is not clearly identifiable $CO_2$ enhancement in the B200 flights on 2018-04-23 (Fig. S3).

Figure 12 shows that MPAS-A has a varying degree of success in simulating the frontal boundary $CO_2$ enhancements: it

represents both the timing and the magnitude of the enhancements very well in some cases (2016-08-04 and 2017-10-18 by B200), but results in substantial errors in either the timing (2016-07-25 B200) or the magnitude (2017-03-10 C130) in other cases. The figure also shows that the MPAS-A simulated $CO_2$ is more similar to WRF-Chem than CT2019: CT2019 tends to substantially underestimate the magnitude of $CO_2$ enhancement while MPAS-A and WRF-Chem tend to overestimate.

Figure 13 shows the MPAS-A simulated equivalent potential temperature ($\theta_e$) and $CO_2$ mole fraction at 18:00 UTC August 4, 2016. The sharp boundary in $\theta_e$ indicates a surface cold front extending from southern Colorado northeastward to Wisconsin. Abrupt horizontal wind direction changes shown in Fig. S4 of the supplement also indicate the cold front and its southeastward movement. Meteorological measurements taken during the flight (not shown) also confirm the cold front passage. The B200 research aircraft crossed the cold front from southeast to northwest at about 400-500 meters above the ground between 17:15

UTC and 19:15 UTC, and its flight track and timing are marked on Fig. 13. The aircraft measurements show an approximately 20 ppm enhancement along the front boundary, which can be clearly identified in the MPAS-A simulated $CO_2$ mole fraction (lower panel of Fig. 13).

Figure 14 compares the three models in their representation of the frontal boundary $CO_2$ variation. The figure shows that

except for summer 2016, MPAS-A has similar level of RMSE as CT2019 and it has more accurate estimation of the observations' standard deviation. As horizontal resolution impacts a model's ability to represent small scale spatial variability (Agusti-Panareda et al., 2019), the coarser resolution of CT2019 ($1° \times 1°$ over North America) is likely the primary cause of its underestimation of the frontal boundary $CO_2$ variability. MPAS-A has lower RMSE than WRF-Chem in winter 2017 and spring 2018, and similar RMSE as WRF-Chem in the other two seasons. In all but summer 2016, MPAS-A has more an

accurate estimation of the observations' standard deviation than WRF-Chem.

## 4 Summary

We implemented the $CO_2$ atmospheric transport processes, including advection, vertical mixing, and convective transport, in the global variable-resolution model MPAS-A. After the model development details are presented, simulation experiments designed for model evaluation are described. Two sets of simulations over a 60-15 km variable-resolution global domain are conducted for model accuracy evaluation using an extensive aircraft measurements over the eastern United States and near-surface hourly measurements from surface and tower stations distributed across the globe. Meteorological initial conditions for these simulations are from the ERA-interim analysis (Dee et al., 2011), and $CO_2$ initial conditions and fluxes are from CT2019 posterior mole fraction and fluxes products (Jacobson et al., 2020). To keep model meteorological fields close to the analysis, meteorology re-initialization are applied at 24-hour interval throughout the simulation periods. Global $CO_2$ mass conservation property is assessed by a one-year continuous simulation without meteorology re-initialization and fluxes, and the results show that MPAS-A is capable of maintaining total dry air mass conservation to the limit of machine precision. During the one-year simulation period, the total $CO_2$ mass change is about $10^{-5}$ of its initial value. The larger variation of $CO_2$ mass than the dry air is due to the complex and strong spatial gradient caused by the surface $CO_2$ fluxes. Another one-year simulation with meteorology re-initialization indicates that changes in dry air density during the re-initialization causes changes in global total $CO_2$ mass, and a scaling method applied after each re-initialization is able to reduce the change from $\sim 10^{-4}$ to $\sim 10^{-6}$ of the global $CO_2$ mass.

The horizontal wind fields of the 60-15 km variable-resolution MPAS-A simulation are evaluated at four pressure levels at 457 radiosonde stations. Furthermore, a comparison with an additional 60 km uniform-resolution MPAS-A simulation shows that the accuracy of the horizontal wind fields is substantially higher at the 15 km cells. The accuracy of MPAS-A $CO_2$ transport is evaluated first at the global scale and then at the regional scale. At the global scale, MPAS-A simulation is evaluated using CT2019, near-surface hourly $CO_2$ measurements from 50 stations and $XCO_2$ measurements from 18 TCCON stations. The resulting statistics are compared with the ECMWF IFS 9km and 80 km resolution simulations over the same period conducted by Agusti-Panareda et al. (2019). The comparison indicates that RMSE of the MPAS-A simulation is similar to the 80 km IFS simulation, but larger than the 9 km IFS simulation.

At the regional scale, a MPAS-A simulation extending from January 1 2016 to June 1 2018 is evaluated using the extensive high-resolution aircraft measurements from four ACT campaign seasons. Compared with a 27 km resolution WRF-Chem simulation and CT2019 posterior $CO_2$ mole fraction, MPAS-A simulated $CO_2$ achieves a comparable level of accuracy (as measured by RMSE). Further evaluation using three metrics proposed by Pal et al. (2020) shows that MPAS-A simulation is capable of representing the observed $CO_2$ features as accurately as the WRF-Chem simulation and CT2019.

The model evaluations using the airborne and near-surface measurements, indicates that the newly developed MPAS-A $CO_2$ transport model is capable of achieving a comparable level of accuracy with the more established $CO_2$ modeling systems,

including the regional model system WRF-Chem, the operational assimilation system CT2019, and the lower resolution (80 km) simulation of the ECMWF IFS global $CO_2$ modeling system. Although further improvements are expected, the MPAS-A $CO_2$ transport model has the potential to contribute to improving our knowledge of atmospheric $CO_2$ transport and fluxes.

**Code and data availability**

Source code for MPAS-A $CO_2$ transport model v6.3 can be retrieved at https://doi.org/10.5281/zenodo.3976320. Source code for WRF-Chem v3.6 used in the manuscript can be obtained from NCAR website at http://www2.mmm.ucar.edu/wrf/users/download/get_source.html. Source code for IFS is only available subject to a licence agreement with ECMWF. ECMWF member-state weather services and their approved partners will get access granted. The IFS source code without modules for assimilation and chemistry can be obtained for educational and academic purposes as part

of the openIFS release (https:// software.ecmwf.int/wiki/display/OIFS/OpenIFS+Home). ACT-America in-situ Airborne $CO_2$ measurement data can be obtained at https://doi.org/10.3334/ORNLDAAC/1593. Surface and tower based $CO_2$ measurement data from ObsPack GLOBALVIEWplus v5.0 can be obtained from NOAA website https://esrl.noaa.gov/gmd/ccgg/obspack/data.php. TCCON data can be obtain from https://tccondata.org/. CarbonTrack $CO_2$ flux and posterior mixing ratio data can be obtained from NOAA website https://www.esrl.noaa.gov/gmd/ccgg/carbontracker/download.php.

**Author contributions**

TZ implemented the $CO_2$ transport processes in MPAS-A v6.3. SF conducted the WRF-Chem 27 km simulations. TZ, SF, KD, and SP designed model evaluation using ACT-America aircraft measurements. TZ and JM designed the model evaluation using continuous in-situ tower measurements. TZ, SF, KD, and SP analyzed the model representation of distinct $CO_2$ spatial features observed by the aircraft measurements. All authors contributed to writing and commenting on the paper.

**Competing interests**

The authors declare that they have no conflict of interest.

**Acknowledgment**

We thank the MPAS-A development team for making their code available for the public. We thank NOAA CarbonTracker team for providing the CT2019 flux and mole fraction data. We thank ECMWF for the ERA-interim analysis data; we thank

ObsPack data providers for the in-situ continuous $CO_2$ measurement data. We thank TCCON PIs for providing the dataset. FS and KD were supported by the Atmospheric Carbon and Transport-America (ACT) Earth Venture Suborbital 2 project funded by NASA's Earth Science Division (Grant NNX15AG76G to Penn State). SP was supported by NASA Grant (Number 80NSSC19K0730) and a Texas Tech University start up research grant. This work was supported in part through computational

resources and services provided by the Institute for Cyber-Enabled Research at Michigan State University. This is contribution 154 of the Central Michigan University Institute for Great Lakes Research. We acknowledge partial support for the publication fee by Central Michigan University FRCE fund. We the two anonymous reviewers for their thorough and constructive comments which helped to improve this paper.

## 5 Financial support

This research has been supported by NASA (grant nos. NNX15AG76G and 80NSSC19K0730).

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

**Figure 1.** Variable-resolution 60-15 km global domain for MPAS-A $CO_2$ simulations conducted for model evaluation using aircraft and near-surface $CO_2$ observations. The highest resolution (15 km) grid covering the most of the North America has cell size less than 250 km$^2$. The cell sizes (represented by color) gradually increase to about 3,600 km$^2$ for the rest of the global domain.

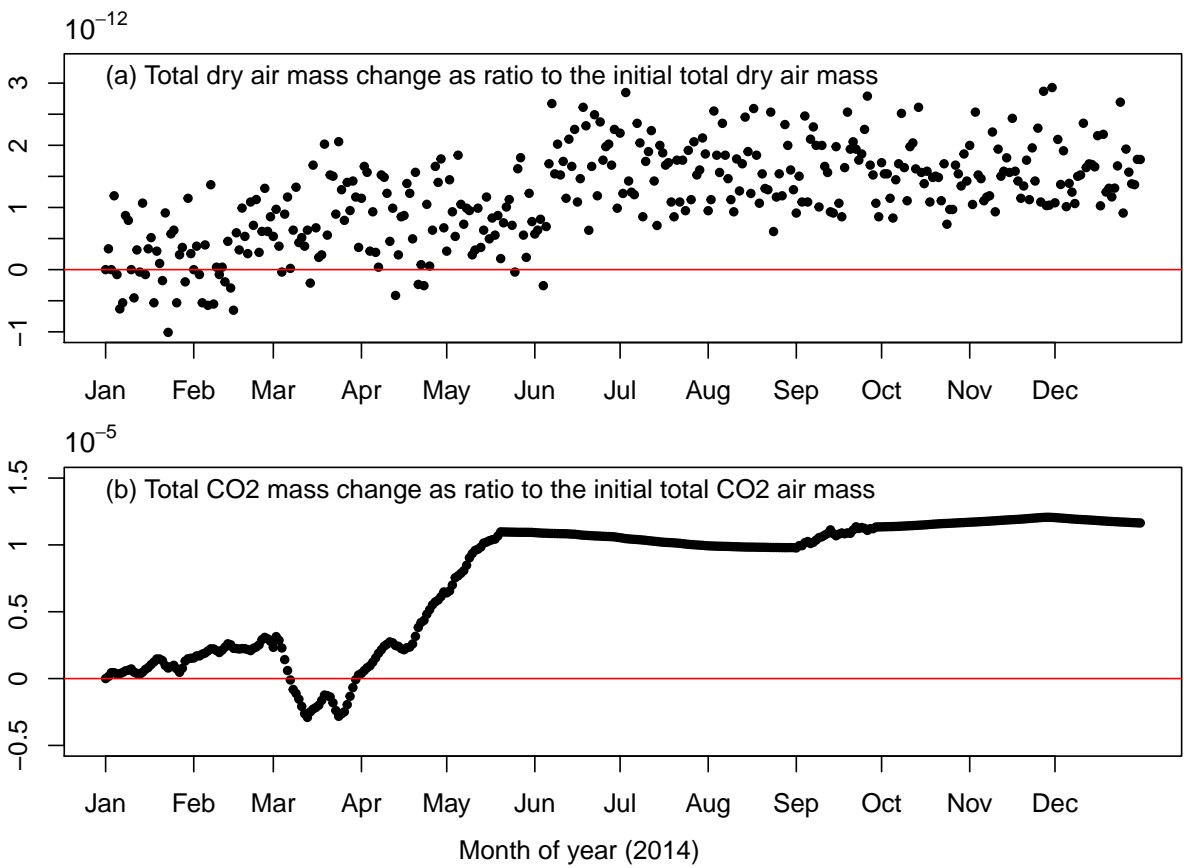

**Figure 2.** Variation of total dry air mass (a) and total $CO_2$ mass (b) as the ratio to their respective starting values during a 1-year continuous MPAS-A simulation without meteorological re-initialization. The X-axis represents the days of 2014, and the Y-axis the ratio of the total mass change to the starting values.

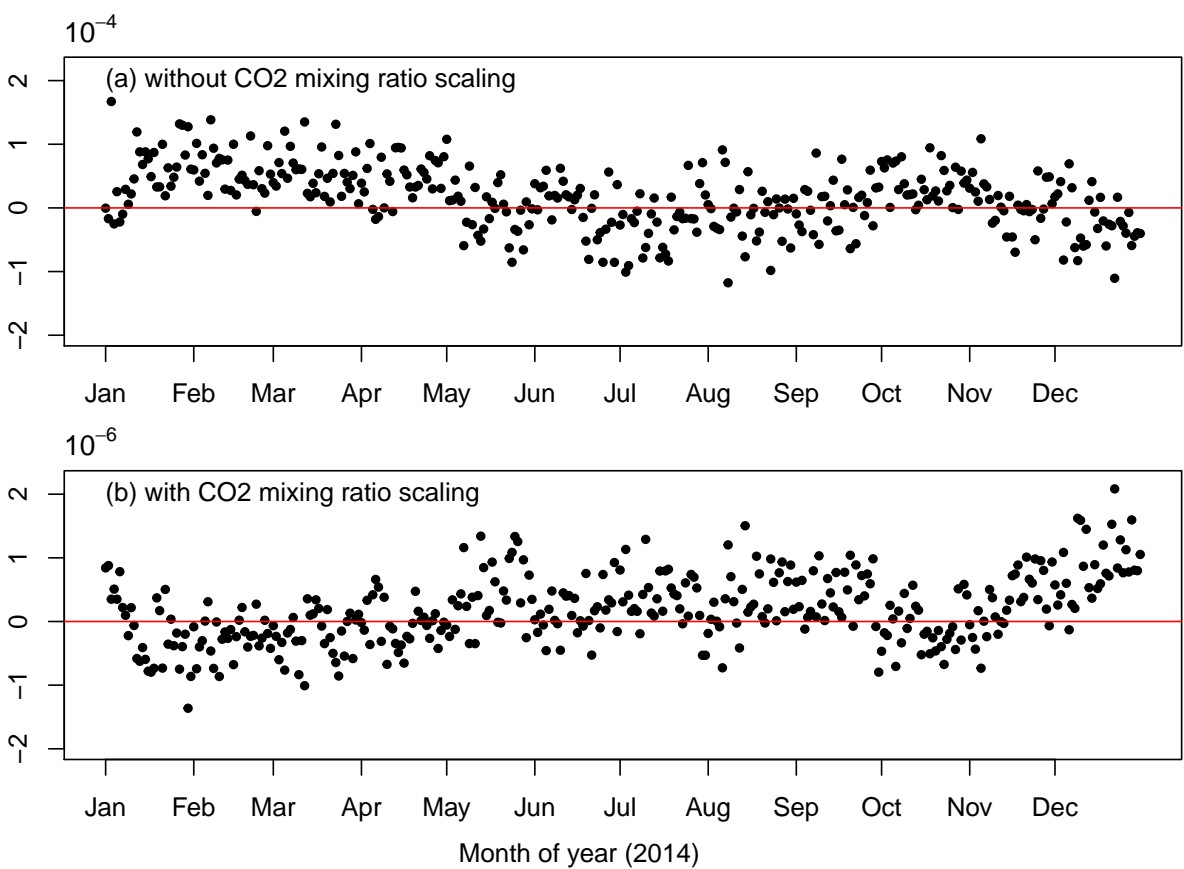

**Figure 3.** Variation of global $CO_2$ mass as the ratio to its value prior to a meteorological re-initialization during a 1-year MPAS-A simulation with meteorological re-initializations at 24-hour intervals. The top figure is from the simulation without applying $CO_2$ mixing ratio scaling as described in Sect. 3.2.2, and the bottom figure is from the simulation with the scaling. In each figure, X-axis represents the days of 2014, and Y-axis the ratio of global $CO_2$ mass variation to its value prior to a meteorological re-initialization.

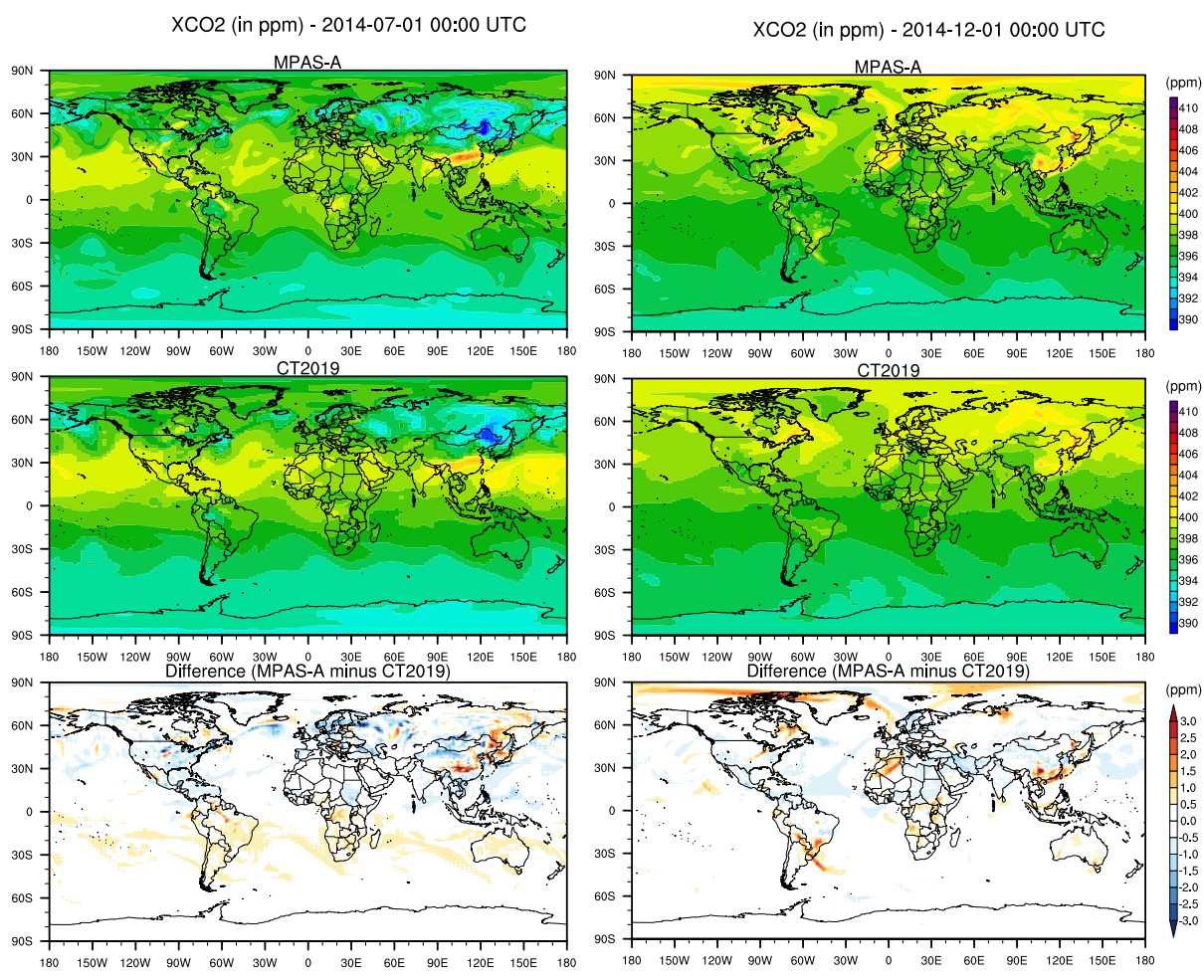

**Figure 4.** Simulated XCO$_2$ of MPAS-A, CT2019, and their difference at 2014-07-01 and 2014-12-01 00:00 UTC.

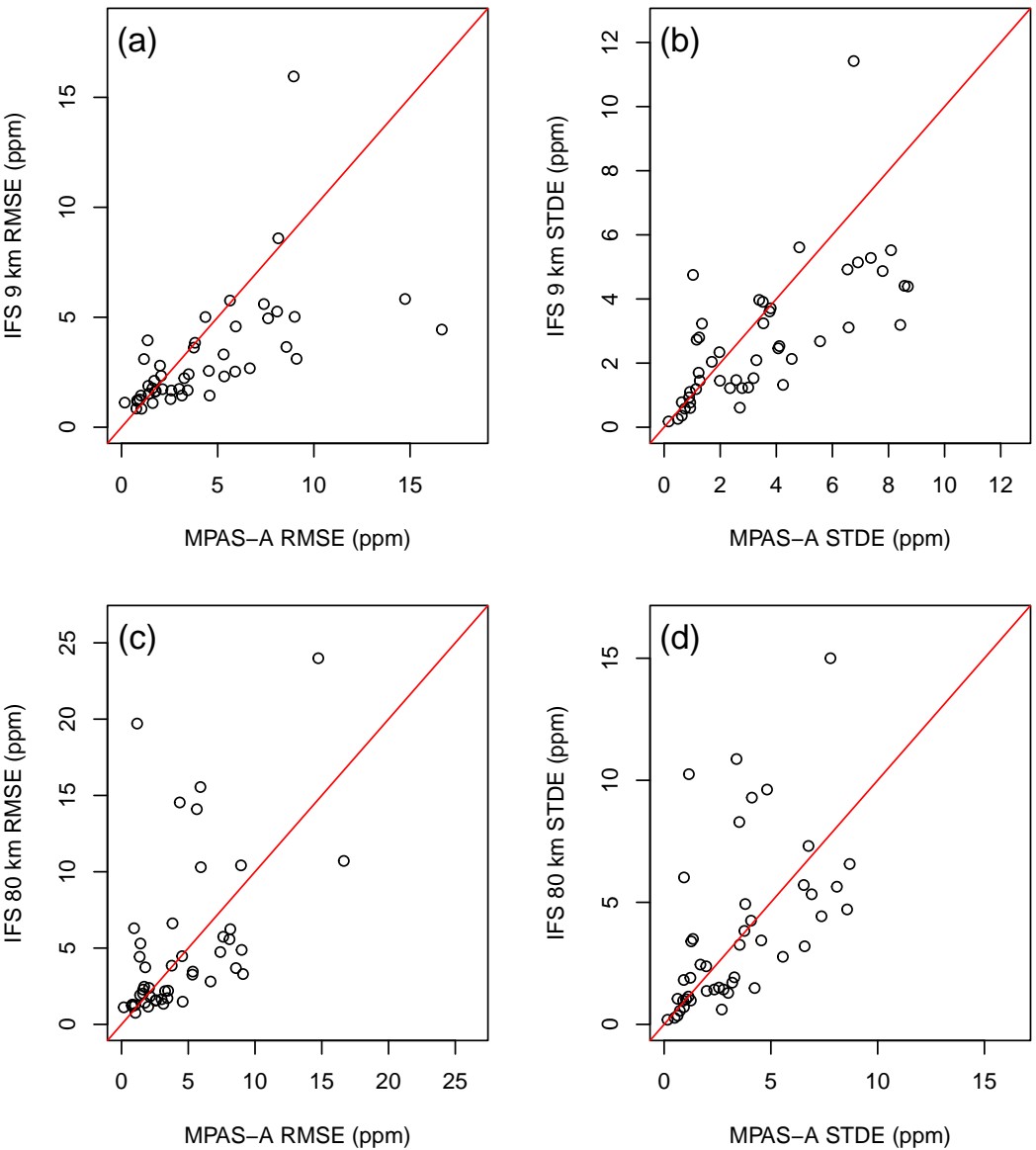

**Figure 5.** Comparison of model simulated hourly $CO_2$ accuracy (RMSE and STDE) between MPAS-A and IFS at 50 surface and tower stations. Each open circle in the figures represent a station. Comparison of MPAS-A with the IFS 9 km resolution simulations are in the top panel (a and b), and comparison with IFS 80 km resolutions simulations are in the top panel (c and d).

**July 2014**

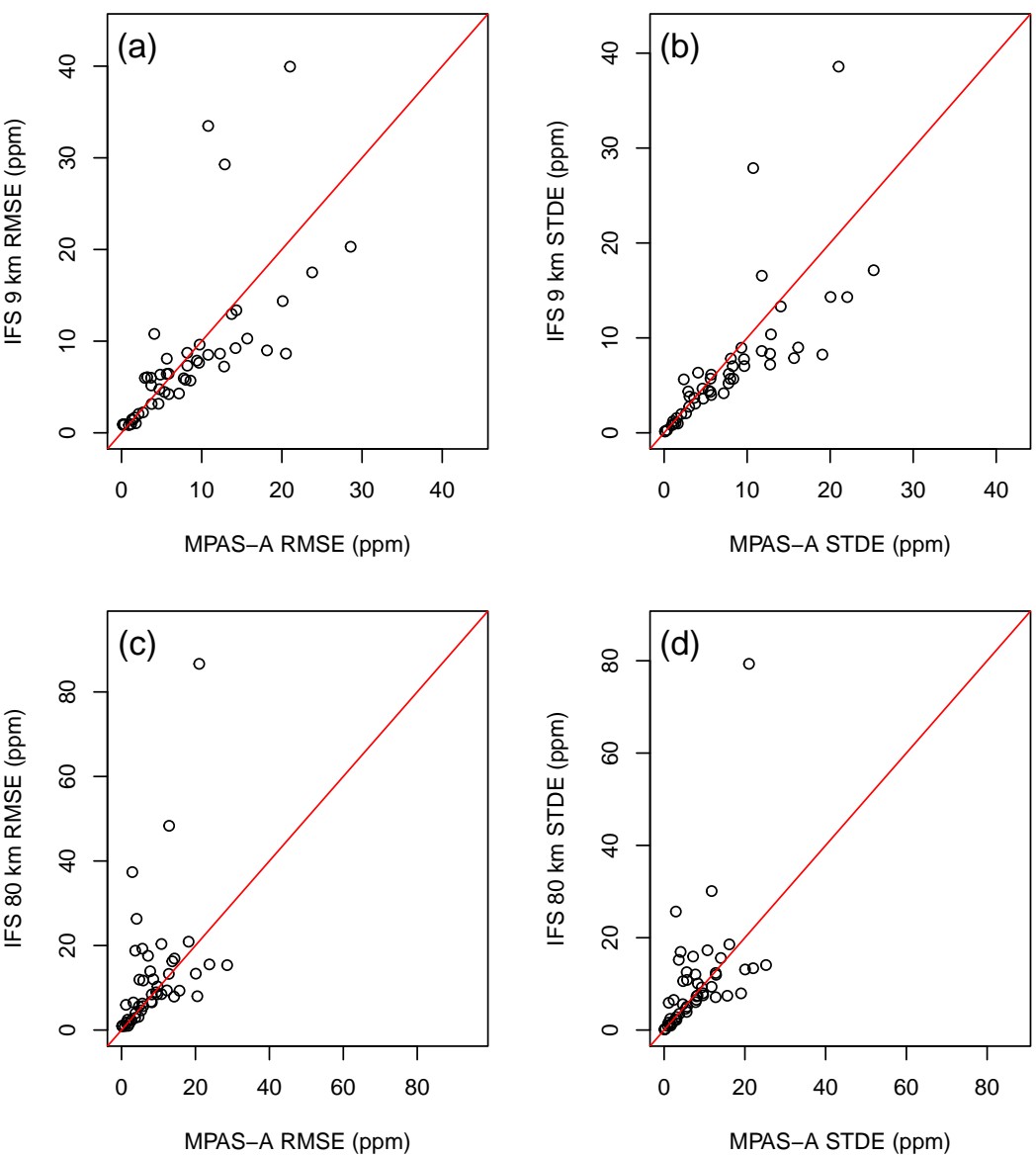

**Figure 6.** Same as Fig. 5, but July 2014.

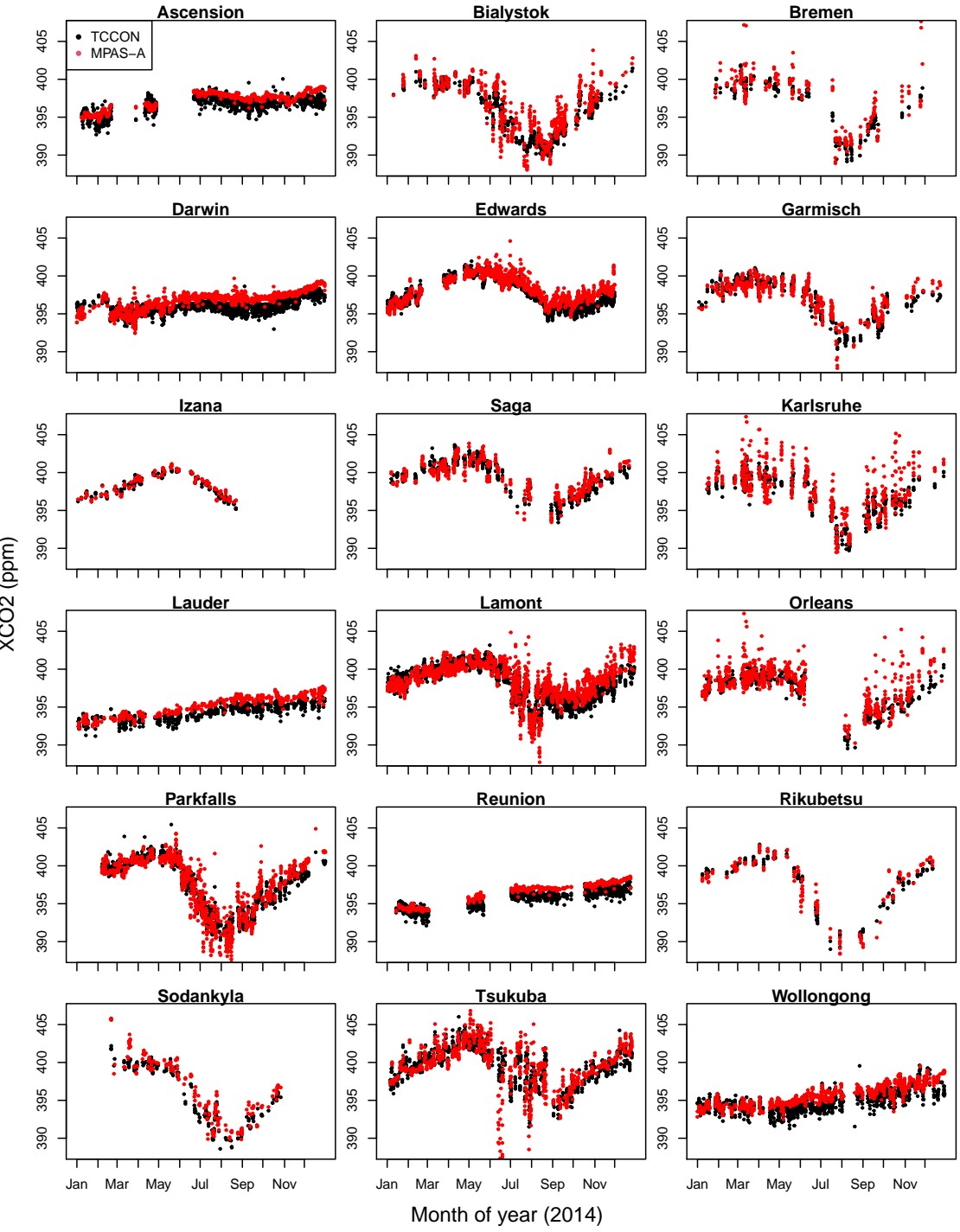

**Figure 7.** Simulated hourly XCO$_2$ of MPAS-A at 18 TCCON sites for the year of 2014.

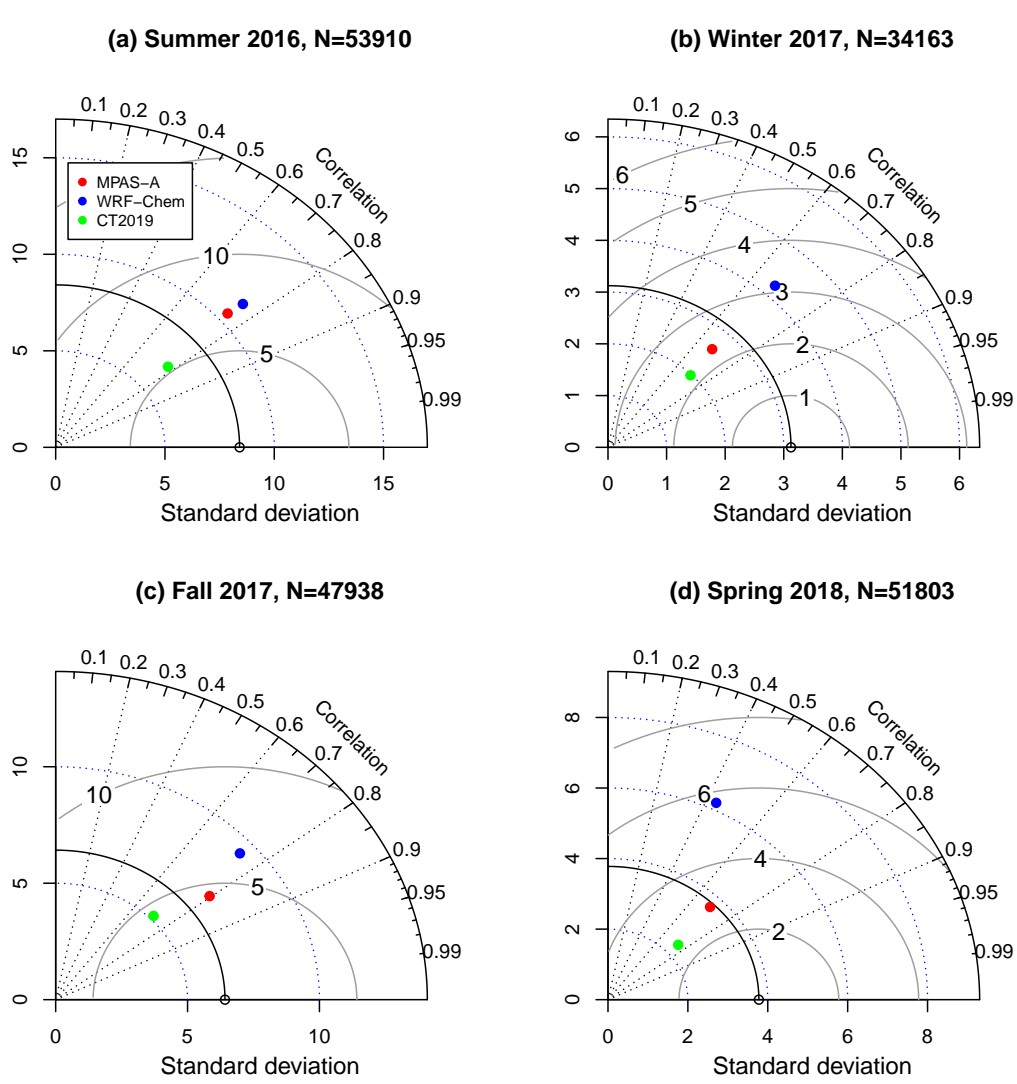

**Figure 8.** Taylor diagram for model evaluation using ACT airborne measurements in the boundary layer.

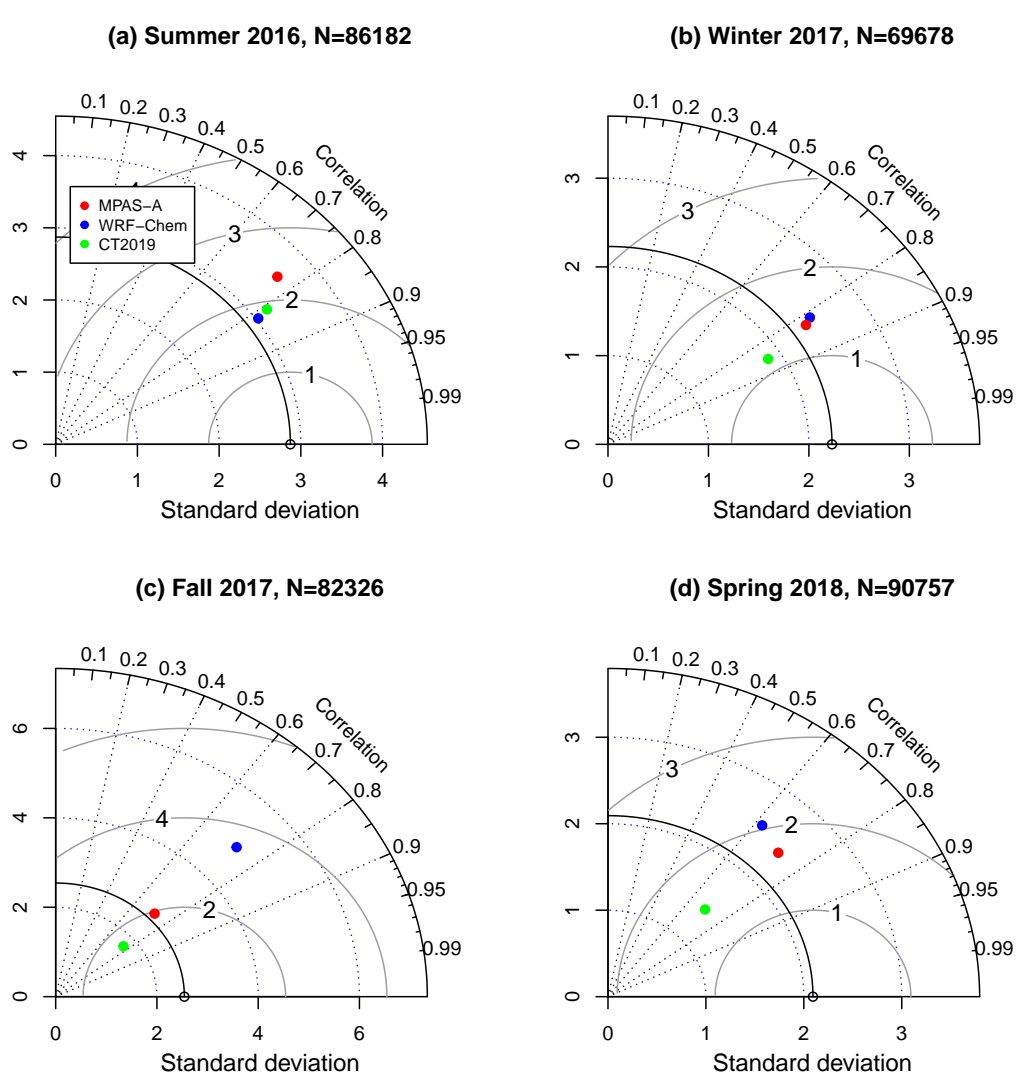

**Figure 9.** Taylor diagram for model evaluation using ACT airborne measurements in the free troposphere.

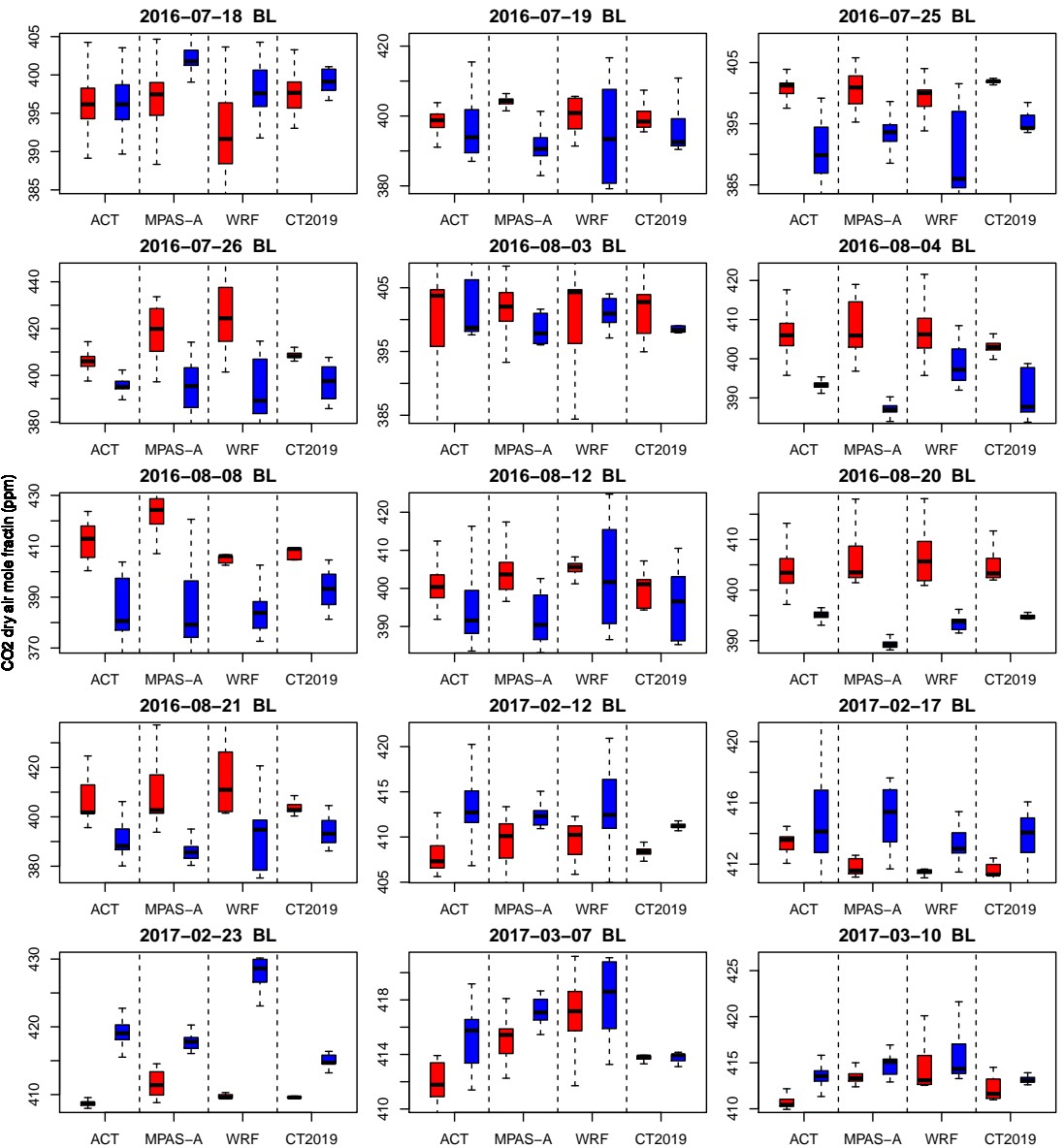

**Figure 10.** Box plots comparing mean boundary layer (BL) $CO_2$ mole fraction of the warm sector (red color) and cold sector (blue color) for 15 frontal crossing flights from summer 2016 and winter 2017 ACT campaign seasons. The flight date of each plot is labeled in its title. Data are combined when both aircraft (C130 and B200) took measurements for a given day. Each sub-figure is separated into four groups by the dotted lines: the first group is from ACT observations, the second is MPAS-A simulation, the third is WRF-Chem simulation, and the last is CT2019. In each boxplot, the bottom and top edge of the box represent the $1^{st}$ (Q1) and $3^{rd}$ (Q3) quartiles, the horizontal line represent the median, the ends of the whisker represents Q1-1.5×IQR and Q3+1.5×IQR respectively, where IQR=Q3-Q1.

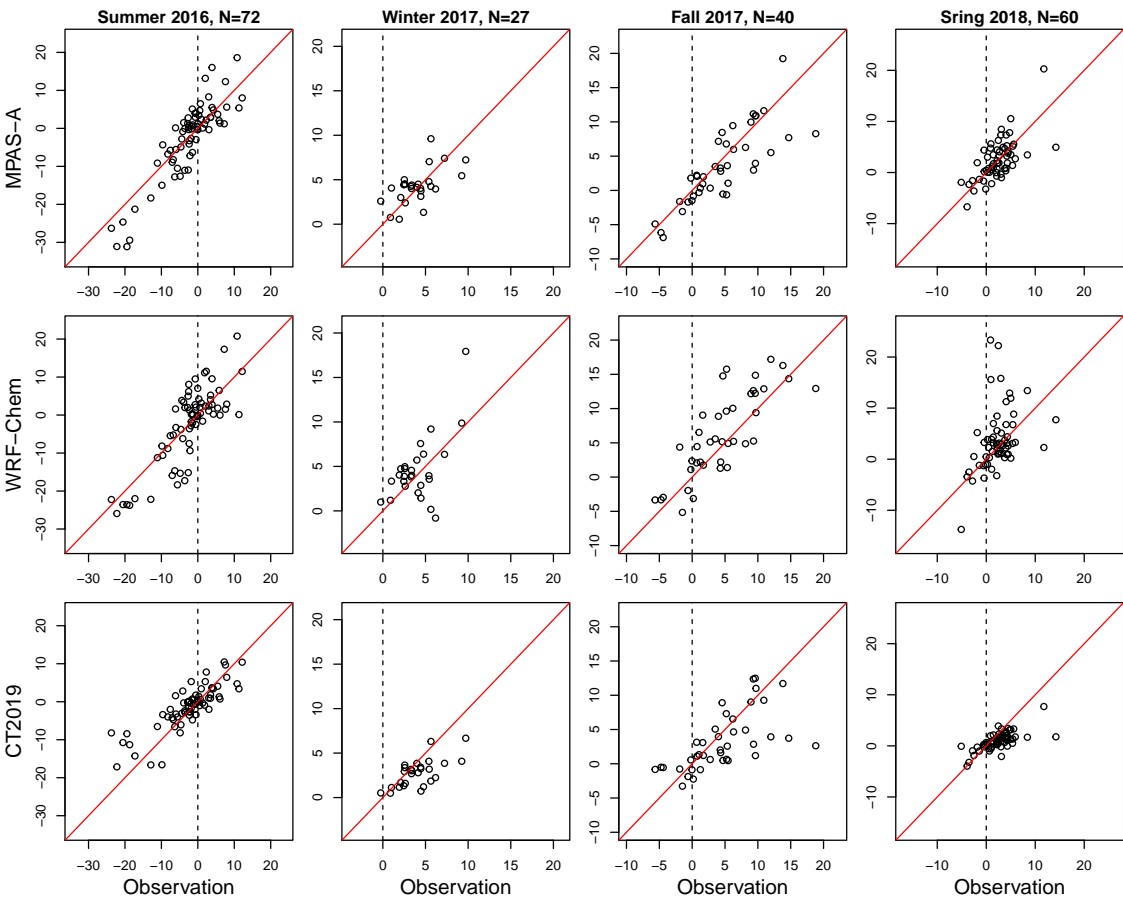

**Figure 11.** The difference of mean $CO_2$ mole fraction between boundary layer (BL) and free troposphere (FT) ($\Delta[CO_2] = [CO_2]_{BL} - [CO_2]_{FT}$) at vertical profiling flight legs. In each subplot, each open circle represents an individual vertical profiling flight leg, and its values on the X-axis and Y-axis represent its $\Delta[CO_2]$ value from the aircraft observations and the model simulation respectively. $\Delta[CO_2]$ from MPAS-A simulations are the first row, WRF-Chem the second row, and CT2019 the third. The four columns in the figure are for the four ACT campaign seasons. The number of vertical profiles in each season is labeled in the column title. The vertical dashed line marks where $\Delta[CO_2] = 0$ based on the aircraft measurements.

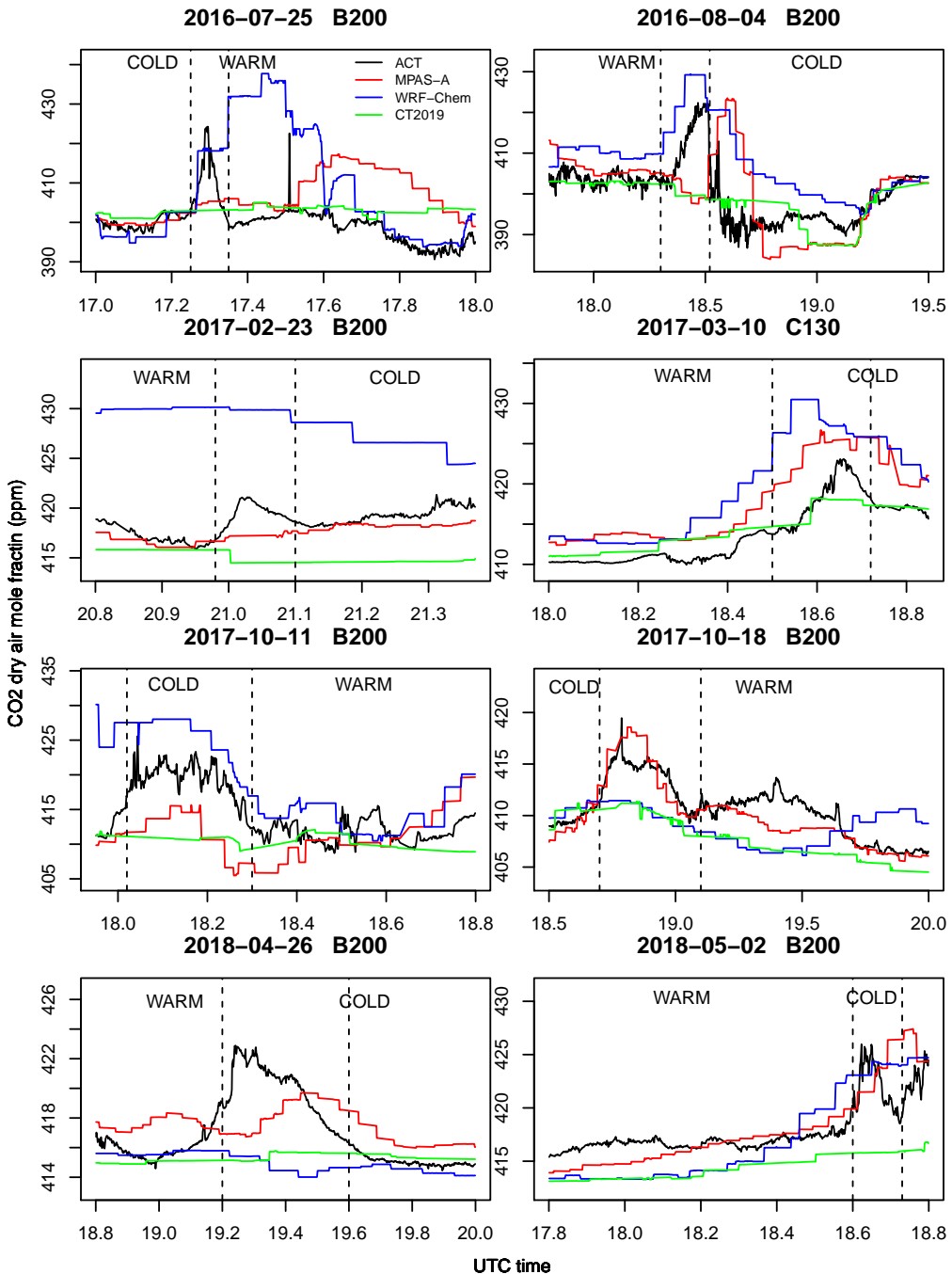

**Figure 12.** Comparison of $CO_2$ mole fraction in frontal-crossing constant altitude flight segments in BL between ACT aircraft measurements and model simulations. Flight date and aircraft type are labeled in title for each flight leg. X-axis is UTC time, and Y-axis is $CO_2$ mole fraction (ppm). Aircraft measurements are in black, MPAS-A in red, WRF-Chem in blue, and CT2019 in green. In each figure, the pair of vertical dashed lines mark $CO_2$ enhancement observed by the aircraft along a frontal boundary, and the warm and cold sectors associated with the frontal boundary are labeled as warm and cold, respectively.

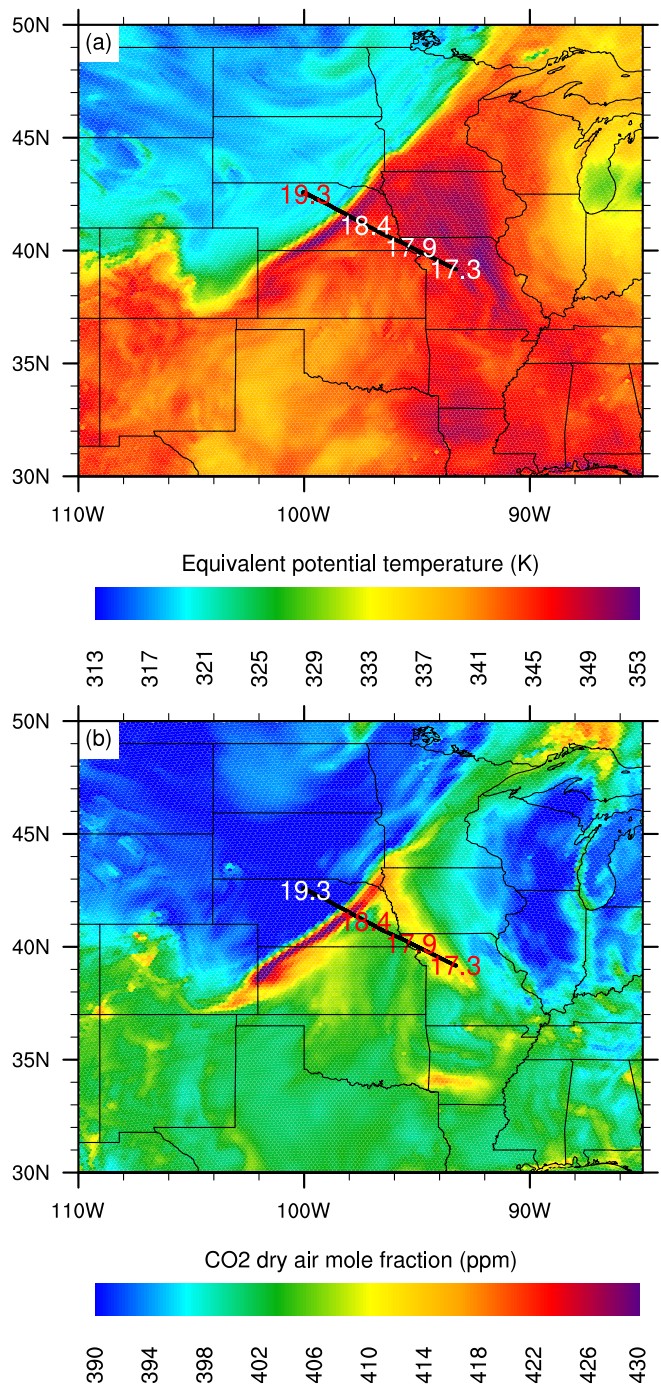

**Figure 13.** MPAS-A simulated equivalent potential temperature ($\theta_e$, top panel) and $CO_2$ mole fraction (bottom panel) at 18:00 UTC 4 August 2016. Both figures are plotted at MPAS-A $6^{th}$ vertical level, which is about 400 meters above the ground.

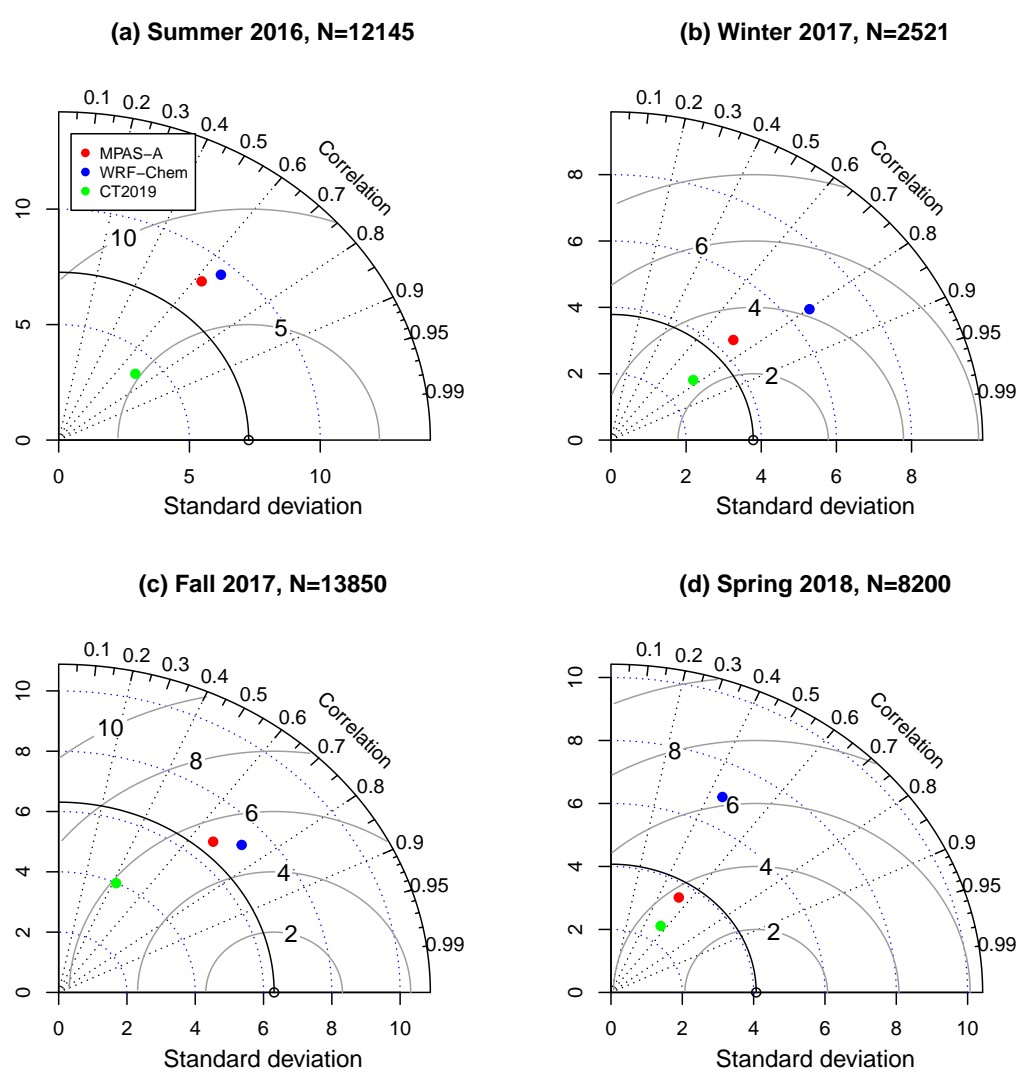

**Figure 14.** Taylor diagram for model evaluation using ACT airborne measurements from front-crossing flights. For each of the four ACT campaign seasons, observation-model data pairs from all front-crossing flights are combined. $N$ is the number of data pairs of used for creating the diagram.

**Table 1.** MPAS-A $CO_2$ transport model configuration.

| Parameterization | Option Used | References |
|---|---|---|
| Longwave | RRTMG LW | Iacono et al. (2008) |
| Shortwave | RRTMG SW | Iacono et al. (2008) |
| PBL | YSU | Hong et al. (2006) |
| Surface layer | Monin-Obukhov | |
| Land Surface Model | Noah | Chen and Dudhia (2001) |
| Cumulus | Kain-Fritsch | Kain (2004) |
| Microphysics | WRF Single Moment 6-class | Hong and Lim (2006) |

**Table 2.** Evaluation of MPAS-A simulated horizontal wind using NOAA IGRA radiosonde data. Radiosonde data from 457 stations over the globe are used for the evaluation. Wind speed and direction are compared at 00:00 UTC and 12:00 UTC at four pressure levels (1000, 850, 500, and 200 hPa) for each day of the MPAS-A simulation. The number of data samples (N) are smaller at 1000 hPa because some stations are located above that pressure level.

| | Pressure level | Mean RMSE vector wind (m/s) | Bias Wind speed (m/s) | Mean difference Wind direction (°) | Number of data |
|---|---|---|---|---|---|
| January 2014 | 1000 hPa | 3.83 | 0.89 | 31.00 | 8,856 |
| | 850 hPa | 4.04 | -0.45 | 23.08 | 22,369 |
| | 500 hPa | 3.72 | -0.50 | 12.98 | 23,222 |
| | 200 hPa | 4.38 | -0.58 | 8.36 | 22,795 |
| July 2014 | 1000 hPa | 3.47 | 0.27 | 32.94 | 7,504 |
| | 850 hPa | 3.56 | -0.51 | 27.12 | 22,832 |
| | 500 hPa | 3.39 | -0.59 | 17.89 | 23,745 |
| | 200 hPa | 4.19 | -0.55 | 12.19 | 23,455 |

**Table 3.** The statistics of MPAS-A simulated horizontal wind validated at radiosonde stations located in North America. In each cell, the first value is from the 60-15 km variable-resolution grid simulation (labeled as 15 km) and the second is from the 60 km uniform grid simulation (labeled as 60 km). Note that the number of data at the 850 hPa and 1000 hPa are different between the two simulations because of the differences in their grids' topography.

| | Pressure level | Mean RMSE vector wind (m/s) 15 km / 60 km | Bias Wind speed (m/s) 15 km / 60 km | Mean difference Wind direction (°) 15 km / 60 km | Number of data 15 km / 60 km |
|---|---|---|---|---|---|
| January 2014 | 1000 hPa | 3.46 / 3.98 | 0.84 / 1.02 | 30.27 / 32.24 | 2,427 / 1,845 |
| | 850 hPa | 3.42 / 4.08 | -0.32 / -0.59 | 21.58 / 24.70 | 6,227 / 6,187 |
| | 500 hPa | 3.37 / 3.68 | -0.41 / -0.54 | 13.10 / 13.89 | 6,659 / 6,659 |
| | 200 hPa | 4.01 / 4.20 | -0.44 / -0.53 | 8.54 / 8.78 | 6,536 / 6,536 |
| July 2014 | 1000 hPa | 3.10 / 3.64 | 0.13 / 0.39 | 32.91 / 33.93 | 2,778 / 2,027 |
| | 850 hPa | 3.34 / 4.08 | -0.35 / -0.60 | 26.08 / 27.66 | 6,395 / 6,321 |
| | 500 hPa | 3.26 / 3.68 | -0.54 / -0.65 | 17.46 / 18.25 | 6,861 / 6,861 |
| | 200 hPa | 3.76 / 4.06 | -0.41 / -0.40 | 10.99 / 11.70 | 6,808 / 6,808 |

**Table 4.** Continuous in-situ stations used for evaluating MPAS-A $CO_2$ simulation accuracy. NA denotes references that are not available.

| Station ID | Latitude | longitude | altitude (m a.m.s.l) | intake (m a.g.l) | Reference | Type |
|---|---|---|---|---|---|---|
| ALT | 82.45 N | 62.51 W | 200 | 10 | Worthy et al. (2003) | remote |
| BRW | 71.32 N | 156.61 W | 11 | 16 | Peterson et al. (1986) | coastal |
| CBY | 69.13 N | 105.06 W | 35 | 12 | NA | continental |
| INU | 68.32 N | 133.53 W | 113 | 10 | Worthy et al. (2003) | continental |
| PAL | 67.97 N | 24.12 E | 560 | 5 | Hatakka et al. (2003) | continental |
| BCK | 62.80 N | 116.05 W | 179 | 60 | NA | continental |
| CHL | 58.74 N | 94.07 W | 29 | 60 | Worthy et al. (2003) | coastal |
| LLB | 54.95 N | 112.45 W | 540 | 10 | Worthy et al. (2003) | continental |
| ETL | 54.35 N | 104.99 W | 492 | 105 | Worthy et al. (2003) | continental |
| MHD | 53.33 N | 9.90 W | 5 | 24 | Ramonet et al. (2010) | coastal |
| WAO | 52.95 N | 1.12 E | 20 | 10 | Wilson (2013) | coastal |
| CES | 51.97 N | 4.93 E | -1 | 207 | Vermeulen et al. (2011) | continental |
| EST | 51.66 N | 110.21 W | 707 | 3 | Worthy et al. (2003) | continental |
| FSD | 49.88 N | 81.57 W | 210 | 40 | Worthy et al. (2003) | continental |
| CPS | 49.82 N | 74.98 W | 381 | 8 | Worthy et al. (2003) | continental |
| ESP | 49.38 N | 126.54 W | 7 | 40 | Worthy et al. (2003) | coastal |
| KAS | 49.23 N | 19.98 E | 1989 | 5 | Necki et al. (2003) | mountain |
| SSL | 47.92 N | 7.92 E | 1205 | 12 | Schmidt et al. (2003) | mountain |
| HUN | 46.95 N | 16.65 E | 248 | 115 | Haszpra et al. (2001) | continental |
| JFJ | 46.55 N | 7.99 E | 3570 | 10 | Schibig et al. (2015) | mountain |
| LEF | 45.95 N | 90.27 W | 472 | 396 | Andrews et al. (2014) | continental |
| PUY | 45.77 N | 2.97 E | 1465 | 10 | Lopez et al. (2015) | mountain |
| AMT | 45.03 N | 68.68 W | 53 | 107 | Andrews et al. (2014) | continental |
| EGB | 44.23 N | 79.78 W | 251 | 3 | Worthy et al. (2003) | continental |
| WSA | 43.93 N | 60.01 W | 5 | 25 | Worthy et al. (2003) | remote |
| VAC | 42.88 N | 3.21 W | 1086 | 20 | Morgui et al. (2013) | mountain |
| TPD | 42.64 N | 80.56 W | 231 | 35 | Worthy et al. (2003) | continental |
| DEC | 40.74 N | 0.79 E | 1 | 10 | Morgui et al. (2013) | coastal |
| HDP | 40.56 N | 111.65 W | 3351 | 17.7 | Stephens et al. (2011) | mountain |
| SPL | 40.45 N | 106.73 W | 3210 | 9.1 | Stephens et al. (2011) | mountain |
| GIC | 40.35 N | 5.18 W | 1436 | 20 | Morgui et al. (2013) | mountain |
| NWR | 40.05 N | 105.59 W | 3523 | 3.5 | Stephens et al. (2011) | mountain |
| BAO | 40.05 N | 105.00 W | 1584 | 300 | Andrews et al. (2014) | continental |
| RYO | 39.03 N | 141.82 E | 260 | 20 | Tsutsumi et al. (2005) | coastal |
| SNP | 38.62 N | 78.35 W | 1008 | 17 | Andrews et al. (2014) | mountain |
| WGC | 38.26 N | 121.49 W | 0 | 483 | Andrews et al. (2014) | coastal |
| SGC | 36.70 N | 5.38 W | 850 | 20 | Morgui et al. (2013) | continental |
| SCT | 33.41 N | 81.83 W | 115 | 305 | Andrews et al. (2014) | continental |

**Table 4.** Continued from previous page

| Station ID | Latitude | longitude | altitude (m a.m.s.l) | intake (m a.g.l) | Reference | Type |
|---|---|---|---|---|---|---|
| WKT | 31.31 N | 97.33 W | 251 | 457 | Andrews et al. (2014) | continental |
| IZO | 28.31 N | 16.50 W | 2373 | 13 | Gomez-Pelaez and Ramos (2005) | mountain |
| YON | 24.47 N | 123.01 E | 30 | 20 | Tsutsumi et al. (2005) | coastal |
| MNM | 24.29 N | 153.98 E | 8 | 20 | Tsutsumi et al. (2005) | remote |
| MLO | 19.54 N | 155.58 W | 3397 | 40 | Thoning et al. (1989) | mountain |
| SMO | 14.25 S | 170.56 W | 42 | 10 | Halter et al. (1988) | remote |
| CPT | 34.35 S | 18.49 E | 230 | 30 | Brunke et al. (2004) | coastal |
| AMS | 37.80 S | 77.54 E | 55 | 20 | Gaudry et al. (1991) | remote |
| CGO | 40.68 S | 144.69 E | 94 | 70 | Francey et al. (2003) | coastal |
| CYA | 66.28 S | 110.52 E | 47 | 7 | Loh et al. (2017) | remote |
| SYO | 69.00 S | 39.58 E | 14 | 8 | NA | remote |
| SPO | 89.98 S | 24.80 W | 2810 | 10 | Conway and Thoning (1990) | remote |

**Table 5.** Comparison of Mean RMSE of hourly $CO_2$ from MPAS-A and IFS 9 km and 80 km simulations. p-values of paired t test between MPAS-A and the IFS simulations are also listed.

| | Number of data | Mean RMSE (ppm) | | | $p$ value of paired $t$-test | |
| --- | --- | --- | --- | --- | --- | --- |
| | | MPAS-A | IFS 9km | IFS 80 km | MPAS-A vs IFS 9 km | MPAS-A vs IFS 80 km |
| January 2014 | 50 | 4.20 | 3.12 | 4.94 | 0.01 | 0.25 |
| July 2014 | 50 | 8.09 | 8.04 | 11.77 | 0.95 | 0.04 |

**Table 6.** Comparison of RMSE of hourly $CO_2$ between the MPAS-A 60-15 km simulation and the IFS 9 km and 80 km simulations at 12 mountain sites (Table 4). The left half of the table is for six mountain sites located in MPAS-A's 15 km cells and the second half is for six mountain sites located in MPAS-A's 60 km cells. The top half of the table is for January 2014 and the bottom half is for July 2014.

| | | Sites at MPAS-A 15 km cells RMSE (ppm) | | | Sites at MPAS-A 60 km cells RMSE (ppm) | | | |
|---|---|---|---|---|---|---|---|---|
| | Site | IFS 9km | IFS 80 km | MPAS-A | Site | IFS 9km | IFS 80 km | MPAS-A |
| January 2014 | HDP | 3.10 | 19.71 | 1.17 | KAS | 4.44 | 10.71 | 16.65 |
| | SPL | 3.95 | 4.43 | 1.36 | SSL | 5.83 | 23.99 | 14.74 |
| | NWR | 1.64 | 3.74 | 1.78 | JFJ | 2.53 | 15.55 | 5.91 |
| | SNP | 5.01 | 14.54 | 4.36 | PUY | 4.58 | 10.30 | 5.94 |
| | IZO | 2.80 | 1.16 | 2.00 | VAC | 1.10 | 2.28 | 1.62 |
| | MLO | 0.85 | 1.25 | 0.77 | GIC | 5.60 | 4.74 | 7.40 |
| July 2014 | HDP | 5.99 | 37.37 | 2.92 | KAS | 4.29 | 17.57 | 7.17 |
| | SPL | 10.79 | 26.32 | 4.09 | SSL | 8.99 | 20.91 | 18.15 |
| | NWR | 5.17 | 18.78 | 3.71 | JFJ | 6.35 | 11.93 | 4.83 |
| | SNP | 29.28 | 48.33 | 12.88 | PUY | 7.23 | 13.29 | 12.80 |
| | IZO | 6.01 | 2.88 | 3.69 | VAC | 5.95 | 13.91 | 7.76 |
| | MLO | 1.47 | 1.68 | 1.31 | GIC | 20.30 | 15.36 | 28.58 |

**Table 7.** TCCON stations used for model evaluations.

| Site | Latitude | Longitude | Reference |
| --- | --- | --- | --- |
| Ascension Island | -7.92 | -14.33 | Feist et al. (2014) |
| Bialystok | 53.23 | 23.02 | Deutscher et al. (2015) |
| Bremen | 53.10 | 8.85 | Notholt et al. (2014) |
| Darwin | -12.43 | 130.93 | Griffith et al. (2014a) |
| Edwards | 34.96 | -117.88 | Iraci et al. (2016) |
| Garmisch | 47.48 | 11.06 | Sussmann and Rettinger (2015) |
| Izana | 28.31 | -16.48 | Blumenstock et al. (2017) |
| Saga | 33.24 | 130.29 | Kawakami et al. (2014) |
| Karlsruhe | 49.10 | 8.44 | Hase et al. (2015) |
| Lauder | -45.04 | 169.68 | Sherlock et al. (2014) |
| Lamont | 36.60 | -97.49 | Wennberg et al. (2014b) |
| Orleans | 47.97 | 2.11 | Warneke et al. (2014) |
| Parkfalls | 45.94 | -90.27 | Wennberg et al. (2014a) |
| Reunion Island | -20.90 | 55.49 | De Mazière et al. (2014) |
| Rikubetsu | 43.46 | 143.77 | Morino et al. (2016b) |
| Sodankyla | 67.37 | 26.63 | Kivi et al. (2014) |
| Tsukuba | 36.05 | 140.12 | Morino et al. (2016a) |
| Wollongong | -34.41 | 150.88 | Griffith et al. (2014b) |

**Table 8.** Statistics for the average hourly $XCO_2$ and average daily $XCO_2$ comparison between TCCON measurements and MPAS-A simulations: RMSE (ppm), bias (ppm), and correlation coefficient $R$. $N$ is the number of data pairs used for computing of the statistics.

| Site | | Average hourly $XCO_2$ | | | | Average daily $XCO_2$ | | |
|---|---|---|---|---|---|---|---|---|
| | $N$ | RMSE (ppm) | Bias (ppm) | $R$ | $N$ | RMSE (ppm) | Bias (ppm) | $R$ |
| Ascension Island | 1113 | 1.01 | 0.72 | 0.81 | 190 | 1.00 | 0.75 | 0.81 |
| Bialystok | 537 | 1.70 | 0.72 | 0.89 | 112 | 1.63 | 0.83 | 0.91 |
| Bremen | 222 | 2.30 | 1.04 | 0.85 | 51 | 2.20 | 1.15 | 0.87 |
| Darwin | 2109 | 1.18 | 0.85 | 0.68 | 296 | 1.06 | 0.79 | 0.77 |
| Edwards | 1515 | 1.01 | 0.57 | 0.90 | 257 | 0.91 | 0.57 | 0.93 |
| Garmisch | 567 | 1.10 | 0.18 | 0.91 | 99 | 1.14 | 0.20 | 0.91 |
| Izana | 210 | 0.51 | 0.24 | 0.94 | 56 | 0.51 | 0.26 | 0.94 |
| Saga | 516 | 0.97 | 0.30 | 0.91 | 107 | 0.95 | 0.29 | 0.91 |
| Karlsruhe | 522 | 1.99 | 0.92 | 0.85 | 93 | 1.73 | 1.05 | 0.88 |
| Lauder | 783 | 1.13 | 0.92 | 0.86 | 158 | 1.09 | 0.91 | 0.88 |
| Lamont | 1881 | 1.30 | 0.44 | 0.85 | 270 | 1.27 | 0.41 | 0.86 |
| Orleans | 573 | 2.06 | 0.95 | 0.75 | 114 | 1.84 | 1.01 | 0.81 |
| Parkfalls | 1200 | 1.35 | 0.17 | 0.93 | 194 | 1.27 | 0.15 | 0.94 |
| Reunion Island | 1092 | 1.03 | 0.86 | 0.91 | 186 | 1.00 | 0.86 | 0.93 |
| Rikubetsu | 180 | 1.26 | -0.03 | 0.93 | 43 | 1.21 | 0.09 | 0.94 |
| Sodankyla | 243 | 1.33 | 0.83 | 0.97 | 54 | 1.26 | 0.85 | 0.97 |
| Tsukuba | 1086 | 1.55 | 0.26 | 0.80 | 169 | 1.42 | 0.22 | 0.82 |
| Wollongong | 1146 | 1.22 | 0.80 | 0.78 | 194 | 1.17 | 0.81 | 0.81 |

**Table 9.** The duration of four ACT aircraft campaign seasons

| Campaign season | Duration |
|---|---|
| Summer 2016 | July 15 to August 28 |
| Winter 2017 | February 1 to March 10 |
| Fall 2017 | October 1 to November 15 |
| Spring 2018 | April 12 to May 20 |

**Table 10.** Evaluation of the simulated horizontal wind of MPAS-A using radiosonde observations at 457 stations located across the globe. Wind speed and direction are compared at 00:00 UTC and 12:00 UTC at four pressure levels (1000, 850, 500, and 200 hPa) for each day of the MPAS-A simulation. Note the number of data samples ($N$) is smaller at 1000 hPa because some stations are located above that pressure level.

| | Pressure level | Mean RMSE vector wind (m/s) | Bias Wind speed (m/s) | Mean difference Wind direction (°) | N |
|---|---|---|---|---|---|
| Summer2016 | 1000 hPa | 3.87 | 0.63 | 34.15 | 11,630 |
| | 850 hPa | 3.75 | -0.41 | 28.70 | 34,107 |
| | 500 hPa | 3.58 | -0.57 | 18.84 | 35,213 |
| | 200 hPa | 4.58 | -0.52 | 13.40 | 34,732 |
| Winter2018 | 1000 hPa | 4.02 | 1.01 | 32.00 | 11,415 |
| | 850 hPa | 4.11 | -0.32 | 25.28 | 27,957 |
| | 500 hPa | 4.03 | -0.43 | 14.67 | 28,977 |
| | 200 hPa | 4.55 | -0.48 | 9.79 | 28,401 |
| Fall2017 | 1000 hPa | 3.76 | 0.99 | 32.00 | 12,396 |
| | 850 hPa | 4.05 | -0.38 | 25.38 | 31,964 |
| | 500 hPa | 3.92 | -0.45 | 14.93 | 32,881 |
| | 200 hPa | 4.63 | -0.46 | 10.39 | 32,252 |
| Spring2018 | 1000 hPa | 3.98 | 0.89 | 33.80 | 10,763 |
| | 850 hPa | 4.05 | -0.30 | 27.85 | 29,886 |
| | 500 hPa | 4.12 | -0.42 | 17.24 | 30,914 |
| | 200 hPa | 4.79 | -0.46 | 12.96 | 30,257 |

**Table 11.** Comparison of mean $CO_2$ dry air mole fraction (ppm) in the boundary layer between the warm and the cold sectors. The table includes 10 frontal crossing flights from the summer 2016 season and 5 from winter 2017 season. The column labeled as diff is the mean value of the warm sector minus that of the cold sector.

| date | ACT | | | MPAS-A | | | WRF-Chem | | | CT2019 | | |
|------|------|------|------|------|------|------|------|------|------|------|------|------|
| yyyy-mm-dd | warm | cold | diff | warm | cold | diff | warm | cold | diff | warm | cold | diff |
| 2016-07-18 | 396.7 | 396.7 | 0.0 | 396.7 | 402.1 | -5.4 | 392.2 | 398.4 | -6.2 | 397.8 | 399.1 | -1.3 |
| 2016-07-19 | 398.2 | 396.6 | 1.6 | 404.9 | 393.5 | 11.4 | 400.2 | 394.3 | 5.9 | 399.6 | 397.9 | 1.7 |
| 2016-07-25 | 400.8 | 390.5 | 10.3 | 400.4 | 393.4 | 7.0 | 399.3 | 389.9 | 9.4 | 402.0 | 395.2 | 6.8 |
| 2016-07-26 | 405.9 | 396.1 | 9.8 | 419.4 | 394.8 | 24.6 | 424.8 | 394.0 | 30.8 | 408.4 | 396.3 | 12.1 |
| 2016-08-03 | 399.8 | 401.7 | -1.9 | 401.7 | 398.4 | 3.3 | 400.8 | 401.1 | -0.3 | 401.3 | 399.1 | 2.2 |
| 2016-08-04 | 407.3 | 393.5 | 13.8 | 408.2 | 391.3 | 16.9 | 407.5 | 399.8 | 7.7 | 403.0 | 390.7 | 12.3 |
| 2016-08-08 | 412.2 | 385.3 | 26.9 | 422.9 | 386.0 | 36.9 | 405.1 | 383.9 | 21.2 | 407.3 | 392.0 | 15.3 |
| 2016-08-12 | 401.4 | 395.1 | 6.3 | 404.4 | 392.1 | 12.3 | 405.4 | 402.8 | 2.6 | 399.6 | 395.3 | 4.3 |
| 2016-08-20 | 404.0 | 395.1 | 8.9 | 406.2 | 389.2 | 17.0 | 406.6 | 393.3 | 13.3 | 404.3 | 395.2 | 9.1 |
| 2016-08-21 | 406.5 | 390.7 | 15.8 | 408.1 | 387.0 | 21.1 | 414.8 | 392.5 | 22.3 | 404.1 | 394.1 | 10.0 |
| 2017-02-12 | 408.1 | 414.2 | -6.1 | 409.7 | 412.1 | -2.4 | 409.7 | 413.1 | -3.4 | 408.3 | 410.9 | -2.6 |
| 2017-02-17 | 413.5 | 414.8 | -1.3 | 411.8 | 415.1 | -3.3 | 411.5 | 413.2 | -1.7 | 411.5 | 414.0 | -2.5 |
| 2017-02-23 | 409.4 | 419.1 | -9.7 | 411.7 | 417.6 | -5.9 | 409.8 | 428.2 | -18.4 | 409.6 | 415.0 | -5.4 |
| 2017-03-07 | 412.0 | 415.2 | -3.2 | 415.3 | 417.1 | -1.8 | 417.4 | 418.4 | -1.0 | 413.7 | 413.8 | -0.1 |
| 2017-03-10 | 410.8 | 413.5 | -2.7 | 413.5 | 415.3 | -1.8 | 414.2 | 416.2 | -2.0 | 412.3 | 413.4 | -1.1 |

**Table 12.** Mean absolute error (MAE) of the simulated $CO_2$ of MPAS-A, WRF-Chem, and CT2019 as validated using ACT campaign aircraft $CO_2$ measurements. $p$ values of paired $t$ tests between MPAS-A and the other two models are included to provide a significance level for the model comparisons.

| Season | Number of profiles | Mean Absolute Error | | | p-value of paired t test | |
|---|---|---|---|---|---|---|
| | | MPAS-A | WRF-Chem | CT2019 | MPAS-A vs CT2019 | MPAS-A vs WRF-Chem |
| Summer 2016 | 72 | 3.80 | 4.38 | 3.03 | 0.06 | 0.21 |
| Winter 2017 | 27 | 1.56 | 2.23 | 1.58 | 0.95 | 0.09 |
| Fall 2017 | 40 | 2.55 | 3.25 | 3.16 | 0.04 | 0.11 |
| Spring 2018 | 60 | 2.29 | 3.75 | 1.99 | 0.23 | 0.01 |