# Peer review of "Development and evaluation of CO2 transport in MPAS-A v6.3"

_Geoscientific Model Development, 2020_

## Referee Comment (RC1) · Anonymous Referee #1 · 3 Nov 2020

This paper compares newly implemented CO2 output from variable resolution runs of the MPAS model to observations and two other modeling systems. The high-resolution region of the MPAS runs was centered over North America so most of the comparisons are made in this region but global comparisons with the relatively low resolution region of the model are also made. The MPAS CO2 output generally compares well with both the observations and other model output in the variety of known sharp gradient features, such as across cold fronts. The results suggest a successful implementation of CO2 into the MPAS model and the possibility to do focused high-resolution model runs in regions of interest.

The techniques and methodology are clearly described and the results are well supported by the figures and discussion. There are many grammatical errors and I have

tried to list some of them here but there are certainly more. My main comments are regarding figure clarification and are minor. I recommend publication with consideration of the specific comments below.

Specific comments:

Figure 6: You should add a label to the y-axis on at least the left column of plots. It's apparent that you are plotting $CO_2$ mole fraction in ppm but the labels should still be included. Also, I would recommend adding the labels 'ACT', 'MPAS', 'WRF-Chem' and 'CT2019' in the appropriate regions on at least one of the plots in the top row, or all of the top row plots to make it clear without reading the caption what is being plotted. Red 'warm sector' and blue 'cold sector' labels somewhere in the figure would also help. In line 3 of the caption 'measurement' should be 'measurements'. In line 5 I think you mean 'quartiles' instead of 'percentiles'.

Section 3.3.3: It would be helpful to include a summary of the warm-cold sector differences for each season and each model or measurement system. This could either be included in Table 4 or just in the text.

Figure 9: It's not clear if you intended to have a Figure 10 on Pg. 36 or continue Figure 9 onto two pages. In the caption of Figure 10 and 11 you refer to 'Figs. 9 and 9', which should either be 'Fig. 9' or 'Figs. 9 and 10' if you've split the figure into two. I would actually recommend moving Figure 9 into the supplement. 48 individual plots are too many for a single figure, or even two figures. One option to summarize the information in Figure 9 is to make normalized composites for each season. These could just emphasize the mole fraction differences across the fronts so that the measurements and models are each subtracted from their respective values at the front location. As the figure is now, the absolute value offsets are most visible rather than the differences across the front. The RMSEs calculated for Fig. 10 could also be based on the normalized mole fractions.

Technical comments:

Pg. 6, line 24: 'first set of simulations...'

Pg. 7, line 14: A question mark appears to be inadvertently left in parentheses.

Pg. 7, line 19: Did you mean 'reanalysis' instead of 'analysis'? Here and in the next line.

Pg. 8, line 16: Should be 'subscript k' instead of 'subscript j' for the vertical level.

Pg. 8, line 18: Should be '$h_{I,k}$' not '$h_{I,j}$'

Pg. 8, line 20: Should be '$rho_{i,k}$' not '$rho_{I,j}$' and '$q_{I,k}$' not '$q_{I,j}$'

Pg. 9, line 3: Add comma after 'MPAS'

Pg. 10, line 26: Why is this information in Table 6 instead of Table 3 since at this point in the text Tables 3-5 haven't been referred to yet? It seems to make sense to switch Tables 3 and 6 so they coincide with where they are discussed in the text.

Pg. 11, line 32: change 'he' to 'the'

Pg. 12, line 9: change 'resulted in' to 'had'

Pg. 12, line 10: 'accuracy than MPAS, likely because it applied...'

Pg. 12, line 12: change 'large' to 'larger'

Pg. 21, line 12: 'compared'

Pg. 12, line 31: 'in the next two sections...'

Pg. 13, line 5: add 'the' before 'warm'

Pg. 13, line 10: remove the second 'well' in this sentence.

Pg. 13, line 27: 'represents'

Pg. 13, line 32: One of the 'BL' subscripts should be changed to 'FT'

[Figure]

Pg. 14, line 2: 'observations'

Pg. 14, line 4: 'are a substantial. . .'

Pg. 14, line 5: 'majority of cases. . .', 'the rest of the three. . .'

Pg. 14, line 15: 'shows'

Pg. 14, line 22: 'seasons'

Pg. 14, line 34: should be 'Fig. S2'

Pg. 15, line 7: 'flights'

Pg. 15, line 30: 'MPAS performs. . .'

Pg. 16, line 24: 'this results in. . .'

Pg. 16, line 33: 'mean sea level. . .'
* * *

---

## Referee Comment (RC2) · Anonymous Referee #2 · 23 Dec 2020

General comments:

This paper implements CO2 transport in a numerical weather prediction model, MPAS-A. The unique feature of this model is that utilizing variable-resolution capability in the model simulation. It enables simulating CO2 and weather forecasts on a higher horizontal resolution (centred on North America in this paper) without lateral boundary conditions. The performance of the developed model is compared with two global modelling systems, CT2019 and IFS, and one regional modelling system, WRF-Chem. Also, the results are verified against radiosonde stations around the globe for weather forecasts and surface in-situ CO2 measurements and aircraft measurements from the ACT-America campaign for CO2 simulations. It seems that the implementation of CO2 transport is possibly successful, while it is not shown clearly. Although the paper fits

well within the scope of GMD, there is a large room for improvement toward being more rigorous in writing and better experimental design. Therefore, several major concerns listed below for the authors should be addressed to meet the quality of paper required by GMD.

Major comments:

1. Although MPAS-A CO2 is a global transport model, the experiment design is organized as though it tries to verify a regional model. Despite 50 surface CO2 stations around the globe are used in the verification, this only looks at the near surface, not at higher altitudes and larger scales. Using an additional global network (e.g., TCCON) and three-dimensional CO2 fields from a global model (e.g., CT2019 or IFS) would provide complementary verification on the performance of the model for the whole domain. You could refer to Agusti-Panareda et al. (2014) and Polavarapu et al. (2016), which are already cited in the paper, to get an idea of how to verify the model on the global scale. Also, for presenting results, it is better to investigate the global simulation results first. Then, you can narrow down the scale from the globe to eastern United States where ACT data are measured.

2. The simulation period is too short for investigating the model performance, including mass conservation (details in specific comments) and CO2 simulation. At least, one or more year simulation is necessary to see how the model works. Because the initial condition of CO2 is taken from CT2019, without a proper spin-up period, the CO2 field from 60 km grids still has a signature of CT2019 transport, not MPAS transport. Authors argue that one benefit of the approach that MAPS CO2 uses is the consistent LBC from the coarse domain. To take advantage of such consistent LBC that MPAS produces, the simulation period should be extended.

3. Mass conservation is one of the major concerns in this paper. Since the model is run without surface CO2 flux, the results shown here are not enough to prove the mass conservation is acquired in the model. The model should run with ingesting surface

[Figure]

CO2 fluxes that have complex and strong spatial gradients. Also, it should run for an extended period as well. Please see the detail for this issue in specific comments.

4. Is it possible to run MPAS CO2 without the local grid refinement? If so, it is possible to investigate the benefit and impact of using local grid refinement on the performance of CO2 simulation by running the model with and without a higher-resolution grid. Also, it is associated with the question about the parametrisation scheme in the specific comments. Maybe authors can highlight the benefit of the unique feature of MPAS by adding coarse resolution results in Table 3 or Table S3, for example.

5. Although two global models (IFS and CT2019) are presented in the paper to compare MPAS results, CT2019 is only used to be compared with ACT data. Why isn't CT2019 used in the verification on the global scale (in section 3.4)?

6. Two different sampling methods (taking nearest land cell to a given location interpolation in space and time) are used in two different sections. But a discussion on this is missing in the paper.

Specific comments:

1. Despite the developed model is named MPAS-A in the title, MPAS-A and MPAS are flurried through the manuscript, making confusion to readers.

2. P1L6: "only major research and operation centers can afford it" would be not necessary because the definition is somewhat vague. This is the same in the introduction.

3. P1L15: Why all hourly data is used? The statistics for sure is exacerbated by including nighttime data.

4. P1L21: The conclusion of comparisons with WRF-Chem and CT2019 is missing

5. P1L24: I don't think "often" is necessary for this sentence.

6. P1L24: It needs to use surface CO2 flux rather than CO2 flux because there are other usages for CO2 fluxes used throughout the manuscript (e.g., horizontal CO2

fluxes and vertical CO2 fluxes).

7. P2L7-8: It needs to add a reference for the sentence.

8. P2L11: What does "simulation resolution" mean by?

9. P2L18-19: This sentence is a little bit confusing. A regional model could require more computational cost than a global model does. Because the cost of a model depends on the configuration (e.g. the number of grids and the period of interest, etc.). Thus, it needs to clarify it.

10. P3L23: It seems that MPAS CO2 is an online transport model. But it is not mentioned explicitly in the paper.

11. P5L20: Does "CO2 eddy diffusivity" mean vertical eddy diffusivity for CO2? And is there any cap (minimum or maximum) applied to Kh to prevent too strong or weak vertical mixing?

12. P7L16: What does third-order advection mean?

13. P7L17: Table 1, are parameterisation schemes here an optimal choice for the best CO2 simulation by MPAS? An explanation of why these parameterisations are used in the simulation is lacking.

14. P7L17: How do parameterisation schemes work for the weather forecast and CO2 transport on a variable-resolution grid? Is it anticipated that parameterisation schemes work identically with the different area of a grid?

15. P7L19: Because of the difference in grid between ERA-I and MPAS, how is the conservation of tracer mass guaranteed after interpolation of wind fields to the MPAS grid? Is there any kind of "pressure fixer" as used in GEOS-Chem or a flux adjustment as done by CarbonTracker (Segers et al., 2002)?

16. P8L25, Fig. 2: Are fluxes used in these simulations? If not, the simulation should be repeated with CO2 surface flux inputs. This is important because the surface fluxes

create strong gradients which challenge a model's numerical schemes. Without surface fluxes, the CO2 field is very smooth and easy to model. Also, can you increase the length of the simulation?

17. P8L28: What does maximal variation mean? The change from one data point to the next? If so, it seems much smaller than 1.95E-12 kg/mˆ3 for dry air in Fig. 2. If it is the maximum value in the plot, it seems to low since the maximum is around 4E-12.

18. P9L11: It sounds like Dee et al. mentioned about dry air mass change. Since the reference is already cited in the paper. This one could be taken off.

19. P9L26, Fig. 3: This test should be repeated with realistic CO2 surface fluxes. You know what the flux is over 3 hours so the change in global CO2 mass is known. Then the expected total CO2 from fluxes, assuming an initial value, can be compared with the model global CO2 mass. Fluxes create strong gradients that are usually challenging for the numerics, and you will see more realistic magnitudes of temporal variations. Those changes in Fig. 3 may be unrealistically low. Also 48 days is too short. Tests for a year would be good.

20. P10L8-17: the paragraph does not include information about section 3.3.1. Please revise the text.

21. P10L16: The current configuration heavily relies on the IC from CarbonTracker, due to the short period of simulation. It does not show the quality of CO2 transport by MAPS CO2 on a global scale. This is another reason why the simulation period should be extended.

22. P10L19: The conclusion for section 3.3.1 is missing.

23. P10L20-21: It is not clear why this sentence is here. Has Michaelis et al. verified MPAS simulation already? If so, please rephrase it.

24. P10L23: For the verification at 00 UTC, are 24 h forecasts at 0 UTC used? Since the meteorological re-initialisation at the 24-hour interval, it needs to clarify it.

25. P10L24: Fig. S3 is referred to earlier than Fig. S2.

26. P10L26: Table 6 is referred to earlier than Table 3, 4 and 5.

27. P11L3: Because section 3.3.2 utilizes data focused on North America, not on the globe, the title should be revised accordingly.

28. P11L9: "regional NWP WRF" sounds strange. Please clarify it.

29. P11L14: Which "the analysis" do you mean? ERA5 reanalysis or ERA-interim analysis?

30. P12L4: The PBL height between models are probably different, how did you consider that in Fig. 4 in order to obtain a fair comparison?

31. P12L4-5: Flux error would be not the main culprit. It can be also attributed to the larger error in the weather forecast in BL than in FT, associated with the accuracy of PBL height in the model simulation.

32. P12L8: Although Fig. 4 shows dots with different colours to represent results from different seasons, the differences between seasons are not explained explicitly in the text. This is the case in Fig. 5.

33. P12L9-10: This sentence is a contradiction with what is mentioned in the introduction (3rd paragraph). This sentence sounds like that a regional model using the nudging scheme is better than a global model that does not require any lateral boundary conditions.

34. P12L12-13: Why does a coarse resolution model simulate better atmospheric $CO_2$ than a higher resolution model?

35. P12L23: Fig. 4 and Fig. 5 can be combined into a single figure using Taylor diagram. Then, the interpretation of results may be easier and more concise.

36. P13L10: In Fig. 6, please add names on the x-axis to make easy to understand

the figure and add the unit on y-axis or in the caption.

37. P13L13, Table 4: I don't see this date (2016-08-24) in Table 4. There are cases where CT2019 does better than the other 2. The date mentioned in the text may be cherry-picked. To get a better sense of how often CT2019 does better/worse than the higher resolution models, please list the differences between warm and cold in a separate column for easy comparison between the models and observations.

38. P13L19: The explanation about the sampling strategy is missing in this section, in particular vertical sampling method. Results might be sensitive to the method.

39. P14L20, Table 5: The sample sizes are quite small so some significance test would be useful to evaluate if the differences in scores between cases are important. In fact, when comparing scores between models, in other Tables as well, significance test would be useful.

40. P14L21-23: The sentence is difficult to understand, please rephrase it

41. P14L32, Fig. 8: Presumably, this figure is shown as an example to demonstrate how MPAS CO2 simulates the CO2 enhancement well at the specific date. Since overall CO2 enhancement is mentioned in Fig. 9. It may be better to swap Fig. 8 and Fig. 9 to make structure organized better.

42. P14L34: It is the wrong figure number. Please change Fig. S2 and Fig. S3.

43. P15L3: Numbers in Fig. 8 are very difficult to read. Please change the font size or colour or both.

44. P15L13: It is difficult to find when the CO2 enhancement is shown. Maybe it would be helpful to add a mark in Fig. 9 to indicate when the CO2 enhancement happened. Otherwise, as there are too many panels in the single figure, you may consider to keep the small number of panels in Fig. 9 and to move the rest of them to supplementary.

45. P15L17: It is difficult to recognize colour lines without information. Please add a

legend for lines in Fig. 9.

46. P15L18: I cannot see the date 2017-10-28 in Fig. 9.

47. P15L19: Looking at the date 2017-11-03, the CO2 enhancements happen not only at the front boundary but also around other locations such as at 19 and 19.8 UTC in the panel. Can you explain why CO2 enhancements are shown at other times either? Isn't the CO2 enhancement a unique feature happening at the front?

48. P15L25: What is the "level-leg flights"?

49. P16L3: Because the developed model is a global model, it would be better to evaluate result over the globe first then narrow down the scales of interest (North America). In that sense, section 3.4 should be placed before section 3.3. Also, including more observations (e.g., TCCON) in section 3.4 is beneficial to evaluate the behaviour of the global scale CO2 transport by MPAS CO2, with an extended simulation period.

50. P16L14: What is the difference between Mean RMSE vector wind and RNSE wind speed in Table 3? Since RMSE wind speed is not mentioned in the text, it is difficult to understand why two different metrics are shown separately.

51. P16L14-15: I am just curious that numbers for the mean difference wind direction in Table 3 are always positive by any chance.

52. P16L15: Table 3 is mentioned later than Table 4 and 5. Please change the order.

53. P16L25-26: Vertical sampling method is missing. It is better to add a discussion about vertical sampling strategy. In addition, why is the horizontal sampling method used in this section different from that in the previous section?

54. P17L8: Because MPAS utilizes variable horizontal resolution (60-15 km), it may be possible to find a benefit of higher horizontal resolutions by splitting results into two groups, one on 60 km grid and the other on 15 km. Results at sites on 15 km grid may be comparable with results in IFS 9 km.

55. P17L8: Why are 46 stations used in the calculation rather than 50 stations? This is a different number from what is mentioned above (50 stations).

56. P17L10: Table S2 is mentioned later than Table S3.

57. P17L17: The sentence sounds strange. Please rephrase it.

58. P37: Figure 10. In the caption, what does "Figs. 9 and 9" mean?

Technical corrections:

1. Overall, it was able to find lots of typos and technical corrections, grammar and format issues. So, presented here might be not the completed set.

2. The subscript for 2 is missing in $CO_2$ in many places, including the main text, captions and figures, throughout the paper.

3. Many acronyms are defined in wrong places or multiple times. Please correct them.

4. P2L13: This is the first place to define PBL.

5. P2L28: This is the first place to define FT.

6. P3L13: This is not the first place to define PBL.

7. P3L18: Remove space between ")" and ","

8. P4L23: "planetary boundary layer (PBL)" -> PBL

9. P4L24: This is the first place to define BL.

10. P6L12: Carbon dioxide -> $CO_2$

11. P7L14: What is "(?)"?

12. P7L21: Carbon dioxide -> $CO_2$

13. P7L24: $CO_2$ fluxes -> surface $CO_2$ fluxes

14. P9L4: Fig.2 -> Fig. 2 (add space)

15. P10L7: It may be a typo of evaluation.

16. P10L14: boundary layer -> BL

17. P10L24: 850 is duplicated

18. P11L9: chemistry transport model -> CTM

19. P11L11: Missing year for Hersbach et al.

20. P11L12: boundary conditions -> lateral boundary conditions

21. P11L12: Jacoboson et al. (2007) is not the proper reference for CarbonTracker.

22. P11L13: CO2 fluxes -> surface CO2 fluxes

23. P11L14: Jacobson al. (2007) is the not proper reference for CarbonTracker.

24. P11L15: atmosphere -> atmospheric

25. P11L16: chemical transport model -> CTM

26. P11L17: planetary boundary layer -> PBL

27. P11L21: Remove space between ")" and ","

28. P11L25: change double parenthesis to single

29. P11L32: It is not the first place to define BL and FT.

30. P11L32: he -> the

31. P12L6: exception -> exceptions

32. P12L9: free troposphere -> FT

33. P13L24: "boundary layer (BL) and free toposphere (FT)" -> BL and FT

34. P14L15: 2015? It might be a typo.

35. P14L15: Fall -> fall

36. P14L15: show -> shows

37. P14L17: tend -> tends

38. P14L29: boundary layer -> BL

39. P15L11: wrong order; 2017-02-23 after 2017-03-10

40. P16L32: "The Schauinsland station" should be moved to P16L29.

41. P17L2: Add space between "23.99" and "ppm".

42. P30: Figure 4: Unit is missing in the caption or figure.

43. P32: Figure 6: Unit is missing in the caption or figure. Add names on x-axis.

44. P43: Table 3. Add space between number and the unit (m/s) and parenthesis for degree. Numbers should be integer.

45. P44: Table 4. Add "date (yyyy-mm-dd)" at the top left.

46. P45: Table 5. Num -> Number

47. P47: Table 7. Make Station IDs capital letters (including main text).

References

Agusti-Panareda, A., Massart, S., Chevallier, F., Boussetta, S., Balsamo, G., Beljaars, A., Ciais, P., Deutscher, N. M., Engelen, R., Jones, L., Kivi, R., Paris, J.-D., Peuch, V.-H., Sherlock, V., Vermeulen, A. T., Wennberg, P. O., and Wunch, D.: Forecasting global atmospheric CO2, Atmospheric Chemistry and Physics, 14, 11 959–11 983, doi:10.5194/acp-14-11959-2014, https://www.atmos-chem-phys.net/14/11959/5 2014/, 2014.

Polavarapu, S. M., Neish, M., Tanguay, M., Girard, C., de Grandpré, J., Semeniuk, K., Gravel, S., Ren, S., Roche, S., Chan, D., and Strong, K.: Greenhouse gas simulations

with a coupled meteorological and transport model: the predictability of CO2, Atmospheric Chemistry and Physics, 16, 12 005–12 038, doi:10.5194/acp-16-12005-2016, 2016.

Segers, A., P. van Velthoven, B. Bregman, and M. Krol (2002), On the computation of mass fluxes for Eulerian transport models from spectral meteorological fields, in Computational Science - ICCS 2002: International Conference, Lecture Notes in Computer Science, vol. 2330, edited by P. Sloot et al., pp. 767–776, Springer-Verlag, New York.

---

## Author Comment (AC1) · 26 Mar 2021

**Response to reviewer 1**

The authors would like to thank the reviewer for the insightful comments which have been thoroughly addressed below and have contributed to improving the clarity of the manuscript. In the following sections, the reviewer's original comments are in blue and our response are in black.

This paper compares newly implemented CO2 output from variable resolution runs of the MPAS model to observations and two other modeling systems. The high-resolution region of the MPAS runs was centered over North America so most of the comparisons are made in this region but global comparisons with the relatively low resolution region of the model are also made. The MPAS CO2 output generally compares well with both the observations and other model output in the variety of known sharp gradient features, such as across cold fronts. The results suggest a successful implementation of CO2 into the MPAS model and the possibility to do focused high-resolution model runs in regions of interest.

The techniques and methodology are clearly described and the results are well supported by the figures and discussion. There are many grammatical errors and I have tried to list some of them here but there are certainly more. My main comments are regarding figure clarification and are minor. I recommend publication with consideration of the specific comments below.

We really appreciate the referee's comprehensive and constructive comments. Through addressing these comments, we have substantially improved the quality of the manuscript.

**Specific comments**

1. Figure 6: You should add a label to the y-axis on at least the left column of plots. It's apparent that you are plotting CO2 mole fraction in ppm but the labels should still be included. Also, I would recommend adding the labels 'ACT', 'MPAS', 'WRF-Chem' and 'CT2019' in the appropriate regions on at least one of the plots in the top row, or all of the top row plots to make it clear without reading the caption what is being plotted. Red 'warm sector' and blue 'cold sector' labels somewhere in the figure would also help. In line 3 of the caption 'measurement' should be 'measurements'. In line 5 I think you mean 'quartiles' instead of 'percentiles'.

Thanks for pointing this out.
Please note this figure is Fig. 12 in the revised the manuscript.
The figure has been updated: (1) In figure, a Y-axis has been added and labeled with $CO_2$ dry air mole fraction (ppm). On the X-axis, labels are added for "ACT", "MPAS-A", "WRF-Chem", "CT2019".
(2) "measurement" is corrected to "measurements".
(3) "percentiles" is corrected to "quartiles".

[Figure]

**Figure Caption**: Comparison of $CO_2$ mole fraction in frontal-crossing level-leg flights in boundary layer between ACT aircraft measurements and model simulations. Flight date and aircraft type are labeled in title for each flight leg. X-axis is UTC time, and Y-axis is $CO_2$ mole fraction (ppm). Aircraft measurements are in black, MPAS-A in red, WRF-Chem in blue, and CT2019 in green. In each figure, the pair of vertical dashed lines mark $CO_2$ enhancement observed by the aircraft along a frontal boundary, and the warm and cold sectors associated with the frontal boundary are labeled as warm and cold, respectively.

2. Section 3.3.3: It would be helpful to include a summary of the warm-cold sector differences for each season and each model or measurement system. This could either be included in Table 4 or just in the text.

Following the referee's suggestion, the following statements have been added to the revised manuscript (P16 Lines 10-16):

*Table 11 lists the mean $CO_2$ of the warm sector, cold sectors, and their difference as calculated from the ACT measurements, MPAS-A, WRF-Chem, and CT2019. The table shows that the MPAS-A simulations are similar to WRF-Chem, and both tend to have larger $CO_2$ differences between the warm and cold sectors than CT2019. For instance, the 2016-08-08 case where the observed mean $CO_2$ difference between warm and cold sector is 26.9 ppm, MPAS-A and WRF simulations resulted in 36.9 ppm and 21.2 ppm respectively, while CT2019 resulted in a 15.3 ppm difference. The above evaluation indicates that MPAS-A $CO_2$ model is capable of well representing the observed $CO_2$ difference between the warm and cold sectors, and its accuracy in this respect is comparable to WRF-Chem and CT2019.*

3. Figure 9: It's not clear if you intended to have a Figure 10 on Pg. 36 or continue Figure 9 onto two pages. In the caption of Figure 10 and 11 you refer to 'Figs. 9 and 9', which should either be 'Fig. 9' or 'Figs. 9 and 10' if you've split the figure into two. I would actually recommend moving Figure 9 into the supplement. 48 individual plots are too many for a single figure, or even two figures.

One option to summarize the information in Figure 9 is to make normalized composites for each season. These could just emphasize the mole fraction differences across the fronts so that the measurements and models are each subtracted from their respective values at the front location. As the figure is now, the absolute value offsets are most visible rather than the differences across the front. The RMSEs calculated for Fig. 10 could also be based on the normalized mole fractions.

Following the reviewer's suggestion, the figure has been remade to include only eight cases while the the full list of cases are provided in the supplement Figure S3.
In the updated figure, the $CO_2$ enhancement along the frontal boundaries in the aircraft measurement are marked with the two dashed vertical lines.

[Figure]

**Figure caption**: Comparison of $CO_2$ mole fraction in frontal-crossing level-leg flights in boundary layer between ACT aircraft measurements and model simulations. Flight date and aircraft type are labeled in title for each flight leg. X-axis is UTC time, and Y-axis is $CO_2$ mole fraction (ppm). Aircraft measurements are in black, MPAS-A in red, WRF-Chem in blue, and CT2019 in green. In each figure, the pair of vertical dashed lines mark $CO_2$ enhancement observed by the aircraft along a frontal boundary, and the warm and cold sectors associated with the frontal boundary are labeled as warm and cold, respectively.

**Technical comments**

Pg. 6, line 24: 'first set of simulations. . .'
Fixed. It has been changed from "first set simulations" to "first set of simulations".

Pg. 7, line 14: A question mark appears to be inadvertently left in parentheses.
Fixed.

Pg. 7, line 19: Did you mean 'reanalysis' instead of 'analysis'? Here and in the next line.
Fixed. 'analysis' has been changed to 'reanalysis' in both cases.

Pg. 8, line 16: Should be 'subscript k' instead of 'subscript j' for the vertical level.
Fixed. Subscript $j$ has been changed to subscript $k$.

Pg. 8, line 18: Should be '$h_{i,k}$' not '$h_{i,j}$'
Fixed.

Pg. 8, line 20: Should be 'rho i,k' not 'rho I,j' and 'q I,k' not 'q I,j'
Fixed. $\rho_{i,j}$ has been changed to $\rho_{i,k}$, $q_{i,j}$ has been changed to $q_{i,k}$.

Pg. 9, line 3: Add comma after 'MPAS'
Fixed.

Pg. 10, line 26: Why is this information in Table 6 instead of Table 3 since at this point in the text Tables 3-5 haven't been referred to yet? It seems to make sense to switch Tables 3 and 6 so they coincide with where they are discussed in the text.
Fixed. The order of the tables has been corrected in the revised manuscript.

Pg. 11, line 32: change 'he' to 'the'
Fixed.

Pg. 12, line 9: change 'resulted in' to 'had'
Fixed.

Pg. 12, line 10: 'accuracy than MPAS, likely because it applied. . .'
Fixed.

Pg. 12, line 12: change 'large' to 'larger'
Fixed.

Pg. 21, line 12: 'compared'
Fixed.

Pg. 12, line 31: 'in the next two sections. . .'
Fixed.

Pg. 13, line 5: add 'the' before 'warm'

Fixed.

Pg. 13, line 10: remove the second 'well' in this sentence.
Fixed.

Pg. 13, line 27: 'represents'
Fixed.

Pg. 13, line 32: One of the 'BL' subscripts should be changed to 'FT
Fixed.

Pg. 14, line 2: 'observations'
Fixed.

Pg. 14, line 4: 'are a substantial. . .'
Fixed.

Pg. 14, line 5: 'majority of cases. . .', 'the rest of the three. . .'
Fixed.

Pg. 14, line 15: 'shows'
Fixed.

Pg. 14, line 22: 'seasons'
Fixed.

Pg. 14, line 34: should be 'Fig. S2'
Fixed.

Pg. 15, line 7: 'flights'
Fixed.

Pg. 15, line 30: 'MPAS performs. . .'
Fixed.

Pg. 16, line 24: 'this results in. . .'
Fixed.

Pg. 16, line 33: 'mean sea level. . .'
Fixed.

---

## Author Comment (AC2) · 26 Mar 2021

**Response to reviewer 2**

The authors would like to thank the reviewer for the insightful comments which have been thoroughly addressed below. The experimental design and clarity of the manuscript have been substantially improved as a result of addressing the reviewer's comments. The major changes we made to the experiment design and model evaluation following the reviewer's suggestions include:

- We have extended the MPAS-A simulation for testing the mass conservation to one-year long and using 3-hourly CT2019 surface $CO_2$ fluxes to drive the simulation.

- We have extended the MPAS-A simulation for evaluating $CO_2$ at the global scale to one-year long with a 6-month spinup period.

- MPAS-A simulation for the $CO_2$ evaluation at regional scale using the ACT airborne measurements is now a 2.5-year long continuous simulation, starting 6 months prior to the first ACT campaign.

- An evaluation using $XCO_2$ from 18 TCCON stations located across the globe has been added.

- A comparison with CT2019 $CO_2$ at the $1° \times 1°$ grid has been added.

In the following sections, the referee's original comments are in blue and our response are in black.

**General comments**

This paper implements CO2 transport in a numerical weather prediction model, MPASA. The unique feature of this model is that utilizing variable-resolution capability in the model simulation. It enables simulating CO2 and weather forecasts on a higher horizontal resolution (centred on North America in this paper) without lateral boundary conditions. The performance of the developed model is compared with two global modelling systems, CT2019 and IFS, and one regional modelling system, WRF-Chem. Also, the results are verified against radiosonde stations around the globe for weather forecasts and surface in-situ CO2 measurements and aircraft measurements from the ACT-America campaign for CO2 simulations. It seems that the implementation of CO2 transport is possibly successful, while it is not shown clearly. Although the paper fits well within the scope of GMD, there is a large room for improvement toward being more rigorous in writing and better experimental design. Therefore, several major concerns listed below for the authors should be addressed to meet the quality of paper required by GMD.

We really appreciate the reviewer's insightful comments. We have thoroughly addressed each of the reviewer's comments and suggestions. Our point-by-point responses are listed below.

**Major comments**

1. Although MPAS-A CO2 is a global transport model, the experiment design is organized as though it tries to verify a regional model. Despite 50 surface CO2 stations around the globe are

used in the verification, this only looks at the near surface, not at higher altitudes and larger scales. Using an additional global network (e.g., TCCON) and three-dimensional CO2 fields from a global model (e.g., CT2019 or IFS) would provide complementary verification on the performance of the model for the whole domain. You could refer to Agusti-Panareda et al. (2014) and Polavarapu et al. (2016), which are already cited in the paper, to get an idea of how to verify the model on the global scale. Also, for presenting results, it is better to investigate the global simulation results first. Then, you can narrow down the scale from the globe to eastern United States where ACT data are measured.

We agree with the reviewer.
Following the reviewer's suggestion we made the following changes in the revised manuscript:
(1) carried out the MPAS-A simulations with an extended period;
(2) evaluated MPAS-A simulation results at 18 TCCON sites;
(3) compared MPAS-A simulation results with CT2019 $CO_2$ mole fraction on the global scale;
(4) reorganized the manuscript so that MPAS-A is evaluated at the global scale first and then at the regional scale.

2. The simulation period is too short for investigating the model performance, including mass conservation (details in specific comments) and CO2 simulation. At least, one or more year simulation is necessary to see how the model works. Because the initial condition of CO2 is taken from CT2019, without a proper spin-up period, the CO2 field from 60 km grids still has a signature of CT2019 transport, not MPAS transport. Authors argue that one benefit of the approach that MAPS CO2 uses is the consistent LBC from the coarse domain. To take advantage of such consistent LBC that MPAS produces, the simulation period should be extended.

We agree with the reviewer.
Following the reviewer's suggestion, we conducted a longer period simulation for assessing the model's mass conservation and $CO_2$ simulation performance. In the revised manuscript:
(1) Mass conservation is assessed based on one-year (year 2014) simulation with surface $CO_2$ flux ingested at 3-hour intervals.
(2) $CO_2$ transport evaluation is based on a MPAS-A simulation that lasts one-year (year 2014) in addition to a 6-month spinup period (July 01 2013 to December 31 2014).
(3) regional scale evaluation using the ACT airborne measurements is based on the extended simulation (January 1 2016 to June 1 2018). As the first ACT campaign started in July 2016, the new simulation has an six-month spin-up period prior to the evaluation.

3. Mass conservation is one of the major concerns in this paper. Since the model is run without surface CO2 flux, the results shown here are not enough to prove the mass conservation is acquired in the model. The model should run with ingesting surface CO2 fluxes that have complex and strong spatial gradients. Also, it should run for an extended period as well. Please see the detail for this issue in specific comments.

We agree with the reviewer that mass conservation should be tested on longer-period simulations and with surface CO2 fluxes ingested during the simulation.
Following this suggestion, we conducted a one-year MPAS-A simulation during which CT2019 surface CO2 flux are ingested at 3-hour intervals. The global conservation of dry air and $CO_2$ are evaluated using this simulation results.

Please see our response to the reviewer's specific comment #16 for more details.

4. Is it possible to run MPAS CO2 without the local grid refinement? If so, it is possible to investigate the benefit and impact of using local grid refinement on the performance of CO2 simulation by running the model with and without a higher-resolution grid. Also, it is associated with the question about the parametrisation scheme in the specific comments. Maybe authors can highlight the benefit of the unique feature of MPAS by adding coarse resolution results in Table 3 or Table S3, for example.

Yes, MPAS-A can run on both variable resolution grids and uniform resolution grids. Following the reviewer's suggestion, we conducted an additional simulation on a uniform global 60 km grid. The comparison of the simulated horizontal wind accuracy between the uniform 60 km simulation and the variable resolution 60-15 km simulation are summarized. The following Table (Table 3 in the revised manuscript) has been added

**Figure Caption**:The statistics of MPAS-A simulated horizontal wind validated at radiosonde stations located in North America. In each cell, the first value is from the 60-15 km variable-resolution grid simulation (labeled as 15 km) and the second is from the 60 km uniform grid simulation (labeled as 60 km). Note that the number of data at the 850 hPa and 1000 hPa are different between the two simulations because of the differences in their grids' topography.

| | Pressure level | Mean RMSE vector wind (m/s) | | Bias Wind speed(m/s) | | Mean difference Wind direction ° | | Number of data | |
|---|---|---|---|---|---|---|---|---|---|
| | | 15 km | 60 km | 15 km | 60 km | 15 km | 60 km | 15 km | 60 km |
| January 2014 | 1000 hPa | 3.46 | 3.98 | 0.84 | 1.02 | 30.27 | 32.24 | 2,427 | 1,845 |
| | 850 hPa | 3.42 | 4.08 | -0.32 | -0.59 | 21.58 | 24.70 | 6,227 | 6,187 |
| | 500 hPa | 3.37 | 3.68 | -0.41 | -0.54 | 13.10 | 13.89 | 6,659 | 6,659 |
| | 200 hPa | 4.01 | 4.20 | -0.44 | -0.53 | 8.54 | 8.78 | 6,536 | 6,536 |
| July 2014 | 1000 hPa | 3.10 | 3.64 | 0.13 | 0.39 | 32.91 | 33.93 | 2,778 | 2,027 |
| | 850 hPa | 3.34 | 4.08 | -0.35 | -0.60 | 26.08 | 27.66 | 6,395 | 6,321 |
| | 500 hPa | 3.26 | 3.68 | -0.54 | -0.65 | 17.46 | 18.25 | 6,861 | 6,861 |
| | 200 hPa | 3.76 | 4.06 | -0.41 | -0.40 | 10.99 | 11.70 | 6,808 | 6,808 |

The following statements are added in the revised manuscript (Page 11 Lines 8-22 ):
*An important finding of Agusti-Panareda et al. (2019) is that higher horizontal resolution generally lead to higher meteorological and $CO_2$ simulation accuracy. To examine the influence of horizontal resolution on MPAS-A's meteorological simulation accuracy, we conducted an additional set of simulation using the identical configuration except that it uses a global 60 km uniform-resolution grid instead of the 60-15 km variable-resolution grid (Fig. 1). Out of the 475 radiosonde stations, 131 are located at 15 km cells in the 60-15 km variable-resolution simulation. These 131 radiosonde stations are all located at 60 km cells in the 60 km uniform-resolution simulation. In Table 3, we calculated and compared horizontal wind accuracy at these 131 radiosonde stations between the 60 km uniform-resolution simulation (labeled as 60 km) and the 60-15 km variable-resolution simulation (labeled as 15 km). The table shows that the horizontal wind fields at these 131 stations are simulated with considerably higher accuracy on the 15 km grid than its 60 km grid counterpart. For instance at 1000 hPa, the mean RMSE wind vector for January 2014 is 3.46 m/s and 3.98 m/s at the 15 km and 60 km grids respectively. The values are 3.10 m/s and 3.64 m/s for July 2014. Table 3 also shows that the difference in the mean RMSE wind vector between the 15 km and 60 km grids*

*is larger near the surface at 850 and 1000 hPa than in the middle and upper troposphere (500 and 200 hPa), which is consistent with the findings of Agusti-Panareda et al. (2019). For both January and July at the four pressure levels, the mean RMSE wind vector at the 131 radiosonde stations at MPAS-A's 15 km grid is either similar to or slightly lower than the mean RMSE wind vector of the around 400 stations from the IFS 9 km resolution simulation (Agusti-Panareda et al., 2019).*

5. Although two global models (IFS and CT2019) are presented in the paper to compare MPAS results, CT2019 is only used to be compared with ACT data. Why isn't CT2019 used in the verification on the global scale (in section 3.4)?

We agree that CT2019 should be used for MPAS-A evaluation at the global scale.
Following (Polavarapu et al., 2016), we directly compared $XCO_2$ from MPAS-A simulation and CT2019 at grid scale. In the revised manuscript, a new section (Sect. 3.3.2) has been added for MPAS-A comparison with CT2019 at the global scale.

[Figure]

**Figure caption**: $XCO_2$ of MPAS-A, CT2019, and their difference at 2014-07-01 and 2014-12-01 00:00 UTC.

The following statements have been added in the revised manuscript (Page 11 Line 23- Page 12 Line 8):
*Having established that the horizontal wind fields simulated by MPAS-A are sufficiently accurate, the $CO_2$ fields can be evaluated. First we directly compare $XCO_2$ from MPAS-A and CT2019*

field at the grid scale. First, $XCO_2$ are calculated at the native grid for MPAS-A (60-15km) and CT2019 ($3° \times 2°$). $XCO_2$ at a given model cell is calculated as the pressure weighted $CO_2$ dry air mixing ratio.

$$XCO_2 = (\sum_{k=1}^{N} p_k q_k^{co_2})/(\sum_{k=1}^{N} p_k)$$

where $p_k$ is modeled air pressure at layer $k$ corrected for water vapor, $q_k^{co_2}$ is $CO_2$ dry air mole fraction at the same level. $N$ is the number of vertical levels in a model. Then, $XCO_2$ from MPAS-A and CT2019 are regridded from their respective grids an identical $1 \times 1°$ grid for a direct comparison. Figure 4 shows the comparison of $XCO_2$ from MPAS-A (top) and CT2019 (middle) and their difference (bottom) for July 1 and December 1 2014 at 00:00 UTC. The figure shows that $XCO_2$ from MPAS-A and CT2019 are generally consistent at the large scales, but differences exist at small spatial scales. The higher horizontal resolution of MPAS-A is evident particularly in July over the northeast and southern China. In December, MPAS-A has higher $XCO_2$ than CT2019 within the Arctic Circle and southern China. Overall the differences between MPAS-A and CT2019 are evident. The magnitude of differences are mostly within 3 ppm, which is similar to the magnitude reported in Polavarapu et al. (2016) for the GEM model. The differences between MPAS-A and CT2019 are expected due to the differences in the two models' horizontal resolution, dynamics, and physical parameterizations. Because no CTM can be expected to have perfect transport, the acceptability of transport is generally judged through comparisons of model simulation with measurements.

6. Two different sampling methods (taking nearest land cell to a given location interpolation in space and time) are used in two different sections. But a discussion on this is missing in the paper.

The statement about interpolation in space and time has been added in the revised manuscript (Page 7 Line 7-18):
*For model-data intercomparison, MPAS-A model data need to be interpolated to the observation space. Following Patra et al. (2008), the model is sampled in the horizontal by taking the nearest cell overland. MPAS-A uses a height-based terrrain-following vertical coordinate (Skamarock et al., 2012). At a given cell, the height of the $k^{\text{th}}$ vertical layer boundary is denoted as $z_k^h$. The height of the layer center is $z_k = 0.5 \times (z_k^h + z_{k+1}^h)$. In MPAS-A, horizontal wind fields are defined at the vertical layer boundaries and $CO_2$ fields are defined at layer centers. For horizontal wind fields validation using radiosonde data (Sect. 3.3.1), the column profile of air pressure and horizontal wind fields defined at layer boundaries are used to interpolate to the measurements' pressure levels. For comparison with near-surface $CO_2$ observations from in-situ stations (Sect. 3.3.3) and aircraft observations (Sect. 3.4), model $CO_2$ defined at layer centers are interpolated to the measurement heights. Vertical interpolation and integration for the comparison with TCCON $XCO_2$ are described in Sect. 3.3.4. MPAS-A simulation outputs are saved at 1-hour intervals. For comparison with radiosonde observations and near-surface $CO_2$ observations, no temporal interpolations are applied: observations are paired with the closest hourly MPAS-A output. For comparison with aircraft observations, the hourly model outputs that bracket an observation's time stamp are used for the temporal interpolation.*

**Specific comments**

1. Despite the developed model is named MPAS-A in the title, MPAS-A and MPAS are flurried through the manuscript, making confusion to readers.

Thank for pointing this out.

In the revised manuscript, the model is consistently referred to as MPAS-A.

2. P1L6: "only major research and operation centers can afford it" would be not necessary because the definition is somewhat vague. This is the same in the introduction.

Agreed.

This statement has been removed from the abstract and introduction.

3. P1L15: Why all hourly data is used? The statistics for sure is exacerbated by including night-time data.

We agree with the reviewer that the statistics are exacerbated by including the nighttime data. Because we compare MPAS-A statistics with ECWMF IFS performance statistics, which are only reported as using all-hourly data in Agusti-Panareda et al. (2019), we also use all-hourly data for calculating the statistics.

4. P1L21: The conclusion of comparisons with WRF-Chem and CT2019 is missing

Thanks for pointing this out.

In the revised manuscript, the following statements have been added in the revised manuscript (Page 1 Line 17-21):

*The regional scale evaluations show that MPAS-A is capable of representing the observed atmospheric $CO_2$ spatial structures related with the mid-latitude synoptic weather system, including the warm versus cold sector distinction, boundary layer to free troposphere difference, and frontal boundary $CO_2$ enhancement. MPAS-A's performance in representing these $CO_2$ spatial structures are comparable with the global model CT2019 and regional model WRF-Chem.*

5. P1L24: I don't think "often" is necessary for this sentence.

Agreed. The word "often" has been removed from this sentence.

6. P1L24: It needs to use surface CO2 flux rather than CO2 flux because there are other usages for CO2 fluxes used throughout the manuscript (e.g., horizontal CO2 fluxes and vertical CO2 fluxes).

Thanks for pointing this out.

"$CO_2$ fluxes" has been replaced with "surface $CO_2$ fluxes".

7. P2L7-8: It needs to add a reference for the sentence.

Agreed. Baker et al. (2006) has been added as a reference for the importance of data-model-mismatch partition.

*Baker, D. F., Doney, S. C., and Schimel, D. S.: Variational data assimilation for atmospheric CO2, Tellus B, 58, 359–365 2006.*

8. P2L11: What does "simulation resolution" mean by?

Thanks for pointing this out.

"simulation resolution" has been replaced with "horizontal resolution a simulation".

9. P2L18-19: This sentence is a little bit confusing. A regional model could require more computational cost than a global model does. Because the cost of a model depends on the configuration (e.g. the number of grids and the period of interest, etc.). Thus, it needs to clarify it.

Thanks for pointing this out.

The sentence has been revised in the revised manuscript (Page 2 Line 17-20):

*Regional (limited area) models, which have lower computational cost than their global model counterpart at the same horizontal resolution, are often used for high resolution $CO_2$ transport and inverse modeling.*

10. P3L23: It seems that MPAS CO2 is an online transport model. But it is not mentioned explicitly in the paper.

Thanks for pointing this out.

The following statement has been added to the revised manuscript (Page 3 Lines 10-11):

*Because the $CO_2$ transport processes are fully integrated into the model's meteorological time steps, the resulting MPAS-A $CO_2$ is an online CTM.*

11. P5L20: Does "CO2 eddy diffusivity" mean vertical eddy diffusivity for CO2? And is there any cap (minimum or maximum) applied to $K_h$ to prevent too strong or weak vertical mixing?

Thanks for pointing this out.

"$CO_2$ eddy diffisivity" has been replaced with "vertical diffusivity for $CO_2$". The following statement has been added to the revised manuscript (Page 5 Lines 15-16):

*We use the same value for $CO_2$ vertical diffusivity as water vapor. The details of $K_h$ calculation can be found in the appendix of Hong et al. (2006), and its value is limited between 0.01 and 1000 $m^2s^{-1}$ to prevent too weak or strong vertical mixing.*

12. P7L16: What does third-order advection mean?

Thanks for pointing this out.

"third-order advection scheme" has been replaced with "third-order accuracy advection scheme" in the revised manuscript (Page 7 Line 27).

13. P7L17: Table 1, are parameterisation schemes here an optimal choice for the best CO2 simulation by MPAS? An explanation of why these parameterisations are used in the simulation is lacking.

Thanks for pointing this out.

The following statements have been added to the revised manuscript (Page 3 Lines 28-30):

*We choose to implement $CO_2$ vertical mixing in the Yonsei University (YSU) PBL scheme (Hong et al., 2006), and $CO_2$ convective transport in Kain-Fritsch (KF) scheme (Kain, 2004) because they are widely used in CTM and have been validated using observations (Borge et al., 2008; Hu et al., 2010; Kretschmer et al., 2012; Polavarapu et al., 2016).*

14. P7L17: How do parameterisation schemes work for the weather forecast and CO2 transport on a variable-resolution grid? Is it anticipated that parameterisation schemes work identically with the different area of a grid?

To address the reviewer's comment, the following sentence has been included in the revised manuscript (Page 6 Lines 3-5):

*Because the calculation of the updraft and downdraft mass fluxes is related to a cell's horizontal area, the KF scheme may behave differently at different areas of MPAS-A's variable-resolution grid.*

15. P7L19: Because of the difference in grid between ERA-I and MPAS, how is the conservation of tracer mass guaranteed after interpolation of wind fields to the MPAS grid? Is there any kind of "pressure fixer" as used in GEOS-Chem or a flux adjustment as done by CarbonTracker (Segers et al., 2002)?

Thanks for pointing this out.

After the meteorological fields (including wind field interpolation) of MPAS-A are initialized using ERA-I, a spatial-invariant scaling factor is applied prior to the start of the subsequent 24-hour model simulation. This approach is similar to the one used in Polavarapu et al. (2016) and is described in Section 3.3.2 of the revised manuscript.

The Semi-Lagrangian advection scheme, as used in ECCC GEM-MACH (Polavarapu et al., 2016), ECMWF IFS (Agusti-Panareda et al., 2014) and TM5 (Krol et al., 2005), do not conserved mass (Williamson, 1990), thus requires a flux adjustment (Segers et al., 2002) or a global mass fixer (Diamantakis and Flemming, 2014). In contrast, MPAS-A uses the explicit grid point advection scheme which conserves mass (Skamarock and Gassmann, 2011). Therefore, neither pressure fixer nor a mass fixer is applied to MPAS-A $CO_2$ transport. The evaluation of dry air and $CO_2$ mass conservation (Section 3.3.1) confirms MPAS-A's mass conservation property.

The following statements have been added to the manuscript (Page 8 Line 27-Page 9 Line 3):

*In comparison, the total dry air mass of ECMWF IFS increases about* $0.01\%$ *of its initial value in a 10-day forecast (Diamantakis and Flemming, 2014). Similarly, the Environment and Climate Change Canada (ECCC) Global Environmental Multiscale (GEM) model loses about* $0.01\%$ *of its initial total dry air mass in a 10-day forecast (Polavarapu et al., 2016). MPAS-A has a significantly lower global dry air mass variation than the two global models because its explicit grid point advection scheme conserves mass (Skamarock and Gassmann, 2011) while the semi-Lagrangian advection scheme used by IFS and GEM does not conserves mass (Williamson, 1990). Thus, no mass fixer (Diamantakis and Flemming, 2014; Polavarapu et al., 2016) is used in MPAS-A.*

16. P8L25, Fig. 2: Are fluxes used in these simulations? If not, the simulation should be repeated with CO2 surface flux inputs. This is important because the surface fluxes create strong gradients which challenge a model's numerical schemes. Without surface fluxes, the CO2 field is very smooth and easy to model. Also, can you increase the length of the simulation?

We agree with the reviewer that surface $CO_2$ flux should be ingested during MPAS-A simulation for the mass conservation test.

Following the reviewer's suggestion, we conducted new simulations that ingesting CT2019 $CO_2$ flux at 3-hour intervals for a one-year (2014) period. The mass conservation of dry air and $CO_2$ in the is quantified as expressed as a ratio to their initial values are plotted in Figure 2 of the revised manuscript. This figure is also included here for reference. The global CO2 mass variation for the one-year simulation is still quite small, in a magnitude about $10^{-5}$ (which is indeed significantly higher than that from the simulation without ingesting surface $CO_2$ flux as reported in the original manuscript). The above simulation and mass conservation results are included in the revised manuscript (Page 8 Lines 15-30):

[Figure]

**Figure caption** Variation of total dry air mass (a) and total $CO_2$ mass (b) as the ratio to their respective starting values during a 1-year continuous MPAS-A simulation without meteorological re-initialization. The X-axis represents the number of hours after the start of the simulation, and the Y-axis the ratio of the total mass change to the starting values.

17. P8L28: What does maximal variation mean? The change from one data point to the next? If so, it seems much smaller than 1.95E-12 kg/m^3 for dry air in Fig. 2. If it is the maximum value in the plot, it seems to low since the maximum is around 4E-12.

Thanks for pointing this out.

The mass conservation figure has been re-plotted using the one-year simulation result (see our response to comment #16). The following statements have been added to the revised manuscript (Page 9 Lines 11-15):

*the variation of global mass of $CO_2$ is quantified as a ratio, $E_{co_2}^t = (M_{co_2}^t - M_{co_2}^0)/M_{co_2}^0$, where $M_{co_2}^0$ and $M_{co2}^t$ are the model's initial and current time step global $CO_2$ mass. $E_{co_2}^t$ at 00:00 UTC of each day of the simulation period is shown in the lower panel of Fig. 2. The figure shows that the maximal magnitude of $E_{co_2}^t$ is about $10^{-5}$. This is much higher compared to $E_{air}^t$ and it is due to the strong gradients caused by surface $CO_2$ flux which challenge the model's numerical scheme.*

18. P9L11: It sounds like Dee et al. mentioned about dry air mass change. Since the reference is already cited in the paper. This one could be taken off.

We apologize for the confusion. Dee et al. (2011) is used here as a reference for the ERA-Interim reanalysis. It does not mention dry air mass change when used for re-initializing meteorological fields for $CO_2$ transport. Because this statement is needed here to explain how the re-initialization impacts $CO_2$ mass conservation, it is kept in the revised manuscript.

19. P9L26, Fig. 3: This test should be repeated with realistic CO2 surface fluxes. You know what the flux is over 3 hours so the change in global CO2 mass is known. Then the expected total CO2 from fluxes, assuming an initial value, can be compared with the model global CO2 mass. Fluxes create strong gradients that are usually challenging for the numerics, and you will see more realistic magnitudes of temporal variations. Those changes in Fig. 3 may be unrealistically low. Also 48 days is too short. Tests for a year would be good.

Agreed. Following the reviewer's suggestion, we conducted a new MPAS-A simulation for an one-year period (2014) with CT2019 surface $CO_2$ fluxes ingested at 3-hour intervals. The resulting global $CO_2$ mass changes in the new simulation is indeed substantially higher. Please see our our response to the reviewer's specific comment 16 and Section 3.2.1 of the revised manuscript for more details.

20. P10L8-17: the paragraph does not include information about section 3.3.1. Please revise the text.

Thanks for pointing this out. Information about Wind field validation (Section 3.3.1) has been added in the revised manuscript (Page 10 Lines 17-19):

*First MPAS-A simulated horizontal wind fields are evaluated using radiosonde measurements from 457 stations. Then the model's $CO_2$ fields are compared with CT2019, near-surface $CO_{2_2}$ measurements from 50 stations, and XCO2 retrievals from 18 TCCON stations.*

21. P10L16: The current configuration heavily relies on the IC from CarbonTracker, due to the short period of simulation. It does not show the quality of CO2 transport by MAPS CO2 on a global scale. This is another reason why the simulation period should be extended.

Agreed.

Following the referee's suggestion, we conducted a new simulation for model validations using the ACT airborne $CO_2$ measurements. The new simulation is a continuous run which starts at Jan 1 2016 and ends at May 31 2018. The first 6 months is for the model spin-up (In the revised manuscript, Page 14 Lines 10-12).

22. P10L19: The conclusion for section 3.3.1 is missing.

Thanks for pointing this out.

In the revised manuscript, conclusion of section 3.3.1 (horizontal wind field validation) has been added.

23. P10L20-21: It is not clear why this sentence is here. Has Michaelis et al. verified MPAS simulation already? If so, please rephrase it.

The sentence with the reference to Michaelis et al. (2019) has been removed.

24. P10L23: For the verification at 00 UTC, are 24 h forecasts at 0 UTC used? Since the meteorological re-initialisation at the 24-hour interval, it needs to clarify it.

Thanks for pointing this out.

The wind field verification at 00 UTC are the 24-hour forecast after the re-initialization at the previous day's 00 UTC. In the revised manuscript, a clarification regarding this issue has been added (Revised manuscript, Page 10 Lines 26-27).

25. P10L24: Fig. S3 is referred to earlier than Fig. S2.
Thanks for pointing this out.
It has been corrected in the revised manuscript.

26. P10L26: Table 6 is referred to earlier than Table 3, 4 and 5.
Thanks for pointing this out.
It has been corrected in the revised manuscript.

27. P11L3: Because section 3.3.2 utilizes data focused on North America, not on the globe, the title should be revised accordingly.
Agreed. The section title has been changed to "Model evaluation at regional scale"

28. P11L9: "regional NWP WRF" sounds strange. Please clarify it.
Thanks for pointing this out.
"Regional NWP WRF" has been replaced with "Regional model WRF".

29. P11L14: Which "the analysis" do you mean? ERA5 reanalysis or ERA-interim analysis?
Thanks for pointing his out.
This sentence has been revised to clarify that the WRF-Chem simulation uses meteorological nudging and re-initialization to keep its meteorological fields close to the ERA5 reanalysis.

30. P12L4: The PBL height between models are probably different, how did you consider that in Fig. 4 in order to obtain a fair comparison?
We agree with the reviewer about the impact of model simulated PBL height on $CO_2$ simulation. This is indeed a difficult problem. In the revised manuscript, we added statements to point out the difference in predicted PBL among the model impact near-surface CO2 accuracy comparison. (Revised manuscript: Page 15, Lines 17-19)

31. P12L4-5: Flux error would be not the main culprit. It can be also attributed to the larger error in the weather forecast in BL than in FT, associated with the accuracy of PBL height in the model simulation.
We agree with the reviewer.
A statement about the impact of the model predicted PBL on the simulated $CO_2$ accuracy has been added to the revised manuscript. Please see our response the comment #30.

32. P12L8: Although Fig. 4 shows dots with different colours to represent results from different seasons, the differences between seasons are not explained explicitly in the text. This is the case in Fig. 5.
Thanks for pointing this out.
In the revised manuscript, we added a description about the three models' accuracy difference for each of the four campaign seasons. This description is based on Taylor diagram following the reviewer's suggestion. Please see our response to the reviewer's specific comment #35 for more details.

Thanks for pointing this out.

In the revised manuscript, the comparison of simulated $CO_2$ between WRF-Chem and MPAS-A has been clarified based on the Taylor diagram (see our response to specific comment #35) and the above statement does not apply anymore.

We think this is related with the level of variability (as measured by standard deviation) between the coarser resolution model (CT2019) and the higher resolution model (MPAS-A). CT2019 has a lower mean RMSE than MPAS-A, but it also has a substantially lower standard deviation, which is at least partially attributable to the model's coarser horizontal resolution ($3° \times 2°$) (page 15 Lines 19-21):

Agreed. It indeed makes interpretation much easier and more concise.

Following the reviewer's suggestion, for each ACT campaign seasons, all the model-data pairs from all flights are combined. The results for each season are further divided into two groups: boundary layer (BL) and free troposphere (FT). The model-data comparisons are then summarized in two sets of Taylor diagrams: one for BL and another for FT. The two figures are shown below and included in the revised manuscript.

[Figure]

Figure 1: Taylor diagram for model evaluation using the ACT airborne BL measurements.

[Figure]

Figure 2: Taylor diagram for model evaluation using the ACT airborne FT measurements.

36. P13L10: In Fig. 6, please add names on the x-axis to make easy to understand the figure and add the unit on y-axis or in the caption.

Thanks for pointing this out.

Names have been added to the x-axis, and unit has been added to the y-axis.

[Figure]

37. P13L13, Table 4: I don't see this date (2016-08-24) in Table 4. There are cases where CT2019 does better than the other 2. The date mentioned in the text may be cherry-picked. To get a better sense of how often CT2019 does better/worse than the higher resolution models, please list the differences between warm and cold in a separate column for easy comparison between the models and observations.

Thanks for pointing this out.

"2016-08-24" was a typo, and the sentence has been deleted. With the new simulation results, the original table 4 has been remade to include the mean difference in $CO_2$ between the warm and cold sectors from ACT airborne measurements and the three models (MPAS-A, WRF-Chem, and CT2019). The standard deviation columns included in the original manuscript have been removed to avoid cluttering. In the new table, the column labeled "diff" is the warm-cold sector $CO_2$ difference.

| date | ACT | | | MPAS-A | | | WRF-Chem | | | CT2019 | | |
|---|---|---|---|---|---|---|---|---|---|---|---|---|
| yyyy-mm-dd | warm | cold | diff | warm | cold | diff | warm | cold | diff | warm | cold | diff |
| 2016-07-18 | 396.7 | 396.7 | 0.0 | 396.7 | 402.1 | -5.4 | 392.2 | 398.4 | -6.2 | 397.8 | 399.1 | -1.3 |
| 2016-07-19 | 398.2 | 396.6 | 1.6 | 404.9 | 393.5 | 11.4 | 400.2 | 394.3 | 5.9 | 399.6 | 397.9 | 1.7 |
| 2016-07-25 | 400.8 | 390.5 | 10.3 | 400.4 | 393.4 | 7.0 | 399.3 | 389.9 | 9.4 | 402.0 | 395.2 | 6.8 |
| 2016-07-26 | 405.9 | 396.1 | 9.8 | 419.4 | 394.8 | 24.6 | 424.8 | 394.0 | 30.8 | 408.4 | 396.3 | 12.1 |
| 2016-08-03 | 399.8 | 401.7 | -1.9 | 401.7 | 398.4 | 3.3 | 400.8 | 401.1 | -0.3 | 401.3 | 399.1 | 2.2 |
| 2016-08-04 | 407.3 | 393.5 | 13.8 | 408.2 | 391.3 | 16.9 | 407.5 | 399.8 | 7.7 | 403.0 | 390.7 | 12.3 |
| 2016-08-08 | 412.2 | 385.3 | 26.9 | 422.9 | 386.0 | 36.9 | 405.1 | 383.9 | 21.2 | 407.3 | 392.0 | 15.3 |
| 2016-08-12 | 401.4 | 395.1 | 6.3 | 404.4 | 392.1 | 12.3 | 405.4 | 402.8 | 2.6 | 399.6 | 395.3 | 4.3 |
| 2016-08-20 | 404.0 | 395.1 | 8.9 | 406.2 | 389.2 | 17.0 | 406.6 | 393.3 | 13.3 | 404.3 | 395.2 | 9.1 |
| 2016-08-21 | 406.5 | 390.7 | 15.8 | 408.1 | 387.0 | 21.1 | 414.8 | 392.5 | 22.3 | 404.1 | 394.1 | 10.0 |
| 2017-02-12 | 408.1 | 414.2 | -6.1 | 409.7 | 412.1 | -2.4 | 409.7 | 413.1 | -3.4 | 408.3 | 410.9 | -2.6 |
| 2017-02-17 | 413.5 | 414.8 | -1.3 | 411.8 | 415.1 | -3.3 | 411.5 | 413.2 | -1.7 | 411.5 | 414.0 | -2.5 |
| 2017-02-23 | 409.4 | 419.1 | -9.7 | 411.7 | 417.6 | -5.9 | 409.8 | 428.2 | -18.4 | 409.6 | 415.0 | -5.4 |
| 2017-03-07 | 412.0 | 415.2 | -3.2 | 415.3 | 417.1 | -1.8 | 417.4 | 418.4 | -1.0 | 413.7 | 413.8 | -0.1 |
| 2017-03-10 | 410.8 | 413.5 | -2.7 | 413.5 | 415.3 | -1.8 | 414.2 | 416.2 | -2.0 | 412.3 | 413.4 | -1.1 |

38. P13L19: The explanation about the sampling strategy is missing in this section, in particular vertical sampling method. Results might be sensitive to the method.

Thanks for pointing this out.

In the revised manuscript, we added an explanation for sampling strategy, including the vertical and horizontal samplings. Linear interpolation to the observation height above the surface is performed in height. This is similar to (Agusti-Panareda et al., 2014) (their appendix A) except that because MPAS-A uses height-based vertical coordinates (IFS has a hybrid-sigma pressure vertical grid) there is no need to calculate the height at the layer centers except a linear interpolate from layer surface to layer center. The horizontal interpolation is to use the nearest grid cell to the location of the observation.

39. P14L20, Table 5: The sample sizes are quite small so some significance test would be useful to evaluate if the differences in scores between cases are important. In fact, when comparing scores between models, in other Tables as well, significance test would be useful.

We agree with the reviewer.

Following the reviewer's suggestion, we carried out paired $t$ tests based on the Absolute Error (AE), which is defined as the absolute of FT-BL $CO_2$ difference between a model simulation and ACT observations:

$$\mathrm{AE} = |\Delta[CO_2]_{\mathrm{model}} - \Delta[CO_2]_{\mathrm{obs}}|$$

and Mean Absolute Error (MAE) is:

$$\mathrm{MAE} = \frac{1}{N} \sum_{i=1}^{N} \mathrm{AE}_i$$

Paired $t$ tests are conducted based on the absolute errors of the three models. The results are summarized in the table below for each season. Using $p = 0.1$ as the cut-off value the table shows that: (1) MPAS-A has smaller MAE than CT2019 in fall 2017 and larger MAE in summer 2016. The differences between the two models in the other two seasons are not significant. (2) MPAS-A has smaller MAE than WRF-Chem in winter 2017 and spring 2018. The differences between the two models in the other two seasons are not significant.

**Table caption**:Mean absolute error (MAE) of MPAS-A, WRF-Chem and CT2019 validated using ACT aircraft $CO_2$ measurements from four campaign seasons. $p$ value of paired $t-$test between MPAS-A and the other two models given for each season to provides a significance level for the model comparisons.

| | Number of | Mean Absolute Error | | | p-value of paired t test | |
| | | | | | MPAS-A vs | MPAS-A vs |
| Season | profiles | MPAS-A | WRF-Chem | CT2019 | CT2019 | WRF-Chem |
| --- | --- | --- | --- | --- | --- | --- |
| Summer 2016 | 72 | 3.80 | 4.38 | 3.03 | 0.06 | 0.21 |
| Winter 2017 | 27 | 1.56 | 2.23 | 1.58 | 0.95 | 0.09 |
| Fall 2017 | 41 | 2.55 | 3.25 | 3.16 | 0.04 | 0.11 |
| Spring 2018 | 59 | 2.29 | 3.75 | 1.99 | 0.23 | 0.01 |

The above table is included in the revise manuscript as Table 12, and the following statements have been added at Page 16 Lines 15-21:

*Table 12 summarize the MAE of the three models for each season. The p values of paired t tests of AE between MPAS-A and the other two models are also listed in the table to provide the significance level of the model comparisons. Using p = 0.1 as the cut-off value the table shows that MPAS-A has smaller MAE than CT2019 in fall 2017 and a larger MAE in summer 2016. The differences between the two models in the other two seasons are not significant. Compared to WRF-Chem, MPAS-A has smaller MAEs in winter 2017 and spring 2018 while the differences in the other two seasons are not significant.*

40. P14L21-23: The sentence is difficult to understand, please rephrase it
Thanks for pointing this out.
In the revised manuscript, mean absolute error (MAE) has been used for model comparisons so that significance levels can be provided following the reviewer's comment (#39). The sentence in question has been replaced in the revised manuscript (Pagge 16 Lines 15-21).

41. P14L32, Fig. 8: Presumably, this figure is shown as an example to demonstrate how MPAS CO2 simulates the CO2 enhancement well at the specific date. Since overall CO2 enhancement

is mentioned in Fig. 9. It may be better to swap Fig. 8 and Fig. 9 to make structure organized better.

Agreed. The two figures have been swapped.

42. P14L34: It is the wrong figure number. Please change Fig. S2 and Fig. S3.

Thanks for pointing this out.

It has been corrected in the revised manuscript.

43. P15L3: Numbers in Fig. 8 are very difficult to read. Please change the font size or colour or both.

Thanks for pointing this out.

The font size in Fig 8 has been increased in the revised manuscript.

44. P15L13: It is difficult to find when the CO2 enhancement is shown. Maybe it would be helpful to add a mark in Fig. 9 to indicate when the CO2 enhancement happened. Otherwise, as there are too many panels in the single figure, you may consider to keep the small number of panels in Fig. 9 and to move the rest of them to supplementary.

Agreed. Figure 9 has been re-plotted to include a smaller number of cases. In addition, dashed lines have been added in the figure to indicate the $CO_2$ enhancement in each case (the dashed lines in the figure below). The figure that includes all cases has been moved to the supplement (Fig. S3).

[Figure]

**Figure caption**: Comparison of $CO_2$ mole fraction in frontal-crossing level-leg flights in boundary layer between ACT aircraft measurements and model simulations. Flight date and aircraft type are labeled in title for each flight leg. X-axis is UTC time, and Y-axis is $CO_2$ mole fraction (ppm). Aircraft measurements are in black, MPAS-A in red, WRF-Chem in blue, and CT2019 in green. In each figure, the pair of vertical dashed lines mark $CO_2$ enhancement observed by the aircraft along a frontal boundary, and the warm and cold sectors associated with the frontal boundary are labeled as warm and cold, respectively.

45. P15L17: It is difficult to recognize colour lines without information. Please add a legend for lines in Fig. 9.
Thanks for pointing this out.
A legend for lines has been added (See the figure in our response to reviewer specific comment #44).

46. P15L18: I cannot see the date 2017-10-28 in Fig. 9.
Thanks for pointing this out.
"2017-10-28" was a typo, which has been removed. The sentence has been revised (Page 18 Lines 5-6): *For instance, there is not clearly identifiable $CO_2$ enhancement in the B200 flights on 2018-04-23.*

47. P15L19: Looking at the date 2017-11-03, the CO2 enhancements happen not only at the front boundary but also around other locations such as at 19 and 19.8 UTC in the panel. Can you explain why CO2 enhancements are shown at other times either? Isn't the CO2 enhancement a unique feature happening at the front?
There are indeed cases whereby a secondary enhancement is observed in aircraft measured $CO_2$. But the elevated $CO_2$ around 19.8 UTC in the 2017-11-03 is caused by a substantial flight altitude change which was not filtered in the original figure. The figure has been remade to exclude the flight the section where the airplane altitude was substantially lower.

48. P15L25: What is the "level-leg flights"?
Thanks for pointing this out. All "level-leg flights" has been replaced with "constant altitude flight segments" in the revised manuscript.

49. P16L3: Because the developed model is a global model, it would be better to evaluate result over the globe first then narrow down the scales of interest (North America). In that sense, section 3.4 should be placed before section 3.3. Also, including more observations (e.g., TCCON) in section 3.4 is beneficial to evaluate the behaviour of the global scale CO2 transport by MPAS CO2, with an extended simulation period.

We agree.
Following the referee's suggestion, we have: (1) re-organized the text so that global validation (using the near-surface towers, TCCON, and CT2019) is presented before the regional validation (using ACT airborne measurements).
(2)compared MPAS-A simulation with CT2019 mole fraction. The details of this comparison is in our response to the referee's major comment 5.
(3) validated MPAS-A using $XCO_2$ at 18 TCCON sites. The results are included Section 3.3.4 of the revised manuscript.

The following figure and table has been added in the revised manuscript as Figure 7 and Table 8, respectively.

[Figure]

**Figure Caption**: MPAS-A simulated hourly $XCO_2$ at 18 TCCON sites for the year of 2014.

**Table caption**:Statistics for the average hourly $XCO_2$ and average daily $XCO_2$ comparison between TCCON measurements and MPAS-A simulations: RMS (ppm), bias (ppm), and correlation coefficient $R$. $N$ is the number of data pairs used for computing of the statistics.

| Site | Hourly mean $XCO_2$ | | | | Daily mean $XCO_2$ | | | |
| --- | --- | --- | --- | --- | --- | --- | --- | --- |
| | N | RMSE (ppm) | Bias (ppm) | R | N | RMS (ppm) | Bias (ppm) | R |
| Ascension Island | 1135 | 0.99 | 0.66 | 0.79 | 227 | 0.92 | 0.68 | 0.83 |
| Bialystok | 648 | 1.67 | 0.71 | 0.89 | 133 | 1.52 | 0.79 | 0.92 |
| Bremen | 212 | 2.15 | 0.91 | 0.86 | 60 | 2.04 | 1.10 | 0.89 |
| Darwin | 1,956 | 1.10 | 0.77 | 0.69 | 306 | 1.00 | 0.70 | 0.78 |
| Edwards | 1,884 | 0.97 | 0.48 | 0.90 | 274 | 0.87 | 0.45 | 0.92 |
| Garmisch | 547 | 1.08 | 0.03 | 0.91 | 114 | 1.08 | 0.07 | 0.91 |
| Izana | 200 | 0.45 | 0.14 | 0.95 | 72 | 0.41 | 0.14 | 0.96 |
| Saga | 518 | 1.13 | 0.23 | 0.87 | 114 | 1.10 | 0.22 | 0.87 |
| Karlsruhe | 489 | 2.15 | 0.92 | 0.81 | 102 | 1.70 | 1.07 | 0.89 |
| Lauder | 652 | 1.10 | 0.88 | 0.86 | 167 | 1.03 | 0.83 | 0.89 |
| Lamont | 2,014 | 1.29 | 0.37 | 0.85 | 295 | 1.21 | 0.38 | 0.87 |
| Orleans | 591 | 2.00 | 0.86 | 0.74 | 144 | 1.68 | 0.85 | 0.81 |
| Parkfalls | 1,164 | 1.38 | -0.01 | 0.93 | 208 | 1.24 | 0.00 | 0.94 |
| Reunion Island | 1,059 | 0.94 | 0.76 | 0.92 | 193 | 0.91 | 0.77 | 0.94 |
| Rikubetsu01 | 183 | 1.22 | -0.09 | 0.94 | 58 | 1.11 | 0.02 | 0.95 |
| Sodankyla01 | 227 | 1.35 | 0.80 | 0.96 | 95 | 1.36 | 0.86 | 0.96 |
| Tsukuba02 | 969 | 2.27 | -0.03 | 0.68 | 172 | 1.93 | 0.03 | 0.75 |
| Wollongong | 961 | 1.18 | 0.72 | 0.76 | 208 | 1.14 | 0.73 | 0.79 |

50. P16L14: What is the difference between Mean RMSE vector wind and RNSE wind speed in Table 3? Since RMSE wind speed is not mentioned in the text, it is difficult to understand why two different metrics are shown separately.

Thanks for pointing this out.

RMSE of wind speed has been removed from the horizontal wind fields validation.

51. P16L14-15: I am just curious that numbers for the mean difference wind direction in Table 3 are always positive by any chance.

Mean difference in wind direction in the table is calculated as

$$\frac{1}{N}\sum_{i=0}^{N}|\theta_i^m - \theta_i^d|,$$

Where $\theta_i^m$ is the model simulated wind direction and $\theta_i^d$ is wind direction from radiosonde measurements. Because of the absolute value used in the calculation ($|\theta_i^m - \theta_i^d|$) ranges from 0° to 180°. This explains why the mean differences of wind direction are always positive.

52. P16L15: Table 3 is mentioned later than Table 4 and 5. Please change the order.

Thanks for pointing this out.

The order of Tables 3, 4, and 5 has been corrected in the revised manuscript.

53. P16L25-26: Vertical sampling method is missing. It is better to add a discussion about vertical sampling strategy. In addition, why is the horizontal sampling method used in this section different from that in the previous section?

Thanks for pointing this out. A paragraph has been added in the revised manuscript (Page 7 Line 7-18):

*For model-data intercomparison,MPAS-A model data need to be interpolated to the observation space. Following Patra et al. (2008), the model is sampled in the horizontal by taking the nearest cell overland. MPAS-A uses a height-based terrain-following vertical coordinate (Skamarock et al., 2012). At a given cell, the height of the $k^{\text{th}}$ vertical layer boundary is denoted as $z_k^h$. The height of the layer center is $z_k = 0.5 \times (z_k^h + z_{k+1}^h)$. In MPAS-A, horizontal wind fields are defined at the vertical layer boundaries and $CO_2$ fields are defined at layer centers. For horizontal wind fields validation using radiosonde data (Sect. 3.3.1), the column profile of air pressure and horizontal wind fields defined at layer boundaries are used to interpolate to the measurements' pressure levels. For comparison with near-surface $CO_2$ observations from in-situ stations (Sect. 3.3.3) and aircraft observations (Sect. 3.4), model $CO_2$ defined at layer centers are interpolated to the measurement heights. Vertical interpolation and integration for the comparison with TCCON $XCO_2$ are described in Sect. 3.3.4. MPAS-A simulation outputs are saved at 1-hour intervals. For comparison with radiosonde observations and near-surface $CO_2$ observations, no temporal interpolations are applied: observations are paired with the closest hourly MPAS-A output. For comparison with aircraft observations, the hourly model outputs that bracket an observation's time stamp are used for the temporal interpolation.*

54. P17L8: Because MPAS utilizes variable horizontal resolution (60-15 km), it may be possible to find a benefit of higher horizontal resolutions by splitting results into two groups, one on 60 km grid and the other on 15 km. Results at sites on 15 km grid may be comparable with results in IFS 9 km.

Following the reviewer's suggestion, the following paragraph and table describing the differences between the stations located at the 15 km and 60 km cells are added. In the revised manuscript (Page 12 Line 33-Page 13 Line 9), the following paragraph has been added along with the table (Table 6) below:

*Agusti-Panareda et al. (2019) found that atmospheric $CO_2$ transport is generally better represented at higher horizontal resolutions, and mountain stations display the largest improvement at higher resolution as they directly benefit from the more realistic orography. There are 12 mountain stations of the 50 stations used for the model validation. Table ?? lists the 12 mountain stations in two groups: the first group includes the six mountain stations located at the 15 km cells of the MPAS-A's 60-15 km variable-resolution grid, and the second group includes the other six stations that are located at the 60 km cells of the grid. The table lists the hourly $CO_2$ RMSE for each of the 12 stations from MPAS-A and IFS 9 km and 80 km simulations are listed for January and July 2014. The table shows that at each of the six mountain stations located at 15 km cells, MPAS-A has lower hourly $CO_2$ RMSE than the IFS 9 km simulation for July 2014. For January 2014, MPAS-A has lower RMSE than IFS 9 km simulation at five out the six stations (the exception is NWR). In comparison, at the six mountain stations located at its 60 km cells, MPAS-A has higher hourly $CO_2$ RMSE than IFS 9 km simulation for both January and July of 2014 with the exception of JFJ for July 2014.*

**Table caption**: Comparison of RMSE of hourly $CO_2$ between the MPAS-A 60-15 km simulation and the IFS 9 km and 80 km simulations at 12 mountain sites. The left half of the table is for six mountain sites located in MPAS-A's 15 km cells and the second half is for six mountain sites located in MPAS-A's 60 km cells. The top half of the table is for January 2014 and the bottom half is for July 2014.

| | | Sites at MPAS-A 15 km cells RMSE | | | Sites at MPAS-A 60 km cells RMSE | | | |
| --- | --- | --- | --- | --- | --- | --- | --- | --- |
| | Site | IFS 9km | IFS 80 km | MPAS-A | Site | IFS 9km | IFS 80 km | MPAS-A |
| January 2014 | HDP | 3.10 | 19.71 | 1.17 | KAS | 4.44 | 10.71 | 16.65 |
| | SPL | 3.95 | 4.43 | 1.36 | SSL | 5.83 | 23.99 | 14.74 |
| | NWR | 1.64 | 3.74 | 1.78 | JFJ | 2.53 | 15.55 | 5.91 |
| | SNP | 5.01 | 14.54 | 4.36 | PUY | 4.58 | 10.30 | 5.94 |
| | IZO | 2.80 | 1.16 | 2.00 | VAC | 1.10 | 2.28 | 1.62 |
| | MLO | 0.85 | 1.25 | 0.77 | GIC | 5.60 | 4.74 | 7.40 |
| July 2014 | HDP | 5.99 | 37.37 | 2.92 | KAS | 4.29 | 17.57 | 7.17 |
| | SPL | 10.79 | 26.32 | 4.09 | SSL | 8.99 | 20.91 | 18.15 |
| | NWR | 5.17 | 18.78 | 3.71 | JFJ | 6.35 | 11.93 | 4.83 |
| | SNP | 29.28 | 48.33 | 12.88 | PUY | 7.23 | 13.29 | 12.80 |
| | IZO | 6.01 | 2.88 | 3.69 | VAC | 5.95 | 13.91 | 7.76 |
| | MLO | 1.47 | 1.68 | 1.31 | GIC | 20.30 | 15.36 | 28.58 |

55. P17L8: Why are 46 stations used in the calculation rather than 50 stations? This is a different number from what is mentioned above (50 stations).
This was a typo.
It has been corrected to 50 stations.

56. P17L10: Table S2 is mentioned later than Table S3.
Thanks for pointing this out.
The order has been corrected in the revised manuscript.

57. P17L17: The sentence sounds strange. Please rephrase it.
Thanks for pointing this out.
This sentence has been rephrased in the revised manuscript (Page 18 Lines 32-33):
*We implemented the $CO_2$ atmospheric transport processes, including advection, vertical mixing, and convective transport, in the global variable-resolution model MPAS-A.*

58. P37: Figure 10. In the caption, what does "Figs. 9 and 9" mean?
Thanks for pointing this out.
It was a typo and has been corrected in the revised manuscript.

**Technical corrections**

1. Overall, it was able to find lots of typos and technical corrections, grammar and format issues. So, presented here might be not the completed set.

Thanks! We have went through careful proofreading for the revised manuscript.

2. The subscript for 2 is missing in CO2 in many places, including the main text, captions and figures, throughout the paper.
All fixed.

3. Many acronyms are defined in wrong places or multiple times. Please correct them.
Double checked and fixed.

4. P2L13: This is the first place to define PBL.
Fixed.

5. P2L28: This is the first place to define FT.
Fixed.

6. P3L13: This is not the first place to define PBL.
Fixed.

7. P3L18: Remove space between ")" and ","
Fixed.

8. P4L23: "planetary boundary layer (PBL)" $\longrightarrow$ PBL
Fixed.

9. P4L24: This is the first place to define BL.
Fixed.

10. P6L12: Carbon dioxide $\longrightarrow CO2$
$Fixed.$

11. P7L14: What is "(?)"?
Fixed.

12. P7L21: Carbon dioxide $\longrightarrow$ CO2
Fixed.

13. P7L24: CO2 fluxes $\longrightarrow$ surface CO2 fluxes
Fixed.

14. P9L4: Fig.2 $\rightarrow$ Fig. 2 (add space)
Fixed.

15. P10L7: It may be a typo of evaluation.
Fixed.

16. P10L14: boundary layer $\longrightarrow$ BL
Fixed.

17. P10L24: 850 is duplicated
Fixed.

18. P11L9: chemistry transport model $\longrightarrow$ CTM
Fixed.

19. P11L11: Missing year for Hersbach et al.
Fixed.

20. P11L12: boundary conditions $\longrightarrow$ lateral boundary conditions
Fixed.

21. P11L12: Jacoboson et al. (2007) is not the proper reference for CarbonTracker.
Fixed.

22. P11L13: CO2 fluxes $\rightarrow$ surface CO2 fluxes
Fixed.

23. P11L14: Jacobson al. (2007) is the not proper reference for CarbonTracker.
Fixed.

24. P11L15: atmosphere $\longrightarrow$ atmospheric
Fixed.

25. P11L16: chemical transport model $\longrightarrow$ CTM
Fixed.

26. P11L17: planetary boundary layer $\longrightarrow PBL$
$Fixed.$

27. P11L21: Remove space between ")" and ","
Fixed.

28. P11L25: change double parenthesis to single
Fixed.

29. P11L32: It is not the first place to define BL and FT.
Fixed.

30. P11L32: he $\longrightarrow$ the
Fixed.

31. P12L6: exception $\longrightarrow$ exceptions
Fixed.

32. P12L9: free troposphere $\longrightarrow$ FT
Fixed.

33. P13L24: "boundary layer (BL) and free toposphere (FT)" $\longrightarrow$ BL and FT
Fixed.

34. P14L15: 2015? It might be a typo.
Fixed.

35. P14L15: Fall $\longrightarrow$ fall
Fixed.

36. P14L15: show $\longrightarrow$ shows
Fixed.

37. P14L17: tend $\longrightarrow$ tends
Fixed.

38. P14L29: boundary layer $\longrightarrow$ BL
Fixed.

39. P15L11: wrong order; 2017-02-23 after 2017-03-10
Fixed.

40. P16L32: "The Schauinsland station" should be moved to P16L29.
Fixed.

41. P17L2: Add space between "23.99" and "ppm".
Fixed.

42. P30: Figure 4: Unit is missing in the caption or figure.
Fixed.

43. P32: Figure 6: Unit is missing in the caption or figure. Add names on x-axis.
Fixed.

44. P43: Table 3. Add space between number and the unit (m/s) and parenthesis for degree. Numbers should be integer.
Fixed.

45. P44: Table 4. Add "date (yyyy-mm-dd)" at the top left.
Added.

46. P45: Table 5. Num $\longrightarrow$ Number
Fixed.

47. P47: Table 7. Make Station IDs capital letters (including main text)
Station IDs have been changed to capital letters.

**References**

[revised manuscript text omitted]

---

## Author Response (AR2)

**Response to reviewer 1**

There are numerous instances of question marks in place of Table or Figure numbers that need to be corrected.

Thanks for pointing this out. The missing Table/Figure numbers (questions marks) were generated by the LaTex in the deleted/replaced portions of the change-track document. The errors have been fixed and they are not in the current version of change-track document, point-by-point response, or the revised manuscript.

**Response to reviewer 2**

I appreciate that the authors made a great effort to improve the manuscript significantly, especially running the model for a longer period of time to obtain a robust result. Many comments are addressed appropriately. However, it was able to find out some minor issues remaining. Plus, although many typos are corrected as the authors declared, still some (or the same) typos can be found again. Nevertheless, I believe that these can be addressed easily.

We appreciate the reviewer's insightful comments and suggestions.

**Specific comments:**

[1] P7L7-L18: This paragraph explains the sampling method used in the paper nicely. However, I am still curious about the sentence "MPAS simulated CO2 fields are interpolated in time and space to match each 5-second airborne data points" at P15L4-5. Because it looks inconsistent with the method explained here (interpolation in space vs. selecting nearest grid point overland). For ACT campaign data, is the sampling method used in section from 3.4.1 to 3.4.3 the same as the method used for sampling near-surface CO2 observations (section 3.3.3)? If they are the same, it needs to revise the sentence. If they are different, then please clarify why they are different.

All the horizontal samplings (including for ACT campaign, near-surface observations, and TCCON $XCO_2$) use the same "selecting nearest grid point overland" as described in P7L7-L18.
To avoid the possible confusion, the sentence at P15L4-5 has been revised to "*MPAS-A simulated $CO_2$ fields are sampled as described in the second paragraph of Section 3 to match the 5-second airborne data points.*" (P15L11-12 of the revised manuscript)

[2] P8L3: the sentence "CO2 mixing ratio is kept unchanged during the meteorology re-initializations" is in contradiction with the CO2 mass conservation method in section 3.2.2. Please clarify it.
Thanks for pointing this out. Since CO mass conservation method during meteorology re-initialization has been described in Section 3.2.2, this sentence (P8L3) is deleted to avoid possible confusion.

[3] P12L2-3: It is hard to understand what is "evident". Please rephrase the sentence.
This sentence has been revised to "*For instance, the difference in horizontal resolution between MPAS-A and CT2019 can be clearly observed in $XCO_2$ in July over both northeast and southern China.*" (P12L10-12 of the revised manuscript)

[4] P12L5-7: Maybe it is better to remind it for readers that both model used the same CO2 fluxes so the difference is only caused by the different model transport.

Agreed. The sentence at P12L6-7 has been revised as to "*Because both models used the same surface CO$_2$ fluxes, the difference in the simulated CO$_2$ fields is caused by the difference in transport: spatial resolution, dynamics, and physical parameterizations.*" (P12L14-16 of the revised manuscript).

[5] P13L1: Is it possible to add 60 km uniform grid result over 15 km cell area (data in Table 3) in the table 6, in order to support the benefit of high-resolution? If you don't have the result for CO2, it is fine with leaving it as is because I don't ask an additional experiment.

Adding CO$_2$ result from the 60-km uniform resolution will require new experiments. Therefore, we choose to leave Table 6 as it is.

[6] P14L31-P15L1: This paragraph looks redundant. Please consider to move it to earlier part (section 3.1 or other proper place).

Thanks for pointing it out. This paragraph has been moved to Section 3.3.2 where CT2019 CO$_2$ fields is used for the first time in the manuscript. (P11L31-34 of the revised manuscript).

[7] P15L6: Which version of ObsPack dataset did you use? Please specify it.

ObsPack v5.0 was used for validation. The version number has been added. (P15L14 of the revised manuscript).

[8] P17L15: Since paired t test is also used in Table 5. The explanation can be moved earlier part of the paper (section 3.3.3), if the same method is used.

The explanation of using paired *t* test for model comparison has been moved to section 3.3.3 (P13L17-18).

[9] P18L30: In summary section, please consider to mention briefly about an additional experiment using 60 km uniform-resolution presented in section 3.3.1.

The following sentence has been added to the summary section "*The horizontal wind fields of the 60-15 km variable-resolution MPAS-A simulation are evaluated at four pressure levels at 457 radiosonde stations. Furthermore, a comparison with an additional 60 km uniform-resolution MPAS-A simulation shows that the accuracy of the horizontal wind fields is substantially higher at the 15 km cells.*" .
(P20L7-9 of the revised manuscript).

[10] P44: Table 2. What does "IGRA" stand for? Since it is not mentioned/explained in the main text. It would be better to add its full name or remove it.

"IGRA" stands for the Integrated Global Radiosonde Archive. "IGRA" has been removed from Table 2 since it is not mentioned in the main text.

**Technical corrections:**
[1] P1L2: CTM -> CTMs
Fixed.

[2] P2L1: CTM -> CTMs
Fixed.

[3] P2L35: model(FV2) -> model (FVS) (add space)
Fixed.

[4] P4L9: If the variable V here is the same one in eq. 1, then you can move this explanation to just after eq. 1.
The explanation of V has been moved from after Eq. 2 to after Eq. 1.

[5] P4L22L: YSU -> YSU scheme
Fixed.

[6] P5L8: YSU ->scheme
Fixed.

[7] P6L1: KF ->scheme
Fixed.

[8] P6L17: It is difficult to understand what "simulation experiments" is. Please revise it.
The sentence has been revised to "In this section we evaluate the newly developed MPAS-A $CO_2$ transport model by comparing its simulation results with observations and other models." . (P6L21-22 of the revised manuscript).

[9] P6L22: ACT -> ACT campaign
Fixed.

[10 ]P8L7: "The four CT2019 fluxes" -> "The four components of CT2019 fluxes"
Fixed.

[11] P8L29: Their model name is actually GEM-MACH-GHG, please check their paper.
GEM has been replaced by GEM-MACH-GHG in all three places in the manuscript.

[12] P9L12: "the model's initial and current time step global CO2 mass" -> "the global CO2 mass at the initial and current time step."
Fixed.

[13] P9L27: "To restore the CO2 mass conservation" -> "To keep the total CO2 mass"
Fixed.

[14] P9L27: MPA-A's -> MPAS-A's
Fixed.

[15] P10L19: CO22 (remove redundant 2)
Fixed.

[16] P10L19: XCO2 (2 for subscript)
Fixed.

[17] P12L6: GEM -> GEM-MACH-GHG
Fixed.

[18] P12L30: "( 11.77" -> "(11.77" (remove space)
Fixed.

[19] P12L31: "a accuracy level" -> "a level of accuracy" or other else
Fixed. It has been changed to "a level of accuracy".

[20] P14L25: project -> campaign
Fixed.

[21] P14L29: analysis -> reanalysis
Fixed.

[22] P14L31: CarbonTracker -> CT2019

[23] P14L33: free troposphere -> FT
Fixed.

[24] P15L6: CarbonTrack -> CT2019
Fixed.

[25] P15L6: CarbonTrack ObsPack -> CarbonTracker ObsPack (or CT2019 ObsPack, depending on the version of dataset you used)
CarbonTrack ObsPack has been replaced by CarbonTracker ObsPack.

[26] P15L14: free troposphere (FT) -> FT
Fixed.

[27] P15L30: boundary layer -> BL
Fixed.

[28] P15L30: free troposphere -> FT
Fixed.

[29] P16L6: boundary layer -> BL
Fixed.

[30] P16L30: boundary layer -> BL
Fixed.

[31] P16L30: free troposphere -> FT
Fixed.

[32] P19L28: CarbonTracker -> CT2019
Fixed.

[33] P19L31: Maybe TCCON dataset information is necessary
TCCON dataset information has been added (P20L13 of the revised manuscript).

[34] P20L14: Maybe proper acknowledgment for TCCON dataset is necessary, depending on their data policy
TCCON acknowledgement has been added (P20L23 of the revised manuscript).

[35] P30: Figure 2.
  1) Please make red solid lines in both panels have the same thickness. It would be good for better visualization.
     Fixed.
  2) "number of hours" looks wrong. Please correct it.
     "number of hours" has been corrected to 'the days of 2014"

[36] P31: Figure 3.
 1) Please make the red solid lines in both panels have the same thickness. It could be good for better visualization.
 Fixed.
 2) "number of days" looks wrong. Please correct it.
    "number of days" changed to "the days of 2014"

[37] P32: Figure 4. XCO2 -> simulated XCO2
Fixed.

[38] P34: Figure 6. Please add period at the end of the caption.
Fixed.

[39] P35: Figure 7.
1)"MPAS-A simulated hourly XCO2" -> "Simulated hourly XCO2 of MPAS-A".
Fixed.
2) please add period at the end of the caption.
Fixed.

[40] P40: Figure12. level-leg flight -> "constant altitude flight segments"
Fixed.

[41] P42: Figure14. "for create" -> "for creating"
Fixed.

[42] P43: Table 1. Please revise the caption.

Fixed.

[43] P44: Table 2. add bracket for the unit (degree) in the 4th column.
Fixed.

[44] P45: Table 3. 1) speed(m/s) -> speed (m/s) (add space), 2) add bracket for the unit (degree) in the 4th column
Fixed.

[45] P50: Table 7. Please revise the caption
The caption has been revised to "TCCON stations used for model evaluations".

[46] P51: Table 8. RMS -> RMSE
Fixed.

[47] P53: Table 10.
 1)  MPAS -> MPAS-A,
 Fixed.
 2)  "MPAS simulated horizontal wind" -> "simulated horizontal wind of MPAS-A",
 Fixed.
 3)  Add a comma after "1000",
 Fixed.
 4)  speed(m/s)-> speed (m/s) (add space),
 Fixed.
 5)  add a bracket for the unit (degree) in the 4th column
 Fixed.

[48 ]P54: Table 11. "2017 winter" -> "Winter 2017"
Fixed.

[49] P55: Table 12. Please revise the first sentence of the caption.
Fixed.